# Two-point boundary correlation functions of dense loop models

**Alexi Morin-Duchesne[1*] and Jesper Lykke Jacobsen[2,3,4†]**

**1** Institut de Recherche en Mathématique et Physique, Université catholique de Louvain, Louvain-la-Neuve, B-1348, Belgium
**2** LPTENS, Ecole Normale Supérieure – PSL Research University, 24 rue Lhomond, F-75231 Paris Cedex 05, France
**3** Sorbonne Universités, UPMC Université Paris 6, CNRS UMR 8549, F-75005 Paris, France
**4** Institut de Physique Théorique, CEA Saclay, F-91191 Gif-sur-Yvette, France

★ alexi.morin-duchesne @ uclouvain.be
† jesper.jacobsen @ ens.fr

## Abstract

We investigate six types of two-point boundary correlation functions in the dense loop model. These are defined as ratios $Z/Z^0$ of partition functions on the $m \times n$ square lattice, with the boundary condition for $Z$ depending on two points $x$ and $y$. We consider: the insertion of an isolated defect (a) and a pair of defects (b) in a Dirichlet boundary condition, the transition (c) between Dirichlet and Neumann boundary conditions, and the connectivity of clusters (d), loops (e) and boundary segments (f) in a Neumann boundary condition.

For the model of critical dense polymers, corresponding to a vanishing loop weight ($\beta = 0$), we find determinant and pfaffian expressions for these correlators. We extract the conformal weights of the underlying conformal fields and find $\Delta = -\frac{1}{8}$, $0$, $-\frac{3}{32}$, $\frac{3}{8}$, $1$, $\frac{\theta}{\pi}(1 + \frac{2\theta}{\pi})$, where $\theta$ encodes the weight of one class of loops for the correlator of type f. These results are obtained by analysing the asymptotics of the exact expressions, and by using the Cardy-Peschel formula in the case where $x$ and $y$ are set to the corners. For type b, we find a $\ln|x - y|$ dependence from the asymptotics, and a $\ln(\ln n)$ term in the corner free energy. This is consistent with the interpretation of the boundary condition of type b as the insertion of a logarithmic field belonging to a rank two Jordan cell.

For the other values of $\beta = 2\cos\lambda$, we use the hypothesis of conformal invariance to predict the conformal weights and find $\Delta = \Delta_{1,2}$, $\Delta_{1,3}$, $\Delta_{0,\frac{1}{2}}$, $\Delta_{1,0}$, $\Delta_{1,-1}$ and $\Delta_{\frac{2\theta}{\lambda}+1,\frac{2\theta}{\lambda}+1}$, extending the results of critical dense polymers. With the results for type f, we reproduce a Coulomb gas prediction for the valence bond entanglement entropy of Jacobsen and Saleur.

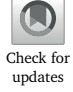

# 1  Introduction

The study of boundary critical phenomena has a long history within the realm of statistical physics that goes back to the heyday of the renormalisation group [1]. More recently, this line of research has been thrusted into the limelight because of its intimate connections with quantum information and entanglement entropy (see [2] and references therein).

The two-dimensional case is of particular interest, since two different approaches offer access to exact results. On one hand, many significant models can be reformulated as integrable one-dimensional quantum spin chains, in which the boundary conditions are taken into account via integrable $K$-matrices [3]. This ultimately leads to exact results for boundary-related properties such as surface critical exponents and surface free energies [4]. Recent results have set the door ajar to obtaining corner free energies in this way [5–8]. The integrable toolbox

can furthermore be employed to compute finite-size corrections, either via the Bethe ansatz technique [9–11] or the approach using functional relations and $Y$-systems [12–15].

On the other hand, conformal field theory (CFT) [16] provides elegant means of obtaining such results directly in the continuum limit [17]. Many of these results have subsequently been made rigorous within the mathematical framework of Stochastic Loewner Evolution (SLE) [18]. The CFT approach highlights the role of conformally invariant boundary conditions and of the so-called *boundary condition changing operators*, which mark the change from one conformally invariant boundary condition to another. The CFT methods can also accommodate the role of corners [19] and the finite-size effects [20, 21].

Within this landscape, models formulated in terms of loop and clusters [22] offer a particular fertile ground for illustrating the rich connections between integrable models and CFT [23, 24]. These models are close to the spirit of SLE, and offer the added advantage that their lattice formulation makes contact with cellular algebras [25] of the Temperley-Lieb type and their representation theory. The study of boundary critical behaviour in such models has led to remarkable successes, such as exact results for the crossing probabilities in critical percolation [26]. It has also been realised that boundary extensions of the Temperley-Lieb algebra [27–29] provide access to continuous families of conformally invariant boundary conditions [30], including in cases with two distinguished boundaries [31, 32].

Particular choices of the fugacity $\beta$ of the loops lead to non semi-simple representations of the Temperley-Lieb algebras, and to continuum limits that are logarithmic CFTs (see [33] for a review). This is true in particular for the most physically meaningful models, such as critical dense polymers, percolation, the Ising model and the 3-state Potts model, respectively corresponding to $\beta = 0, 1, \sqrt{2}, \sqrt{3}$. These models are ripe with technical subtleties and surprising results, and the confrontation of different approaches to extracting their critical properties is usually well justified.

The purpose of this paper is to present a detailed study of various boundary correlation functions in the dense loop model on the $m \times n$ square lattice. We express six different types of correlation functions in the form of determinants and pfaffians and compare their asymptotic expansions with the predictions of logarithmic CFT. The choice of correlation functions makes contact with several recent developments on the CFT side and confirms its remarkable predictive power. More interestingly, our exact results also provide some elements which are not yet fully understood from the CFT perspective. These include a $\ln(\ln n)$ contribution to the corner free energy, as well as partial results on the fusion rules of geometrically defined operators.

The outline of the paper is as follows. In Section 2, we first write down the conformal data of the CFT underlying the dense loop model. We introduce the lattice model on the $m \times n$ rectangle and review its description in terms of the Temperley-Lieb algebra. We define the six types of correlators and express them in terms of matrix elements in the XXZ spin-chain. In Section 3, we set $\beta = 0$ corresponding to the model of dense polymers and the XX spin-chain. Using free-fermion techniques, we write determinant or pfaffian expressions for each correlator, which we then use to extract the conformal weights of the boundary condition changing fields, as well as the logarithmic behaviour in one case. Some of the technical details are relegated to Sections 6 to 8. In Section 4, we return to generic values of $\beta$ and obtain conformal predictions for the weights of the six fields. The derivation uses the known finite-size corrections of the transfer matrix eigenvalues in the standard representations of the Temperley-Lieb algebra and its one- and two-blob generalisations. These are reviewed in Section 9. Finally in Section 5, we present an overview of the results and a discussion of an unanswered conundrum about the lattice interpretation for the fusion of some fields lying outside the Kac table.

## 2 Dense loop models and boundary correlators

### 2.1 Conformal data

The dense loop models are characterised by the fugacity $\beta$ of the contractible loops. For $\beta \in (-2, 2)$, we use the parameterisation

$$\beta = 2\cos\lambda = 2\cos\big(\pi(1-t)\big), \qquad t \in (0,1). \tag{2.1}$$

The central charge and conformal weights of the underlying conformal field theory are

$$c = 13 - 6(t + t^{-1}), \qquad \Delta_{r,s} = \frac{1-rs}{2} + \frac{r^2-1}{4t} + \frac{(s^2-1)t}{4}. \tag{2.2}$$

For the model of critical dense polymers, the loop fugacity is zero, corresponding to $t = \frac{1}{2}$ and

$$c = -2, \qquad \Delta_{r,s} = \frac{(2r-s)^2 - 1}{8}. \tag{2.3}$$

The two-point correlation functions defined in Section 2.6 are ratios of partition functions for two instances of the same lattice model which differ in the choice of the boundary condition. In Section 3, we evaluate the conformal weights of the boundary condition changing fields for the model of critical dense polymers, using two techniques. The first is to derive exact expressions for the two-point correlators on the upper half-plane $\mathbb{H}$ and compare with the expressions expected from conformal field theory. For $\varphi$ a primary field of conformal weight $\Delta$, the two-point function on $\mathbb{H}$ is

$$\langle\varphi(z_0)\varphi(z_1)\rangle = \frac{K}{|z_0 - z_1|^{2\Delta}}. \tag{2.4}$$

Similarly, consider a pair $(\varphi, \omega)$ of logarithmic fields with conformal weight $\Delta$ that $L_0$ mixes in a rank-two Jordan cell, with $\varphi$ the eigenstate and $\omega$ the Jordan partner. The two-point functions are

$$\langle\varphi(z_0)\varphi(z_1)\rangle = 0, \qquad \langle\varphi(z_0)\omega(z_1)\rangle = \frac{K_0}{|z_0 - z_1|^{2\Delta}},$$
$$\langle\omega(z_0)\omega(z_1)\rangle = \frac{K_1 - 2K_0 \ln|z_0 - z_1|}{|z_0 - z_1|^{2\Delta}}. \tag{2.5}$$

The second technique is to consider the correlation function on a semi-infinite strip of finite width $n$, with the two fields inserted in the corners. A partition function for the loop model on this geometry is typically divergent, whereas the ratio of two such partition function is finite and has the following $\frac{1}{n}$ expansion:

$$\lim_{m\to\infty} \ln Z/Z' = -n(f_s - f_s') - 2\ln n \sum_{\text{corners}} (\Delta - \Delta') + \dots . \tag{2.6}$$

Here, $f_s$ and $f_s'$ are the surface free energies corresponding to $Z$ and $Z'$, and $\Delta$ and $\Delta'$ are the weights of the corresponding fields in each corner. The $\ln n$ term is a corner contribution to the free energy and is a generalisation of the Cardy-Peschel formula [19] to the case where a field is inserted in the corner [34].

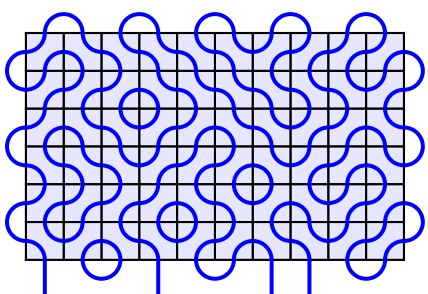

Figure 1: A loop configuration on the 6 × 10 rectangle.

## 2.2 The dense loop model

We consider the dense loop model on a square lattice of size $m \times n$ with $m$ and $n$ even integers. A configuration of the loop model is a choice of ◸ or ◿ for each of the $mn$ tiles. An example is given in Figure 1. The arcs drawn on the tiles combine to form loop segments that are space-filling. The boundary of the left, right and top segments of the rectangle are set to consist of arcs that connect nearest neighbour sites. We call those *simple arcs*. On the lower segment, we attach a collection of simple arcs and vertical segments called *defects*. In Section 2.6, we impose specific arrangements of the arcs and defects on this lower segment to define six types of boundary correlation functions. We borrow the terminology from the Coulomb gas formalism, where loops are contour curves for the height function, and use the terms *Dirichlet* and *Neumann* to designate boundary condition which respectively consist of collections of arcs and defects.

We refer to a collection of loop segments connecting two defects on the boundary as a *boundary loop*, and to a closed loop that does not touch the boundary as a *bulk loop*. Bulk loops are weighted by $\beta$ and their number is denoted $n_\beta$. The number of boundary loops is $\frac{d}{2}$ where $d$ is the number of defects attached to the boundary. This is true for all configurations. We weigh the boundary loops by the fugacity $\gamma = 1$. The weight of a loop configuration $\sigma$ and the partition function are then

$$w_\sigma = \beta^{n_\beta}, \qquad Z = \sum_\sigma w_\sigma. \tag{2.7}$$

## 2.3 The Temperley-Lieb algebra

**Definition.** The dense loop model is described by an algebra of connectivity diagrams, the Temperley-Lieb algebra [35] $\mathsf{TL}_n(\beta)$. Its representation theory is well understood [36–41]. This algebra is generated by the identity $I$ and elements $e_j$, $j = 1, \ldots, n-1$, which are depicted as

$$I = \boxed{\big| \big| \big| \cdots \big| \big|}_{1\ 2\ 3 \quad n}, \qquad e_j = \boxed{\big| \cdots \big| \cup \cap \big| \cdots \big|}_{1 \quad j \quad n}. \tag{2.8}$$

The $e_j$ satisfy the relations

$$(e_j)^2 = \beta e_j, \qquad e_j e_{j\pm 1} e_j = e_j, \qquad e_j e_k = e_k e_j \quad (|j-k| > 1). \tag{2.9}$$

The other connectivities $a$ in $\mathsf{TL}_n(\beta)$ are words in the $e_j$. The diagram for $a$ is obtained by stacking the diagrams of the corresponding $e_j$ and by straightening the loops. For example for

$n = 5$:

$$a = e_4 e_2 e_1 e_3 e_2 = \qquad = \qquad .\qquad (2.10)$$

The set of connectivities is made of all diagrams wherein the $2n$ nodes are connected by loop segments without intersections. The rule for the product $a_1 a_2$ of two connectivities is as follows: one stacks $a_2$ above $a_1$, straightens the loop segments and includes a multiplicative weight of $\beta$ for each closed loop. For example:

$$= \beta^2 \qquad .\qquad (2.11)$$

**Link modules and standard modules.** One module over $\mathsf{TL}_n(\beta)$ is the so-called *link module* $\mathsf{L}_n$. It is built on the vector space generated by the link states with $n$ nodes and an arbitrary number $d$ of defects, with $0 \leqslant d \leqslant n$ and $d \equiv n \bmod 2$. For instance, $\mathsf{L}_4$ is spanned by six link states:

$$\qquad .\qquad (2.12)$$

We define $\mathsf{L}_n$ to depend on a free parameter $\gamma$. To compute the action of an element $a \in \mathsf{TL}_n$ on $v \in \mathsf{L}_n$, one draws $v$ above $a$, straightens the loop segments and includes a factor of $\beta$ for each closed loop. If two defects annihilate, a multiplicative factor of $\gamma$ is included. For instance:

$$= \beta\gamma \qquad .\qquad (2.13)$$

If $\gamma = 0$, the number of defects is conserved and the link module decomposes as a direct sum of *standard modules*, which we denote $\mathsf{V}_{n,d}$. If $\gamma \neq 0$, the dependence on $\gamma$ can be removed by a change of basis in $\mathsf{L}_n$ which replaces $v$ by $v' = \gamma^{-d/2} v$, with $d$ the number of defects of $v$. In this new basis, the annihilation of two defects produces a weight 1. As a result, the study of the representation content of $\mathsf{L}_n$ reduces to two cases: $\gamma = 0$ and $\gamma = 1$.

**Bilinear forms.** We define the product $v \cdot v'$ as an application from $\mathsf{L}_n \times \mathsf{L}_n$ to $\mathbb{C}$. For two link states $v$ and $v'$ with respectively $d$ and $d'$ defects, $v \cdot v'$ is defined as

$$v \cdot v' = \beta^{n_\beta} \gamma^{(d+d')/2} \qquad (2.14)$$

where $n_\beta$ is the number of closed loops in the diagram where $v$ is flipped in a horizontal mirror and attached to $v'$. For instance, for $v = \qquad$ and $v' = \qquad$, we have

$$\longrightarrow \qquad v \cdot v' = \beta^2 \gamma^3. \qquad (2.15)$$

Depending on $\beta$ and $\gamma$, it may happen that $v \cdot v \leqslant 0$ for some $v \in \mathsf{L}_n$, so this bilinear form is not a scalar product in general. In the next section, we discuss generalisations of this product wherein the weight $\gamma$ depends on how the defects are connected.

We note that if $v$ and $v'$ have the same number $d$ of defects, then $\gamma^{-d}(v \cdot v')$ coincides with the usual Gram bilinear form for standard modules.

**XXZ modules.**    The generators $e_j$ are realised in the XXZ spin-chain by

$$\mathsf{X}_n(e_j) = \underbrace{\mathbb{I}_2 \otimes \cdots \otimes \mathbb{I}_2}_{j-1} \otimes \begin{pmatrix} 0 & 0 & 0 & 0 \\ 0 & q & 1 & 0 \\ 0 & 1 & q^{-1} & 0 \\ 0 & 0 & 0 & 0 \end{pmatrix} \otimes \underbrace{\mathbb{I}_2 \otimes \cdots \otimes \mathbb{I}_2}_{n-j-1}. \tag{2.16}$$

These matrices satisfy the relations (2.9) for $\beta = q + q^{-1}$, so $\mathsf{X}_n$ is a representation of $\mathsf{TL}_n(q + q^{-1})$. The corresponding spin-chain Hamiltonian is the XXZ Hamiltonian with the special boundary magnetic fields of Pasquier and Saleur [10]:

$$\begin{aligned} H &= -\sum_{j=1}^{n-1} \mathsf{X}_n(e_j) \\ &= -\frac{1}{2}\Big( \sum_{j=1}^{n-1} \sigma_j^x \sigma_{j+1}^x + \sigma_j^y \sigma_{j+1}^y - \frac{q+q^{-1}}{2}(\sigma_j^z \sigma_{j+1}^z - \mathbb{I}) \Big) - \frac{q-q^{-1}}{4}(\sigma_1^z - \sigma_n^z). \end{aligned} \tag{2.17}$$

## 2.4  Homomorphisms and generalised bilinear forms

One can construct a homomorphism between the modules $\mathsf{L}_n$ and $\mathsf{X}_n$. Each link state $v$ in $\mathsf{L}_n$ is mapped to an element of $(\mathbb{C}^2)^{\otimes n}$ which we denote by $|v\rangle$. This map is defined from the following local maps:

$$|\,\overset{\frown}{\phantom{a}}\,\rangle = q^{1/2}|\uparrow\downarrow\rangle + q^{-1/2}|\downarrow\uparrow\rangle \qquad |\,\rule[0pt]{0.4pt}{8pt}\,\rangle = |\uparrow\rangle + |\downarrow\rangle. \tag{2.18}$$

In general for $v \in \mathsf{L}_n$, $|v\rangle$ is obtained by applying multiplicatively (2.18) to each component (arcs and defects) of $v$. For example,

$$|\,\rule[0pt]{0.4pt}{8pt}\overset{\frown}{\phantom{a}}\,\rangle = q^{1/2}|\uparrow\uparrow\downarrow\rangle + q^{1/2}|\downarrow\uparrow\downarrow\rangle + q^{-1/2}|\uparrow\downarrow\uparrow\rangle + q^{-1/2}|\downarrow\downarrow\uparrow\rangle. \tag{2.19}$$

It is not hard to check that

$$\mathsf{X}_n(e_j)|v\rangle = |e_j v\rangle, \qquad j = 1, \ldots, n-1, \qquad v \in \mathsf{L}_n, \tag{2.20}$$

with $\beta = q + q^{-1}$ and $\gamma = q^{1/2} + q^{-1/2}$. Equivalently, this map is a homomorphism between $\mathsf{L}_n$ and $\mathsf{X}_n$.

To study bilinear forms realised in the XXZ representation, we define $\langle v| = |v\rangle^\dagger$ wherein $q$ is treated as a real parameter. We have the following local relations:

$$\langle\,\overset{\frown}{\phantom{a}}\,|\,\overset{\frown}{\phantom{a}}\,\rangle = q + q^{-1}, \qquad \langle\,\rule[0pt]{0.4pt}{8pt}\,|\,\rule[0pt]{0.4pt}{8pt}\,\rangle = 2, \qquad \langle\,\overset{\frown}{\phantom{a}}\,|\,\rule[0pt]{0.4pt}{8pt}\rule[0pt]{0.4pt}{8pt}\,\rangle = \langle\,\rule[0pt]{0.4pt}{8pt}\rule[0pt]{0.4pt}{8pt}\,|\,\overset{\frown}{\phantom{a}}\,\rangle = q^{1/2} + q^{-1/2}. \tag{2.21}$$

More generally, for two link states $v, v' \in \mathsf{L}_n$, $\langle v|v'\rangle$ evaluates to

$$\langle v|v'\rangle = (q+q^{-1})^{n_\beta}\, 2^{n_{vv'}} (q^{1/2} + q^{-1/2})^{n_v + n_{v'}} \tag{2.22}$$

where the numbers $n_v$, $n_{v'}$ and $n_{vv'}$ are read from in the diagram where $v$ is flipped and attached to $v'$: $n_v$ counts the pairs of defects of $v$ connected pairwise, $n_{v'}$ counts the pairs of defects of $v'$ connected pairwise, and $n_{vv'}$ counts the defects of $v$ connected to defects of $v'$. In the example (2.15), we have $n_v = 0$, $n_{v'} = 1$ and $n_{vv'} = 2$.

It is possible to consider more refined bilinear forms. Indeed, let us define

$$|\,\overset{s}{\rule[0pt]{0.4pt}{8pt}}\,\rangle = s^{\sigma^z}|\,\rule[0pt]{0.4pt}{8pt}\,\rangle = s|\uparrow\rangle + s^{-1}|\downarrow\rangle. \tag{2.23}$$

If $v$ has $d$ defects, we associate a parameter $s_i$, $i = 1, \ldots, d$, to each of its defects. Likewise we associate a parameter $t_j$, $j = 1, \ldots, d'$, to each of the $d'$ defects of $v'$. We then consider the

multi-variable product $\langle v|v'\rangle_{s,t}$, where $s$ and $t$ respectively denote the sets of parameters $s_i$ and $t_j$. The local relations for this generalised product are

$$\langle \underset{}{\cup} | \underset{}{\cup} \rangle_{s,t} = q + q^{-1}, \qquad \langle \overset{s_i \; s_j}{\underset{}{\sqcup\sqcup}} | \underset{}{\cup} \rangle_{s,t} = q^{1/2} s_i s_j^{-1} + q^{-1/2} s_i^{-1} s_j, \tag{2.24a}$$

$$\langle \overset{s_i \; t_j}{\underset{}{\sqcup\!\sqcup}} | \underset{}{\sqcup\sqcup} \rangle_{s,t} = s_i t_j + (s_i t_j)^{-1}, \qquad \langle \underset{}{\cup} | \overset{t_i \; t_j}{\underset{}{\sqcup\sqcup}} \rangle_{s,t} = q^{1/2} t_i t_j^{-1} + q^{-1/2} t_i^{-1} t_j. \tag{2.24b}$$

As a result, the refined product takes the form

$$\langle v|v'\rangle_{s,t} = (q + q^{-1})^{n_\beta} \prod_{\ell \in S} \gamma(\ell) \tag{2.25}$$

where $S$ is the set of loops connecting two defects and the weight $\gamma(\ell)$ is specific to $\ell$: It is selected as in (2.24) according to whether $\ell$ connects $v$ to $v'$, $v$ to itself or $v'$ to itself, and is a function of the parameters $s_i, t_j$ at the endpoints of $\ell$.

## 2.5 Transfer tangles and partition functions

The double-row transfer tangle $D(u)$ is an element of the algebra $\mathsf{TL}_n(\beta)$:

$$D(u) = \left( \begin{array}{|c|c|c|c|c|} \hline u & u & \cdots & \cdots & u \\ \hline u & u & \cdots & \cdots & u \\ \hline \end{array} \right), \qquad \boxed{u} = \frac{\sin(\lambda - u)}{\sin \lambda} \; \diagup \diagdown + \frac{\sin u}{\sin \lambda} \; \diagdown \diagup , \tag{2.26}$$

where $u$ is the spectral parameter and $\lambda = \pi(1 - t)$ is the crossing parameter, satisfying $\beta = 2 \cos \lambda$. The transfer matrix at different values of $u$ commute: $[D(u), D(v)] = 0$. The Hamiltonian

$$\mathcal{H} = -\sum_{j=1}^{n-1} e_j \tag{2.27}$$

is an element of this commuting family.

We consider a partition function on the rectangle, as in Figure 1, with the link state

$$v_0 = \underset{}{\cup} \; \underset{}{\cup} \; \cdots \; \underset{}{\cup} \tag{2.28}$$

applied at the top. Another link state $v \in \mathsf{L}_n$ is attached to the bottom of the rectangle; it will be specified in various ways below. The partition function (2.7) is computed using the XXZ representation and its realisation of the bilinear product defined in Section 2.4:

$$Z = \frac{\langle v | \mathsf{X}_n(D(\frac{\lambda}{2}))^{m/2} | v_0 \rangle}{(q^{1/2} + q^{-1/2})^{d/2}} \left( \frac{\sin \lambda}{\sin \lambda/2} \right)^{mn}, \tag{2.29}$$

where $d$ is the numbers of defects of $v$. The spectral parameter is set to $u = \lambda/2$, the isotropic value. The factor $(q^{1/2} + q^{-1/2})^{d/2}$ in the denominator ensures that each boundary loop is weighted by $\gamma = 1$.

In Section 2.6, we define correlation functions as ratios $Z/Z^0$ of partition functions that differ only by the choice of boundary condition, $v$ and $v'_0$, of the bottom segment:

$$\frac{Z}{Z^0} = \frac{1}{(q^{1/2} + q^{-1/2})^{(d-d'_0)/2}} \frac{\langle v | \mathsf{X}_n(D(\frac{\lambda}{2}))^{m/2} | v_0 \rangle}{\langle v'_0 | \mathsf{X}_n(D(\frac{\lambda}{2}))^{m/2} | v_0 \rangle}, \tag{2.30}$$

where $d'_0$ is the number of defects of $v'_0$. In Section 2.6, $Z$ is chosen to depend on two specified points $x$ and $y$ of the boundary in various ways, and $Z^0$ is the reference partition function. In

Sections 3 and 4, we study the behaviour of these ratios as functions of $x$ and $y$, with $Z^0$ kept fixed.

For $\beta \in \mathbb{R}^*$, the natural choice for $Z^0$ is to set single arcs everywhere on the lower segment, namely $v'_0 = v_0$. However for $\beta = 0$, the partition function with $v'_0 = v_0$ is zero: There are only bulk loops, all of which have zero fugacity. In this case, for $Z^0$, we set on the bottom segment of the rectangle a link state with two adjacent defects in positions $x$ and $x + 1$:

$$v'_0 = \underset{x}{\underbrace{\phantom{xxx}}} . \tag{2.31}$$

The nodes are labeled by the integers $1, \dots, n$ and $x$ is an odd integer in this range.

The corresponding partition function $Z^0$ is independent of the position $x$. To understand why, we consider the geometry of Figure 1 with the entire boundary decorated by simple arcs and count the configurations that contain exactly one loop. This number is non-zero and is a well-defined partition function for the model of critical dense polymers. One way to compute it is to select a simple arc from the lower segment, say the one tying the nodes $x$ and $x+1$. We impose that a loop has weight zero except if it passes through this special arc, in which case it has weight 1. This number can be computed using the link representation of $\mathsf{TL}_n(\beta = 0)$ by replacing the special arc by two defects: Setting $\gamma = 1$, the unique loop has weight 1 as required. The result is independent of which arc of the lower boundary is selected to be the special one, thus confirming that $Z^0$ is indeed independent of $x$. We immediately note that this is consistent with the conformal interpretation wherein the operator that inserts two adjacent defects has conformal weight $\Delta^0 = \Delta_{1,3} = 0$, see Section 4.2.

## 2.6  Six types of boundary correlators

In this section, we define correlation functions as ratios of partition functions. In the rectangular geometry, we denote these ratios by

$$C_{m,n}(x, y) = \frac{Z}{Z^0}. \tag{2.32}$$

Our calculations in Section 3 are performed by taking the limit $m \to \infty$, in which case the rectangle becomes a semi-infinite strip:

$$C_n(x, y) = \lim_{m \to \infty} C_{m,n}(x, y), \tag{2.33}$$

with $n$, $x$ and $y$ finite. For the various types of correlation functions discussed below, the partition functions $Z$ and $Z^0$ diverge as $m \to \infty$, but their ratio has a well-defined limit. We also define

$$C(x, y) = \lim_{n \to \infty} C_n(x, y), \tag{2.34}$$

with $x, y$ kept finite, in which case the semi-infinite strip becomes an infinite quadrant. We note that $C(x, y)$ is well-defined for the correlators of types a and b defined below, but not for c, d, e and f. In these cases, the difference in boundary condition beetween the partition functions in the numerator and the denominator is macroscopic. Indeed, the Dirichlet and Neumann surface free energies are different and the first term in (2.6) is non-zero, so $C_n(x, y)$ does not converge as $n \to \infty$.

We define six types of two-point correlation functions. For each, the corresponding boundary condition is illustrated in Figure 2.

(a) *Correlator between two isolated defects*: We assign the state

$$v^{\mathrm{a}} = \underset{x}{\underbrace{\phantom{xx}}} \underset{y}{\underbrace{\phantom{xx}}} \tag{2.35}$$

to the lower boundary, with $1 \leqslant x < y \leqslant n$, $x$ odd and $y$ even. We denote by $C^{\mathrm{a}}_{m,n}(x, y)$ the corresponding ratio of partition functions.

(b) *Correlator between two pairs of defects*: We assign the state

$$v^{\text{b}} = \underbrace{\phantom{xxxx}}_{x} \underbrace{\phantom{xxxx}}_{y} \tag{2.36}$$

to the lower boundary, with $1 \leqslant x < y \leqslant n$ and $x, y$ odd. We denote by $C^{\text{b}}_{m,n}(x,y)$ the corresponding ratio of partition functions.

(c) *Correlator for macroscopic collections of defects*: We assign the state

$$v^{\text{c}} = \underbrace{\phantom{xxxx}}_{x} \underbrace{\phantom{xxxx}}_{y} \tag{2.37}$$

to the lower boundary, with $1 \leqslant x < y \leqslant n$, $x$ odd and $y$ even. The number of defects is thus even. We denote by $C^{\text{c}}_{m,n}(x,y)$ the corresponding ratio of partition functions.

(d) *Correlator for cluster connectivities*: We assign the state

$$v^{\text{d}} = \underbrace{\big|\,\big|\,\big|\,\big| \cdots \big|\,\big|\,\big|\,\big|}_{0\ \ 1\ \ 2\ \cdots\qquad\quad n} \tag{2.38}$$

to the lower boundary. We label the midpoints between the defects by the integers $0, \dots, n$. The labels $0$ and $n$ then correspond to the left and right corners. We select two positions $x$ and $y$ with $y - x$ a positive even integer. In each loop configuration, there is a cluster $c_x$ attached to the boundary at $x$ whose contours are drawn by loop segments. Likewise there is cluster $c_y$ attached to the boundary at $y$. We write $c_x = c_y$ if $x$ and $y$ lie in the same cluster, and $c_x \neq c_y$ otherwise. We define the partition function restricted to configurations where $c_x = c_y$ and the corresponding correlation function as:

$$Z^{\text{d}} = \sum_{\sigma} w_{\sigma} \delta_{c_x, c_y}, \qquad C^{\text{d}}_{m,n}(x,y) = \frac{Z^{\text{d}}}{Z^0}. \tag{2.39}$$

(e) *Correlator for loop connectivities*: To the lower boundary, we assign the state $v^{\text{d}}$ and select two of the defects in positions $1 \leqslant x < y \leqslant n$ with $y - x$ odd. We denote by $\ell_x$ and $\ell_y$ the boundary loops attached to $x$ and $y$ and write $\ell_x = \ell_y$ if $x$ and $y$ are connected by a boundary loop. The corresponding restricted partition functions and correlation function are defined as

$$Z^{\text{e}} = \sum_{\sigma} w_{\sigma} \delta_{\ell_x, \ell_y}, \qquad C^{\text{e}}_{m,n}(x,y) = \frac{Z^{\text{e}}}{Z^0}. \tag{2.40}$$

We note that the constraint $\ell_x = \ell_y$ can be expressed in terms of the connectivities of clusters, in the vocabulary introduced for type d. Indeed, if $\ell_x = \ell_y$, then the two clusters adjacent to $\ell_x$ must be connected to the two clusters adjacent to $\ell_y$.

(f) *Correlator for segments connectivities* and *valence bond entanglement entropy*: To the lower boundary, we assign the state $v^{\text{d}}$. We choose $x, y$ in the range $0, \dots, n$ that are mid-points between nodes, as in case d. These split the lower edge of the rectangle in three segments: (1) between $0$ and $x$, (2) between $x$ and $y$, and (3) between $y$ and $n$. In Figure 2 (f), the segments 1 and 3 are drawn in black, and the segment 2 in purple. In a given configuration, there are $n_{ij}$ boundary loops connecting the segment $(i)$ to the segment $(j)$. In Figure 2 (f) for instance, we have $n_{12} = 1$, $n_{23} = 3$ and $n_{13} = 1$. We define the partition function and correlation function wherein loops connecting the segment 2 to the other segments are given a weight $\tau$:

$$Z^{\text{f}} = \sum_{\sigma} \beta^{n_\beta} \tau^{n_{12} + n_{23}}, \qquad C^{\text{f}}_{m,n}(x,y) = \frac{Z^{\text{f}}}{Z^0}. \tag{2.41}$$

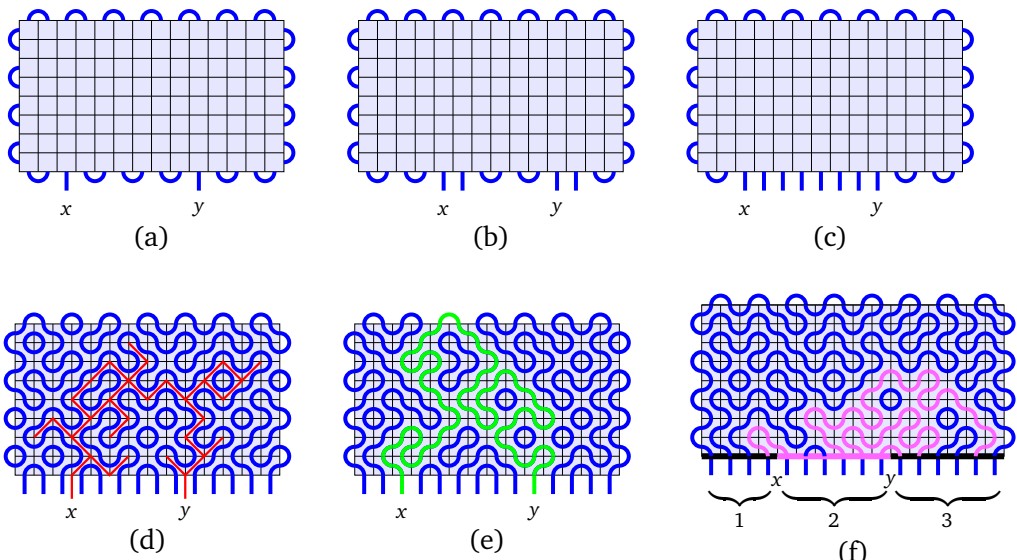

Figure 2: The boundary conditions for each of the six types of two-point correlators.

We note that for $\tau = 0$, this specialises to $Z^{\mathrm{f}}|_{\tau=0} = Z^{\mathrm{d}}$ for $y - x$ even, and to $Z^{\mathrm{f}}|_{\tau=0} = 0$ for $y - x$ odd. For $\tau = 1$, $Z^{\mathrm{f}}$ is independent of $x$ and $y$ and reproduces $Z^{\mathrm{c}}$ with $x = 1$ and $y = n$.

The valence bond entanglement entropy [42,43] is defined as the expectation value of $n_{12} + n_{23}$ and is obtained from a logarithmic derivative:

$$\langle n_{12} + n_{23} \rangle_{m,n} = \frac{\sum_{\sigma}(n_{12} + n_{23})\beta^{n_{\beta}}}{\sum_{\sigma}\beta^{n_{\beta}}} = \frac{\mathrm{d}(\ln C^{\mathrm{f}}_{m,n}(x,y))}{\mathrm{d}\tau}\Bigg|_{\tau=1}. \tag{2.42}$$

## 2.7 Spin-chain expressions for the partition functions

The correlation functions of type a, b and c are computed from (2.30) by respectively specialising $v$ to $v^{\mathrm{a}}$, $v^{\mathrm{b}}$ and $v^{\mathrm{c}}$. For the correlation functions of type d, e and f, we make use of the generalised bilinear forms discussed in Section 2.4 with specific choices of the $s_i$ parameters. The result is

$$\frac{Z^{\mathrm{d,e,f}}}{Z^0} = \frac{1}{(q^{1/2} + q^{-1/2})^{(n-d_0')/2}} \frac{\langle v^{\mathrm{d}}|\mathsf{X}_n(\boldsymbol{D}(\frac{\lambda}{2}))^{m/2}|v_0\rangle_s}{\langle v_0'|\mathsf{X}_n(\boldsymbol{D}(\frac{\lambda}{2}))^{m/2}|v_0\rangle}, \tag{2.43}$$

where the label $\boldsymbol{t}$ is removed because the matrix element does not involve any $t_j$.

Indeed, for correlators of type d, we split the lower edge of the lattice in three segments as in the first panel of Figure 3 and set

$$s_i = \begin{cases} q^{-1} & i = 1, \ldots, x, \\ \mathrm{i}q^{-1/2} & i = x+1, \ldots, y, \\ 1 & i = y+1, \ldots, n. \end{cases} \tag{2.44}$$

With this choice, a loop tying two defects from the same segment is given a weight $q^{1/2} + q^{-1/2}$, whereas a loop tying defects from two adjacent segments has weight zero. This constrains the cluster at $x$ to be connected to the cluster at $y$, as required. Finally, a loop tying the first segment to the third is also given the weight $q^{1/2} + q^{-1/2}$, so the bilinear form assigns the correct weight to each contributing configuration.

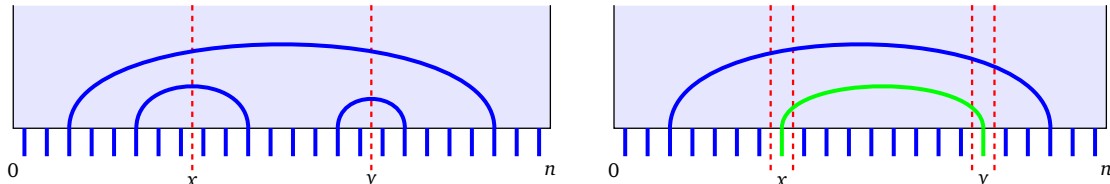

Figure 3: Division of the boundary in segments for correlators of type d and e.

For the correlators of type e, the lower edge of the lattice is divided in five segments, two of which consist of single nodes, as in the right panel of Figure 3. We specify the $s_i$ parameters to

$$s_i = \begin{cases} q^{-1} & i = 1, \ldots, x - 1, \\ iq^{-1/2} & i = x, \\ q^{-1} & i = x + 1, \ldots, y - 1, \\ iq^{-1/2} & i = y, \\ 1 & i = y + 1, \ldots, n. \end{cases} \tag{2.45}$$

With this choice, any loop that connects the defect at $y$ with a defect in the first, third or fifth segment is assigned a weight zero in $\langle v^d | v \rangle_s$. If $v$ is such that $\langle v^d | v \rangle_s \neq 0$, then a loop connects the nodes $x$ and $y$ and is assigned the weight $q^{1/2} + q^{-1/2}$. Other loops either tie a segment to itself, or the first and fifth segments, and the weight is $q^{1/2} + q^{-1/2}$ in each case, as required.

We compute $Z^f$ using the same ideas. We split the lower edge in three segments as for type d and set the parameters $s_i$ to

$$s_i = \begin{cases} q^{-1} & i = 1, \ldots, x, \\ e^{i\theta} & i = x + 1, \ldots, y, \\ 1 & i = y + 1, \ldots, n. \end{cases} \tag{2.46}$$

where $\theta$ is a free parameter. With this choice, the bilinear form assigns a weight $q^{1/2}e^{i\theta} + q^{-1/2}e^{-i\theta}$ to a loop connecting segment 2 to another segment, and the weight $q^{1/2} + q^{-1/2}$ to loops connecting the segments 1 and 3. Because $n_{12} + n_{23} + n_{13} = n/2$, we have

$$\langle v^d | v \rangle_s = (q^{1/2} + q^{-1/2})^{n/2} \left( \frac{q^{1/2}e^{i\theta} + q^{-1/2}e^{-i\theta}}{q^{1/2} + q^{-1/2}} \right)^{n_{12} + n_{23}} \tag{2.47}$$

and therefore (2.43) holds with

$$\tau = \frac{q^{1/2}e^{i\theta} + q^{-1/2}e^{-i\theta}}{q^{1/2} + q^{-1/2}}. \tag{2.48}$$

## 3 Exact results for critical dense polymers

In this section, we restrict our attention to the model of critical dense polymers wherein bulk loops have fugacity $\beta = 0$. In any given configuration, all loop segments are attached to the boundary. For this model, we give determinant and pfaffian formulas for the six types of two-point correlation functions defined in Section 2.6. We analyse the asymptotic behaviour and extract the conformal weights of the corresponding boundary condition changing fields.

### 3.1 The XX Hamiltonian

The spin-chain Hamiltonian corresponding to $\beta = 0$ is the XX chain:

$$H = -\sum_{j=1}^{n-1} X_n(e_j) \Big|_{q=i} = -\frac{1}{2} \left( \sum_{j=1}^{n-1} \sigma_j^x \sigma_{j+1}^x + \sigma_j^y \sigma_{j+1}^y \right) - \frac{i}{2} (\sigma_1^z - \sigma_n^z). \tag{3.1}$$

This Hamiltonian was for instance studied in [44, 45]. In terms of the fermions

$$c_j = (-1)^{j-1}\Big(\prod_{k=1}^{j-1}\sigma_k^z\Big)\sigma_j^-, \qquad c_j^\dagger = (-1)^{j-1}\Big(\prod_{k=1}^{j-1}\sigma_k^z\Big)\sigma_j^+, \tag{3.2}$$

the Hamiltonian takes the form

$$H = -\Big(\sum_{j=1}^{n-1} c_{j+1}^\dagger c_j + c_j^\dagger c_{j+1}\Big) - \mathrm{i}(c_1^\dagger c_1 - c_n^\dagger c_n). \tag{3.3}$$

We define the operators

$$a_j = \omega c_j + \omega^{-1} c_{j+1}, \qquad a_j^{\mathrm{t}} = \omega c_j^\dagger + \omega^{-1} c_{j+1}^\dagger, \qquad \omega = \mathrm{e}^{\mathrm{i}\pi/4}, \tag{3.4}$$

which satisfy

$$\{a_j, a_k\} = 0 = \{a_j^{\mathrm{t}}, a_k^{\mathrm{t}}\}, \qquad \{a_j, a_k^{\mathrm{t}}\} = \delta_{j,k-1} + \delta_{j,k+1}. \tag{3.5}$$

The Hamiltonian can be expressed in Jordan-normal form using the following operators:

$$\eta_k = \frac{1}{\kappa_k}\sum_{j=1}^{n-1}\sin(\tfrac{\pi k j}{n})a_j, \qquad \eta_k^{\mathrm{t}} = \frac{1}{\kappa_k}\sum_{j=1}^{n-1}\sin(\tfrac{\pi k j}{n})a_j^{\mathrm{t}}, \qquad \kappa_k = \sqrt{n\cos(\tfrac{\pi k}{n})}, \tag{3.6}$$

which satisfy the fermionic commutation relations

$$\{\eta_k, \eta_\ell\} = \{\eta_k^{\mathrm{t}}, \eta_\ell^{\mathrm{t}}\} = 0, \qquad \{\eta_k, \eta_\ell^{\mathrm{t}}\} = \delta_{k,\ell}. \tag{3.7}$$

We only consider the case where $n$ is even, for which $k$ takes values in $\{1,\ldots,\frac{n-2}{2}\}\cup\{\frac{n+2}{2},\ldots,n-1\}$. We complement this set of operators with

$$\phi = \sum_{j=1}^{n}\mathrm{i}^{-(j-1)}c_j, \qquad \phi^{\mathrm{t}} = \sum_{j=1}^{n}\mathrm{i}^{-(j-1)}c_j^\dagger, \tag{3.8}$$

and

$$\chi = -\frac{2}{n}\sum_{j=1}^{n}\mathrm{i}^{-(j-1)}\big\lfloor\tfrac{j}{2}-\tfrac{n}{4}\big\rfloor c_j, \qquad \chi^{\mathrm{t}} = -\frac{2}{n}\sum_{j=1}^{n}\mathrm{i}^{-(j-1)}\big\lfloor\tfrac{j}{2}-\tfrac{n}{4}\big\rfloor c_j^\dagger. \tag{3.9}$$

The operators $\phi$, $\phi^{\mathrm{t}}$, $\chi$ and $\chi^{\mathrm{t}}$ all anticommute with $\eta_k$ and $\eta_k^{\mathrm{t}}$. We also have

$$\{\phi,\chi\} = \{\phi^{\mathrm{t}},\chi^{\mathrm{t}}\} = \{\phi,\phi\} = \{\phi^{\mathrm{t}},\phi^{\mathrm{t}}\} = \{\chi,\chi\} = \{\chi^{\mathrm{t}},\chi^{\mathrm{t}}\} = 0, \quad \{\phi,\chi^{\mathrm{t}}\} = \{\phi^{\mathrm{t}},\chi\} = 1. \tag{3.10}$$

The Hamiltonian then takes the form

$$H = \phi^{\mathrm{t}}\phi + \sum_{\substack{k=1\\k\neq n/2}}^{n-1}\lambda_k\eta_k^{\mathrm{t}}\eta_k, \qquad \lambda_k = -2\cos(\tfrac{\pi k}{n}). \tag{3.11}$$

The first term is responsible for Jordan cells of rank 2. A full set of $2^n$ eigenstates and generalised eigenstates is obtained by acting on $|0\rangle = |\downarrow\downarrow\cdots\downarrow\rangle$ with the operators $\phi^{\mathrm{t}}, \chi^{\mathrm{t}}$ and $\eta_k^{\mathrm{t}}$, with $k$ in the set given above.

### 3.2 Correlators on the semi-infinite strip

To compute ratios of the form (2.30), one needs the eigenvalues and eigenvectors of $\mathsf{X}_n(\boldsymbol{D}(u))$. The eigenvalues are known [14, 46]. Because $\mathsf{X}_n(\boldsymbol{D}(u))$ commutes with $H = \mathsf{X}_n(\mathcal{H})$, the two operators share the same set of generalised eigenvectors.

In the limit $m \to \infty$, only the eigenspace of maximal eigenvalue $\Lambda_0$ contributes to (2.30). The generalised eigenspace for $\Lambda_0$ is four-dimensional. There are three proper eigenstates with magnetisation $-1, 0, 1$ which we denote $|w_{-1}\rangle$, $|w_0\rangle$ and $|w_1\rangle$, and one Jordan partner to $|w_0\rangle$ denoted $|\tilde{w}_0\rangle$. Explicitly, these states are given by

$$|w_{-1}\rangle = \eta_1^t \eta_2^t \dots \eta_{n/2-1}^t |0\rangle, \qquad |w_0\rangle = \phi^t |w_{-1}\rangle, \qquad |\tilde{w}_0\rangle = \chi^t |w_{-1}\rangle, \qquad |w_1\rangle = \phi^t \chi^t |w_{-1}\rangle, \tag{3.12}$$

and satisfy

$$\mathsf{X}_n(\boldsymbol{D}(u))|w_k\rangle = \Lambda_0 |w_k\rangle, \qquad \mathsf{X}_n(\boldsymbol{D}(u))|\tilde{w}_0\rangle = \Lambda_0 |\tilde{w}_0\rangle + f(u)|w_0\rangle, \tag{3.13}$$

where $f(u)$ is a trigonometric function of $u$, with $f(\frac{\pi}{4}) \neq 0$. Defining $\langle w| = |w\rangle^t$ yields the left generalised eigenstates. The ground-state eigenspace is four-dimensional. Restricted to this subspace, the identity operator is given by

$$\mathbb{1}\Big|_{\Lambda_0 \text{ eigenspace}} = |w_{-1}\rangle\langle w_{-1}| + |w_1\rangle\langle w_1| + |\tilde{w}_0\rangle\langle w_0| + |w_0\rangle\langle \tilde{w}_0|. \tag{3.14}$$

For the correlators of type a, b and c, we compute (2.30) in the limit $m \to \infty$ with $v = v^{\mathrm{a}}$, $v^{\mathrm{b}}$ and $v^{\mathrm{c}}$ respectively. For the correlators of type d, e and f, we instead compute (2.43) with the corresponding values of the $s_i$ parameters. In each case, the state $|v_0\rangle$ is

$$|v_0\rangle = a_1^t a_3^t a_5^t \cdots a_{n-1}^t |0\rangle. \tag{3.15}$$

It has zero magnetisation, so $\langle w_{-1}|v_0\rangle = \langle w_1|v_0\rangle = 0$. Because $\phi$ anticommutes with $a_j^t$, we also have $\langle w_0|v_0\rangle = 0$. Denoting by $\Lambda_1$ the second largest eigenvalue of $\mathsf{X}_n(\boldsymbol{D}(u))$ in the interval $0 < u < \frac{\pi}{2}$, we find

$$\left(\frac{\mathsf{X}_n(\boldsymbol{D}(\frac{\pi}{4}))}{\Lambda_0}\right)^{m/2} |v_0\rangle = |w_0\rangle\langle \tilde{w}_0|v_0\rangle + O\big((\Lambda_1/\Lambda_0)^{m/2}\big) \tag{3.16}$$

and therefore

$$C_n^{\mathrm{a,b,c}}(x,y) = \frac{1}{2^{(d-2)/4}} \frac{\langle v^{\mathrm{a,b,c}}|w_0\rangle}{\langle v_0'|w_0\rangle}, \qquad C_n^{\mathrm{d,e,f}}(x,y) = \frac{1}{2^{(n-2)/4}} \frac{\langle v^{\mathrm{d}}|w_0\rangle_s}{\langle v_0'|w_0\rangle}, \tag{3.17}$$

where $v_0'$ is given in (2.31). As expected, the results are independent of $v_0$, the state at the boundary that is infinitely far away.

### 3.3 Type a: two isolated defects

The two-point correlation function on the semi-infinite strip between two isolated defects is computed from (3.17) with $v^{\mathrm{a}}$ given in (2.35). The homormorphism defined in Section 2.4 gives

$$|v^{\mathrm{a}}\rangle = \left(\prod_{i=1}^{(x-1)/2} a_{2i-1}^t\right)(1 + c_x^\dagger)\left(\prod_{j=(x+1)/2}^{(y-2)/2} a_{2j}^t\right)(1 + c_y^\dagger)\left(\prod_{k=(y+2)/2}^{n/2} a_{2k-1}^t\right)|0\rangle. \tag{3.18}$$

For convenience, we choose $v'_0$ such that its leftmost defect sits in the same position $x$ as the leftmost defect of $v^{\mathrm{a}}$:

$$|v'_0\rangle = \left(\prod_{i=1}^{(x-1)/2} a^{\mathrm{t}}_{2i-1}\right)(1+c^\dagger_x)(1+c^\dagger_{x+1})\left(\prod_{k=(x+3)/2}^{n/2} a^{\mathrm{t}}_{2k-1}\right)|0\rangle. \tag{3.19}$$

Using the explicit form of $|w_0\rangle$, we can write down determinant expressions for $\langle v^{\mathrm{a}}|w_0\rangle$ and $\langle v'_0|w_0\rangle$. Because $\{a_j,\phi^{\mathrm{t}}\}=0$, the part involving $\phi^{\mathrm{t}}$ and the operators acting in positions $x$ and $y$ factors out, namely

$$\langle v^{\mathrm{a}}|w_0\rangle = \langle 0|\left((-1)^{\frac{x-1}{2}}c_x+(-1)^{\frac{y-2}{2}}c_y\right)\phi^{\mathrm{t}}|0\rangle \langle 0|\prod_{\substack{k=n/2 \\ \text{step}=-1}}^{(y+2)/2} a_{2k-1} \prod_{\substack{j=(y-2)/2 \\ \text{step}=-1}}^{(x+1)/2} a_{2j}$$

$$\times \prod_{\substack{i=(x-1)/2 \\ \text{step}=-1}}^{1} a_{2i-1}|w_{-1}\rangle. \tag{3.20}$$

Using the commutation relations for $\{c_j,\phi^{\mathrm{t}}\}$, we find that the first matrix element equals $\sqrt{2}\,\omega^{-1}$. The second matrix element is rewritten using Wick's theorem: For $\phi_\ell$ and $\varphi^{\mathrm{t}}_k$ fermionic annihilation and creation operators, the following equality holds:

$$\langle 0|\phi_L\ldots\phi_2\phi_1\varphi^{\mathrm{t}}_1\varphi^{\mathrm{t}}_2\ldots\varphi^{\mathrm{t}}_L|0\rangle = \det_{k,\ell=1}^{L}\{\phi_\ell,\varphi^{\mathrm{t}}_k\}. \tag{3.21}$$

We apply this for $L=\frac{n-2}{2}$, with

$$\phi_\ell = \begin{cases} a_{2\ell-1} & \ell=1,\ldots,\frac{x-1}{2}, \\ a_{2\ell} & \ell=\frac{x+1}{2},\ldots,\frac{y-1}{2}, \\ a_{2\ell+1} & \ell=\frac{y+1}{2},\ldots,\frac{n-2}{2}, \end{cases} \qquad \varphi^{\mathrm{t}}_k = \eta^{\mathrm{t}}_k. \tag{3.22}$$

These satisfy the commutation rules $\{a_\ell,\eta^{\mathrm{t}}_k\}=2\cos(\frac{\pi k}{n})\sin(\frac{\pi k\ell}{n})$. The result is the determinant of a matrix $M^{x,y}$ of size $\frac{n-2}{2}$:

$$\langle v^{\mathrm{a}}|w_0\rangle = \sqrt{2}\,\omega^{-1}\prod_{k=1}^{(n-2)/2}\frac{2\cos(\frac{\pi k}{n})}{\kappa_k}\det M^{x,y}, \tag{3.23}$$

where

$$M^{x,y}_{k,\ell} = \begin{cases} \sin\left(\frac{2\pi k}{n}(\ell-\frac{1}{2})\right) & \ell=1,\ldots,\frac{x-1}{2}, \\ \sin\left(\frac{2\pi k}{n}\ell\right) & \ell=\frac{x+1}{2},\ldots,\frac{y-2}{2}, \\ \sin\left(\frac{2\pi k}{n}(\ell+\frac{1}{2})\right) & \ell=\frac{y}{2},\ldots,\frac{n-2}{2}. \end{cases} \tag{3.24}$$

The factors $2\cos(\frac{\pi k}{n})$, which do not depend on the index $\ell$, were factorised from the determinant. The expression for $\langle v'_0|w_0\rangle$ is obtained from (3.23) and (3.24) by substituting $y=x+1$. In this case, we abbreviate $M^{x,x+1}=M^x$. Its determinant evaluates to

$$\det M^x = (-1)^{(n-2)(n-4)/8}\frac{n^{\frac{n-4}{4}}}{2^{\frac{n-3}{2}}}. \tag{3.25}$$

Using

$$\prod_{k=1}^{(n-2)/2} 2\cos(\tfrac{\pi k}{n}) = \sqrt{n/2}, \tag{3.26}$$

we find

$$\langle v_0'|w_0\rangle = \frac{\omega^{-1}(-1)^{(n-2)(n-4)/8}}{\prod_{k=1}^{(n-2)/2}\kappa_k}\frac{n^{(n-2)/4}}{2^{(n-3)/2}}. \tag{3.27}$$

It is independent of $x$, as expected from the discussion at the end of Section 2.5.

**The case $(x,y) = (1,n)$.** Below, we present a closed-form expression for the determinant of $M^{x,y}$ for arbitrary $x,y$, but let us first present a limiting case: $x=1$ and $y=n$. In this case,

$$\det M^{1,n} = (-1)^{(n-2)(n-4)/8}\frac{n^{\frac{n-2}{4}}}{2^{\frac{n-2}{2}}}. \tag{3.28}$$

This allows us to write an exact expression for $\ln C_n^{\mathrm{a}}(1,n)$:

$$\ln C_n^{\mathrm{a}}(1,n) = \tfrac{1}{2}\ln n - \tfrac{1}{2}\ln 2. \tag{3.29}$$

This is consistent with (2.6) with boundary condition changing fields of weight

$$\Delta^{\mathrm{a}} = -\frac{1}{8} \qquad \Delta^0 = 0 \tag{3.30}$$

in each of the two corners.

**The general $(x,y)$ case.** The above corner free energy analysis allows one to determine $\Delta^{\mathrm{a}} - \Delta^0$, but not each conformal weight individually. To confirm the identification (3.30), we pursue the computation of $C_n^{\mathrm{a}}(x,y)$ for arbitrary values $x,y$. We have

$$C_n^{\mathrm{a}}(x,y) = \frac{\det M^{x,y}}{\det M^x} = \det N, \qquad N = (M^x)^{-1}M^{x,y}. \tag{3.31}$$

The inverse of $M^x$ is given by

$$(M^x)_{k,\ell}^{-1} = \frac{4}{n}\begin{cases} \sin\left(\frac{\pi\ell}{n}(2k-1)\right)+(-1)^{k-\frac{x-1}{2}}\sin\left(\frac{\pi\ell x}{n}\right) & k \leqslant \frac{x-1}{2}, \\ \sin\left(\frac{\pi\ell}{n}(2k+1)\right)+(-1)^{k-\frac{x+1}{2}}\sin\left(\frac{\pi\ell x}{n}\right) & k \geqslant \frac{x+1}{2}. \end{cases} \tag{3.32}$$

The columns of $M^{x,y}$ and $M^x$ labeled by $\ell = \frac{x+1}{2},\ldots,\frac{y-2}{2}$ are different, but the other ones are identical. As a consequence, $N_{k,\ell} = \delta_{k,\ell}$ for $\ell = 1,\ldots,\frac{x-1}{2}$ and $\ell = \frac{y}{2},\ldots,\frac{n-2}{2}$. The ratio of determinants in (3.31) then reduces to the determinant of a matrix of size $\frac{y-x-1}{2}$:

$$C_n^{\mathrm{a}}(x,y) = \det_{k,\ell=\frac{x+1}{2}}^{\frac{y-2}{2}} N_{k,\ell}. \tag{3.33}$$

For $\frac{x+1}{2} \leqslant k,\ell \leqslant \frac{y-2}{2}$, the matrix elements $N_{k,\ell}$ are obtained by a direct computation:

$$N_{k,\ell} = \frac{(-1)^{k+\ell}\sin(\frac{2\pi\ell}{n})\sin(\frac{\pi}{n}(k+\frac{x+1}{2}))\sin(\frac{\pi}{n}(k-\frac{x-1}{2}))}{n\sin(\frac{\pi}{n}(k+\ell+\frac{1}{2}))\sin(\frac{\pi}{n}(k-\ell+\frac{1}{2}))\sin(\frac{\pi}{n}(\ell+\frac{x}{2}))\sin(\frac{\pi}{n}(\ell-\frac{x}{2}))}. \tag{3.34}$$

They are independent of $y$. This yields

$$C_n^{\mathrm{a}}(x,y) = \prod_{k=\frac{x+1}{2}}^{\frac{y-2}{2}}\frac{\sin(\frac{2\pi k}{n})\sin(\frac{\pi}{n}(k+\frac{x+1}{2}))\sin(\frac{\pi}{n}(k-\frac{x-1}{2}))}{n\sin(\frac{\pi}{n}(k+\frac{x}{2}))\sin(\frac{\pi}{n}(k-\frac{x}{2}))}$$
$$\times \det_{k,\ell=\frac{x+1}{2}}^{\frac{y-2}{2}}\frac{1}{\sin(\frac{\pi}{n}(k+\ell+\frac{1}{2}))\sin(\frac{\pi}{n}(k-\ell+\frac{1}{2}))}. \tag{3.35}$$

The remaining determinant is evaluated using Cauchy's identity

$$
\det_{k,\ell=a}^{b} \frac{1}{w_k - z_\ell} = \frac{\prod_{a \leqslant k < \ell \leqslant b}(w_k - w_\ell)(z_\ell - z_k)}{\prod_{k,\ell=a}^{b}(w_k - z_\ell)}
\tag{3.36}
$$

with $w_k = -\frac{1}{2}\cos(\frac{2\pi}{n}(k + \frac{1}{2}))$ and $z_\ell = -\frac{1}{2}\cos(\frac{2\pi\ell}{n})$. The result is a closed-form expression for $C_n^{\mathrm{a}}(x,y)$:

$$
C_n^{\mathrm{a}}(x,y) = \prod_{k=\frac{x+1}{2}}^{\frac{y-2}{2}} \frac{\sin(\frac{2\pi k}{n})\sin(\frac{\pi}{n}(k + \frac{x+1}{2}))\sin(\frac{\pi}{n}(k - \frac{x-1}{2}))}{n\sin(\frac{\pi}{n}(k + \frac{x}{2}))\sin(\frac{\pi}{n}(k - \frac{x}{2}))}
$$
$$
\times \frac{\prod_{\frac{x+1}{2} \leqslant k < \ell \leqslant \frac{y-2}{2}} \sin(\frac{\pi}{n}(k+\ell))\sin(\frac{\pi}{n}(k+\ell+1))\sin(\frac{\pi}{n}(k-\ell))\sin(\frac{\pi}{n}(\ell-k))}{\prod_{k,\ell=\frac{x+1}{2}}^{\frac{j-2}{2}} \sin(\frac{\pi}{n}(k+\ell+\frac{1}{2}))\sin(\frac{\pi}{n}(k-\ell+\frac{1}{2}))}.
\tag{3.37}
$$

The $n \to \infty$ limit is well-defined and non-zero. It is obtained by replacing each sine function by its argument:

$$
C^{\mathrm{a}}(x,y) = \left(\frac{2}{\pi}\right)^{\frac{y-x-1}{2}} \times
$$
$$
\prod_{k=\frac{x+1}{2}}^{\frac{y-2}{2}} \frac{k(k + \frac{x+1}{2})(k - \frac{x-1}{2})}{(k + \frac{x}{2})(k - \frac{x}{2})} \frac{\prod_{\frac{x+1}{2} \leqslant k < \ell \leqslant \frac{y-2}{2}}(k+\ell)(k+\ell+1)(k-\ell)(\ell-k)}{\prod_{k,\ell=\frac{x+1}{2}}^{\frac{y-2}{2}}(k+\ell+\frac{1}{2})(k-\ell+\frac{1}{2})}.
\tag{3.38}
$$

This expression can be written in terms of the Barnes $G$-functions:

$$
G(z+1) = \Gamma(z)G(z), \qquad \Gamma(z+1) = z\,\Gamma(z).
\tag{3.39}
$$

After simplification, the result reads

$$
\begin{aligned}
C^{\mathrm{a}}(x,y) = {}& \frac{G(y)G(y+1)}{G^2(y+\frac{1}{2})} \frac{G(\frac{y}{2}+\frac{1}{4})G^2(\frac{y}{2}+\frac{3}{4})G(\frac{y}{2}+\frac{5}{4})}{G(\frac{y}{2})G(\frac{y}{2}+\frac{1}{2})G(\frac{y}{2}+1)G(\frac{y}{2}+\frac{3}{2})} \frac{G(\frac{y}{2}-\frac{x}{2}+\frac{1}{2})G(\frac{y}{2}-\frac{x}{2}+\frac{3}{2})}{G^2(\frac{y}{2}-\frac{x}{2}+1)} \\
& \times \frac{G(x+\frac{3}{2})G(x+1)}{G(x+\frac{1}{2})G(x+2)} \frac{G(\frac{x}{2}+\frac{1}{2})G(\frac{x}{2}+1)G(\frac{x}{2}+\frac{3}{2})G(\frac{x}{2}+2)}{G(\frac{x}{2}+\frac{3}{4})G^2(\frac{x}{2}+\frac{5}{4})G(\frac{x}{2}+\frac{7}{4})} \\
& \times \frac{G(\frac{y}{2}+\frac{x}{2})G(\frac{y}{2}+\frac{x}{2}+1)}{G^2(\frac{y}{2}+\frac{x}{2}+\frac{1}{2})} G^2(3/2).
\end{aligned}
\tag{3.40}
$$

We consider the behaviour of $\ln C^{\mathrm{a}}(x,y)$ for $x, y, y-x \gg 1$. The large-$z$ asymptotic expansion of the logarithm of the Barnes $G$-function is

$$
\ln G(1+z) = \left(\frac{z^2}{2} - \frac{1}{12}\right)\ln z - \frac{3z^2}{4} + \frac{z}{2}\ln 2\pi + \frac{1}{12} - \ln A + O(z^{-2}),
\tag{3.41}
$$

where $A$ is the Glaisher-Kinkelin constant. We thus set $x = rx'$, $y = ry'$, take $r \to \infty$ with $x', y'$ finite, and find:

$$
\ln C^{\mathrm{a}}(x,y) = \frac{1}{4}\ln(y+x) + \frac{1}{4}\ln(y-x) - \frac{1}{8}\ln x - \frac{1}{8}\ln y - \frac{1}{2}\ln 2 + \ln G^2(\tfrac{3}{2}) + O(r^{-1}).
\tag{3.42}
$$

The power-law behaviour of $C^{\mathrm{a}}(x,y)$ in the geometry of the quadrant is therefore

$$
C^{\mathrm{a}}(x,y) = K \frac{(y+x)^{\frac{1}{4}}(y-x)^{\frac{1}{4}}}{x^{\frac{1}{8}}y^{\frac{1}{8}}}, \qquad K = \frac{G^2(\frac{3}{2})}{\sqrt{2}}.
\tag{3.43}
$$

To recover the result on the upper half-plane, we consider the regime $x, y \gg y - x$, in which case (3.43) becomes

$$C^{\mathrm{a}}(x, y) \xrightarrow{x, y \gg y - x} 2^{1/4} K(y - x)^{1/4}. \tag{3.44}$$

This is consistent with (2.4) with

$$\Delta^{\mathrm{a}} = -\frac{1}{8}. \tag{3.45}$$

## 3.4 Type b: two pairs of defects

The two-point correlation function on the semi-infinite strip between two pairs of defects is computed from (3.17) with $v^{\mathrm{b}}$ given in (2.36). We immediately note that the loop configurations contributing to $Z^{\mathrm{b}}$ can be split in two families according to the way that the points $x$, $x + 1$, $y$ and $y + 1$ are connected:

$$\text{and} \tag{3.46}$$

We refer to the refined partition functions as $Z^{\mathrm{b}}_{\frown}$ and $Z^{\mathrm{b}}_{\frown\frown}$, with $Z^{\mathrm{b}} = Z^{\mathrm{b}}_{\frown} + Z^{\mathrm{b}}_{\frown\frown}$. We can compute $Z_{\frown}$ and $Z_{\frown\frown}$ separately by using the generalised bilinear forms discussed in Section 2.4:

$$\frac{\langle v^{\mathrm{b}} | s_1^{\sigma_x^z} s_2^{\sigma_{x+1}^z} s_3^{\sigma_y^z} s_4^{\sigma_{y+1}^z} | w_0 \rangle}{\langle v_0' | w_0 \rangle} = \frac{\gamma_{12}\gamma_{34}}{\sqrt{2}} \frac{Z^{\mathrm{b}}_{\frown}}{Z^0} + \frac{\gamma_{14}\gamma_{23}}{\sqrt{2}} \frac{Z^{\mathrm{b}}_{\frown\frown}}{Z^0}, \qquad \gamma_{ab} = \omega \frac{s_a}{s_b} + \omega^{-1} \frac{s_b}{s_a}. \tag{3.47}$$

Alternatively, there is a simple argument to show that

$$Z^{\mathrm{b}}_{\frown} = Z^0 \tag{3.48}$$

for $\beta = 0$. Indeed, there is a simple bijective map between configurations contributing to $Z^{\mathrm{b}}_{\frown}$ and to $Z^0$: In each loop configuration contributing to $Z^{\mathrm{b}}_{\frown}$, one ties together the defects in $y$ and $y + 1$ with a simple arc. It is easy to see that this map preserves the weight of each configuration, and therefore $Z^{\mathrm{b}}_{\frown} = Z^0$.

**The general $x, y$ case.** We thus proceed to compute $\langle v^{\mathrm{b}} | w_0 \rangle$ without the added $s_i$ factors in (3.47), knowing in advance that $C^{\mathrm{b}}(x, y) = 1 + C^{\mathrm{b}}_{\frown\frown}(x, y)$ with $C^{\mathrm{b}}_{\frown\frown}(x, y) = Z^{\mathrm{b}}_{\frown\frown}/Z^0$. From Section 2.4, we have

$$|v^{\mathrm{b}}\rangle = \left( \prod_{i=1}^{(x-1)/2} a_{2i-1}^{\mathrm{t}} \right)(1 + c_x^{\dagger})(1 + c_{x+1}^{\dagger})\left( \prod_{j=(x+1)/2}^{(y-3)/2} a_{2j+1}^{\mathrm{t}} \right)(1 + c_y^{\dagger})(1 + c_{y+1}^{\dagger})$$

$$\times \left( \prod_{k=(y+1)/2}^{(n-2)/2} a_{2k+1}^{\mathrm{t}} \right)|0\rangle, \tag{3.49}$$

where $x$ and $y$ are odd. This yields

$$\langle v^{\mathrm{b}} | w_0 \rangle = \sum_{\{\ell_1, \ell_2\} \subset S} \sigma_{\ell_1, \ell_2} \langle 0 | a_{n-1} \cdots a_{y+2} a_{y-2} \cdots a_{x+2} a_{x-2} \cdots a_3 a_1 c_{\ell_2} c_{\ell_1} \phi^{\mathrm{t}} \eta_1^{\mathrm{t}} \eta_2^{\mathrm{t}} \cdots \eta_{(n-2)/2}^{\mathrm{t}} |0\rangle, \tag{3.50}$$

where $\ell_1 < \ell_2$ in the sum and

$$\sigma_{\ell_1, \ell_2} = \begin{cases} 1 & \ell_2 - \ell_1 = 1, \\ (-1)^{\frac{y-x-2}{2}} & \text{otherwise,} \end{cases} \qquad S = \{x, x+1, y, y+1\}. \tag{3.51}$$

Using Wick's theorem, each term in (3.50) is expressed as a determinant. We find

$$C_n^{\mathrm{b}}(x,y) = \frac{(-1)^{(y-1)/2}\omega^{-1}}{\sqrt{2}} \sum_{\{\ell_1,\ell_2\} \subset S} \sigma_{\ell_1,\ell_2} \frac{\det P^{x,y}}{\det \hat{M}^x}. \tag{3.52}$$

The explicit forms of the matrices are

$$\hat{M}^x = \begin{pmatrix} g_{1,1}+g_{1,2} & h_{1,1} & h_{1,3} & \cdots & h_{1,x-4} & h_{1,x-2} & h_{1,x+2} & h_{1,x+4} & \cdots & h_{1,n-1} \\ g_{2,1}+g_{2,2} & h_{2,1} & h_{2,3} & \cdots & h_{2,x-4} & h_{2,x-2} & h_{2,x+2} & h_{2,x+4} & \cdots & h_{2,n-1} \\ \vdots & \vdots & \vdots & & \vdots & \vdots & \vdots & \vdots & & \vdots \\ g_{\frac{n-2}{2},1}+g_{\frac{n-2}{2},2} & h_{\frac{n-2}{2},1} & h_{\frac{n-2}{2},3} & \cdots & h_{\frac{n-2}{2},x-4} & h_{\frac{n-2}{2},x-2} & h_{\frac{n-2}{2},x+2} & h_{\frac{n-2}{2},x+4} & \cdots & h_{\frac{n-2}{2},n-1} \\ \sqrt{2} & 0 & 0 & \cdots & 0 & 0 & 0 & 0 & \cdots & 0 \end{pmatrix}, \tag{3.53a}$$

$$P^{x,y} = \begin{pmatrix} g_{1,m_1} & h_{1,1} & h_{1,3} & \cdots & h_{1,x-2} & h_{1,x+2} & \cdots & h_{1,y-2} & g_{1,m_2} & h_{1,y+2} & \cdots & h_{1,n-1} \\ g_{2,m_1} & h_{2,1} & h_{2,3} & \cdots & h_{2,x-2} & h_{2,x+2} & \cdots & h_{2,y-2} & g_{2,m_2} & h_{2,y+2} & \cdots & h_{2,n-1} \\ \vdots & \vdots & \vdots & & \vdots & \vdots & & \vdots & \vdots & \vdots & & \vdots \\ g_{\frac{n-2}{2},m_1} & h_{\frac{n-2}{2},1} & h_{\frac{n-2}{2},3} & \cdots & h_{\frac{n-2}{2},x-2} & h_{\frac{n-2}{2},x+2} & \cdots & h_{\frac{n-2}{2},y-2} & g_{\frac{n-2}{2},m_2} & h_{\frac{n-2}{2},y+2} & \cdots & h_{\frac{n-2}{2},n-1} \\ \mathrm{i}^{-\ell_1} & 0 & 0 & \cdots & 0 & 0 & \cdots & 0 & \mathrm{i}^{-\ell_2} & 0 & \cdots & 0 \end{pmatrix}, \tag{3.53b}$$

with

$$g_{k,\ell} = \frac{\kappa_k\{\eta_k^{\mathrm{t}}, c_\ell\}}{2\cos(\frac{\pi k}{n})} = \frac{\omega^{-1}\sin(\frac{\pi k(\ell-1)}{n}) + \omega\sin(\frac{\pi k\ell}{n})}{2\cos(\frac{\pi k}{n})}, \qquad h_{k,\ell} = \frac{\kappa_k\{\eta_k^{\mathrm{t}}, a_\ell\}}{2\cos(\frac{\pi k}{n})} = \sin(\tfrac{\pi k\ell}{n}). \tag{3.54}$$

In particular, the powers of i in the last row of $P^{x,y}$ come from the commutators of $\phi^{\mathrm{t}}$ and $c_{\ell_1}, c_{\ell_2}$.

The matrices $P^{x,y}$ and $\hat{M}^x$ are identical except in the columns 1 and $\frac{y+1}{2}$. As a result, $\det\big((\hat{M}^x)^{-1}P^{x,y}\big)$ simplifies to the determinant of a $2 \times 2$ matrix. The upper-right $\frac{n-2}{2} \times \frac{n-2}{2}$ minor of $\hat{M}^x$ is just the matrix $M^x$ defined in Section 3.3. The inverse of $\hat{M}^x$ is thus easily written down in terms of (3.32). We find:

$$\det\big((\hat{M}^x)^{-1}P^{x,y}\big) = \frac{1}{\sqrt{2}} \det \begin{pmatrix} \mathrm{i}^{-\ell_1} & \mathrm{i}^{-\ell_2} \\ f_{(y-1)/2,\ell_1} & f_{(y-1)/2,\ell_2} \end{pmatrix}, \tag{3.55}$$

where

$$f_{k,\ell} = \sum_{j=1}^{(n-2)/2} (\hat{M}^x)^{-1}_{k,j} g_{j,\ell} = \omega^{-1}\tilde{f}_{k,\ell-1} + \omega\tilde{f}_{k,\ell}, \tag{3.56a}$$

$$\tilde{f}_{k,\ell} = \frac{2}{n} \sum_{j=1}^{(n-2)/2} \Big( \sin\big(\tfrac{\pi j}{n}(2k+1)\big) + (-1)^{k-(x+1)/2}\sin(\tfrac{\pi jx}{n}) \Big) \frac{\sin(\frac{\pi j\ell}{n})}{\cos(\frac{\pi j}{n})}. \tag{3.56b}$$

The resulting expression for $C_n^{\mathrm{b}}(x,y)$ is thus considerably different from the one for $C_n^{\mathrm{a}}(x,y)$ found in Section 3.3. We evaluate the determinant in (3.55), explicitly write down each term of the sum (3.52) and find

$$C_n^{\mathrm{b}}(x,y) = \frac{1}{2}\Big((-1)^{\frac{y-x-2}{2}}(\tilde{f}_{x-1} + 2\tilde{f}_x + (1+2\mathrm{i})\tilde{f}_{x+1}) + (1-2\mathrm{i})\tilde{f}_{y-1} + 2\tilde{f}_y + \tilde{f}_{y+1}\Big), \tag{3.57}$$

where we abbreviate $\tilde{f}_{(y-1)/2,\ell} = \tilde{f}_\ell$. The function $\tilde{f}_{k,\ell}$ is simplified in Section 6. For $\ell$ even, $\tilde{f}_{k,\ell}$ admits a simple form given in (6.2), from which we read:

$$\tilde{f}_{x-1} = \tilde{f}_{y+1} = 0, \qquad \tilde{f}_{x+1} = (-1)^{\frac{y-x-2}{2}}, \qquad \tilde{f}_{y-1} = 1. \tag{3.58}$$

We thus have

$$C_n^{\mathrm{b}}(x,y) = 1 + (-1)^{\frac{y-x-2}{2}}\tilde{f}_x + \tilde{f}_y. \tag{3.59}$$

A simplified expression for $\tilde{f}_{k,\ell}$ with $\ell$ odd is (6.4). We now take the limit $n \to \infty$. To compute $C^{\mathrm{b}}(x,y)$, we use (6.6) and find after simplification:

$$\lim_{n\to\infty} \tilde{f}_x = \frac{(-1)^{\frac{y-x-2}{2}}}{\pi}\left(\sum_{k=0}^{(y+x-2)/2} \frac{1}{k+\frac{1}{2}} + \sum_{k=0}^{(y-x-2)/2} \frac{1}{k+\frac{1}{2}} - \sum_{k=0}^{x-1}\frac{1}{k+\frac{1}{2}}\right), \tag{3.61}$$

$$\lim_{n\to\infty} \tilde{f}_y = \frac{1}{\pi}\left(\sum_{k=0}^{(y+x-2)/2} \frac{1}{k+\frac{1}{2}} + \sum_{k=0}^{(y-x-2)/2} \frac{1}{k+\frac{1}{2}} - \sum_{k=0}^{y-1}\frac{1}{k+\frac{1}{2}}\right), \tag{3.62}$$

and finally

$$C^{\mathrm{b}}(x,y) = 1 + \frac{1}{\pi}\left(2\sum_{k=0}^{(y+x-2)/2} \frac{1}{k+\frac{1}{2}} + 2\sum_{k=0}^{(y-x-2)/2} \frac{1}{k+\frac{1}{2}} - \sum_{k=0}^{y-1}\frac{1}{k+\frac{1}{2}} - \sum_{k=0}^{x-1}\frac{1}{k+\frac{1}{2}}\right). \tag{3.63}$$

We study the behaviour in the regime $x, y, y - x \gg 1$ by setting $x = r x'$, $y = r y'$, expanding in powers of $1/r$ and using

$$\sum_{k=0}^{t} \frac{1}{k+\frac{1}{2}} = \ln t + 2\ln 2 + \gamma + O(t^{-1}), \tag{3.64}$$

where $\gamma$ is the Euler-Mascheroni constant. This yields

$$C^{\mathrm{b}}(x,y) = 1 + \frac{1}{\pi}\left(2\ln(y+x) + 2\ln(y-x) - \ln y - \ln x\right) + \frac{2\gamma}{\pi} + O(r^{-1}). \tag{3.65}$$

The result on the upper half-plane is obtained by taking $x, y \gg y - x$:

$$C^{\mathrm{b}}(x,y) \xrightarrow{x,y\gg y-x} 1 + \frac{2}{\pi}\ln(y-x) + \frac{2}{\pi}(\gamma + \ln 2) \tag{3.66}$$

from which we read off

$$C^{\mathrm{b}}_{\circ\circ}(x,y) \xrightarrow{x,y\gg y-x} \frac{2}{\pi}\ln(y-x) + \frac{2}{\pi}(\gamma + \ln 2). \tag{3.67}$$

This logarithmic behaviour is consistent with the rightmost equation in (2.5), with

$$\Delta^{\mathrm{b}} = 0. \tag{3.68}$$

**The case $(x,y) = (1, n-1)$.** We investigate in greater detail the case where the two pairs of defects are at the two corners. In this case, we find that

$$\tilde{f}_1 = (-1)^{n/2}\tilde{f}_{n-1} = \frac{4}{n}(-1)^{n/2}\sum_{\substack{j=1 \\ j\equiv\frac{n-2}{2}\bmod 2}}^{(n-2)/2} \frac{\sin^2\left(\frac{\pi j}{n}\right)}{\cos\left(\frac{\pi j}{n}\right)}. \tag{3.69}$$

Analysing the large-$n$ asymptotic expansion of this function, we obtain

$$C^{\mathrm{b}}_n(1,n) = 1 + (-1)^{n/2}f_1 + \tilde{f}_{n-1} \simeq \frac{4}{\pi}\ln n + 1 + \frac{4}{\pi}(\gamma + 2\ln 2 - \ln \pi - 1). \tag{3.70}$$

As discussed in Section 7, the coefficient $4/\pi$ in front of $\ln n$ in (3.70) is universal. In the conformal description, the field inserted at the corner is not primary and is instead the logarithmic partner of the identity field. Instead of the usual $\ln n$ contribution of the corner free energy, we find an expansion of the form

$$\ln(Z^{\mathrm{b}}/Z^0) = \ln(\ln n) + \ln(4/\pi) + o(n^0). \tag{3.71}$$

The presence of a $\ln(\ln n)$ dependence is unusual and is a distinctive feature of the logarithmic field inserted in the corner. Similar $\ln(\ln n)$ terms were previously found for the large $n$ asymptotics of the entanglement entropy [47, 48].

## 3.5 Type c: macroscopic collections of defects

The two-point correlation function of type c on the semi-infinite strip is computed from (3.17) with $v^c$ given in (2.37), for which

$$|v^c\rangle = \prod_{i=1}^{(x-1)/2} a_{2i-1}^t \prod_{j=x}^{y} (1 + c_j^\dagger) \prod_{k=(y+2)/2}^{n/2} a_{2k-1}^t |0\rangle. \tag{3.72}$$

The corresponding product reads

$$\langle v^c | w_0 \rangle = \sum_{\substack{L \subset \{x,\ldots,y\} \\ |L|=(y-x+1)/2}} \langle 0| \prod_{\substack{k=n/2 \\ \text{step}=-1}}^{(y+2)/2} a_{2k-1} \prod_{\substack{j=(y-x+1)/2 \\ \text{step}=-1}}^{1} c_{\ell_j} \prod_{\substack{i=(x-1)/2 \\ \text{step}=-1}}^{1} a_{2i-1} |w_0\rangle, \tag{3.73}$$

with $L = \{\ell_1, \ell_2, \ldots, \ell_{(y-x+1)/2}\}$.

**The case $(x, y) = (1, n)$.** We start by discussing the case $x = 1, y = n$. Using Wick's theorem, we find

$$\langle v^c | w_0 \rangle = \frac{(-1)^{(n-2)/2} \omega^{-1}}{\prod_{k=1}^{(n-2)/2} \kappa_k} \sum_{\substack{L \subset \{1,\ldots,n\} \\ |L|=n/2}} \det Q_L, \tag{3.74}$$

where $Q$ is a rectangular matrix of size $n/2 \times n$, with entries

$$Q_{k,\ell} = \omega^{-1} \sin(\tfrac{\pi k(\ell-1)}{n}) + \omega \sin(\tfrac{\pi k \ell}{n}). \tag{3.75}$$

In (3.74), $Q_L$ denotes the restriction of $Q$ to the columns with indices in $L = \{\ell_1, \ell_2, \ldots, \ell_{n/2}\}$. The label $k$ for the rows takes the values $1, \ldots, n/2$. We use the Cauchy-Binet formula for pfaffians given in the following lemma [49].

LEMMA 3.1. *Let $r, t$ be positive integers with $r \leq t$ and $r, t$ even. For $M$ an $r \times t$ matrix and $X$ a $t \times t$ antisymmetric matrix, we have*

$$\sum_{L \subset \{1,\ldots,t\}, |L|=r} \det(M_L) \operatorname{pf}(X_{L,L}) = \operatorname{pf}(MXM^t), \tag{3.76}$$

*where $X_{L,L}$ is the restriction of $X$ to rows and columns with indices in $L$.*

We apply this lemma with

$$X = \begin{pmatrix} 0 & 1 & 1 & 1 & \cdots & 1 \\ -1 & 0 & 1 & 1 & \cdots & 1 \\ -1 & -1 & 0 & 1 & \cdots & 1 \\ -1 & -1 & -1 & 0 & \cdots & 1 \\ \vdots & \vdots & \vdots & \vdots & \ddots & 1 \\ -1 & -1 & -1 & -1 & -1 & 0 \end{pmatrix}, \tag{3.77}$$

for which $\operatorname{pf}(X_{L,L}) = 1$ for all $L$.

For $n/2$ even, $Q$ has an even number of rows and the lemma is applied with the matrix $X$ of size $n$:

$$\langle v^c | w_0 \rangle = \frac{(-1)^{(n-2)/2} \omega^{-1}}{\prod_{k=1}^{(n-2)/2} \kappa_k} \operatorname{pf}(QXQ^t). \tag{3.78}$$

The matrix elements of $QXQ^t$ are obtained by an explicit computation:

$$\left(QXQ^t\right)_{k,\ell} = \begin{cases} 0 & k \equiv \ell \bmod 2, \\ \dfrac{\cos(\frac{\pi\ell}{2n})\cos(\frac{\pi\ell}{n})\sin(\frac{\pi k}{n})}{\sin(\frac{\pi\ell}{2n})\sin\left(\frac{\pi(k-\ell)}{2n}\right)\sin\left(\frac{\pi(k+\ell)}{2n}\right)} & k \equiv 0 \bmod 2,\ \ell \equiv 1 \bmod 2. \end{cases} \tag{3.79}$$

The matrix elements with $k$ odd and $\ell$ even are obtained from (3.79) by recalling that $QXQ^t$ is antisymmetric. Because the entries are zero for $k \equiv \ell \bmod 2$, the rows and columns of $QXQ^t$ can be reordered in such a way that

$$\mathrm{pf}\left(QXQ^t\right) = (-1)^{n(n-4)/32}\mathrm{pf}\begin{pmatrix} 0 & -Y^t \\ Y & 0 \end{pmatrix} = (-1)^{n/4}\det Y. \tag{3.80}$$

This yields

$$\mathrm{pf}\left(QXQ^t\right) = (-1)^{n/4}\prod_{k=1}^{n/4}\frac{\cos\left(\frac{\pi}{n}(k-\frac{1}{2})\right)\cos\left(\frac{2\pi}{n}(k-\frac{1}{2})\right)\sin(\frac{2\pi k}{n})}{\sin\left(\frac{\pi}{n}(k-\frac{1}{2})\right)}$$
$$\times \det_{k,\ell=1}^{n/4}\frac{1}{\sin\left(\frac{\pi}{n}(k-\ell+\frac{1}{2})\right)\sin\left(\frac{\pi}{n}(k+\ell-\frac{1}{2})\right)}. \tag{3.81}$$

Expanding the denominator as $\frac{1}{2}(\cos(\frac{\pi}{n}(2\ell+1)) - \cos(\frac{2\pi k}{n}))$, we evaluate the determinant using (3.36) and find, for $n/2$ even:

$$\langle v^c | w_0 \rangle = \frac{(-1)^{(n-4)/4}\omega^{-1}}{\prod_{k=1}^{(n-2)/2}\kappa_k}\prod_{k=1}^{n/4}\frac{\cos\left(\frac{\pi}{n}(k-\frac{1}{2})\right)\cos\left(\frac{2\pi}{n}(k-\frac{1}{2})\right)\sin(\frac{2\pi k}{n})}{\sin\left(\frac{\pi}{n}(k-\frac{1}{2})\right)} \tag{3.82}$$
$$\times \frac{\prod_{1\leqslant k<\ell\leqslant n/4}\sin\left(\frac{\pi}{n}(k+\ell-1)\right)\sin\left(\frac{\pi}{n}(\ell-k)\right)\sin\left(\frac{\pi}{n}(k+\ell)\right)\sin\left(\frac{\pi}{n}(k-\ell)\right)}{\prod_{k,\ell=1}^{n/4}\sin\left(\frac{\pi}{n}(k-\ell+\frac{1}{2})\right)\sin\left(\frac{\pi}{n}(k+\ell-\frac{1}{2})\right)}.$$

For $n/2$ odd, Lemma 3.1 cannot be applied directly because $Q$ has an odd number of rows. Instead, we use the following corollary.

COROLLARY 3.2. *Let $r \leqslant t$ with $t$ odd. For $M$ an $r \times t$ matrix and $X$ a $(t+1)\times(t+1)$ antisymmetric matrix, we have*

$$\sum_{L\subset\{1,\dots,t\},\,|L|=r}\det(M_L) = \sum_{\substack{L'\subset\{1,\dots,t+1\}\\|L'|=r+1}}\det\hat{M}_{L'} = \mathrm{pf}(\hat{M}X\hat{M}^t) \tag{3.83}$$

*where $X$ is defined in (3.78) and $\hat{M}$ is defined as*

$$\hat{M} = \begin{pmatrix} \boxed{\phantom{xx}M\phantom{xx}} & \begin{matrix} 0 \\ \vdots \\ 0 \\ 0 \end{matrix} \\ \begin{matrix} 0 & \cdots & 0 \end{matrix} & 1 \end{pmatrix}. \tag{3.84}$$

Using the same technique as for $n/2$ even, we find, for $n/2$ odd:

$$
\begin{aligned}
\langle v^{\mathrm{c}}|w_0\rangle &= \frac{(-1)^{(n-2)/2}\omega^{-1}}{\prod_{k=1}^{(n-2)/2}\kappa_k}\mathrm{pf}\big(\hat{Q}X\hat{Q}^{\mathrm{t}}\big) \\
&= \frac{\sqrt{2}\,\omega^{-1}}{\prod_{k=1}^{(n-2)/2}\kappa_k}\prod_{k=1}^{(n-2)/4}\frac{\cos\left(\frac{\pi}{n}(k-\frac{1}{2})\right)\cos\left(\frac{2\pi k}{n}\right)\sin(\frac{2\pi k}{n})}{\sin\left(\frac{\pi}{n}(k-\frac{1}{2})\right)} \\
&\quad\times \frac{\prod_{1\leqslant k<\ell\leqslant(n+2)/4}\sin\left(\frac{\pi}{n}(k+\ell-1)\right)\sin\left(\frac{\pi}{n}(\ell-k)\right)}{\prod_{k=1}^{(n-2)/4}\prod_{\ell=1}^{(n+2)/4}\sin\left(\frac{\pi}{n}(k-\ell+\frac{1}{2})\right)\sin\left(\frac{\pi}{n}(k+\ell-\frac{1}{2})\right)} \\
&\quad\times \prod_{1\leqslant k<\ell\leqslant(n-2)/4}\sin\left(\frac{\pi}{n}(k+\ell)\right)\sin\left(\frac{\pi}{n}(k-\ell)\right).
\end{aligned}
\tag{3.85}
$$

Computing the $\frac{1}{n}$ expansion of the logarithm of $C_n^{\mathrm{c}}(1,n) = 2^{-(n-2)/4}\langle v^{\mathrm{c}}|w_0\rangle/\langle v_0'|w_0\rangle$, we find

$$
\ln C_n^{\mathrm{c}}(1,n) = n\frac{G}{\pi} + \frac{3}{8}\ln n + \frac{1}{8}\left(1 - \frac{35}{3}\ln 2 - 12\ln A + \ln\pi\right) + O(n^{-1}),
\tag{3.86}
$$

where $G$ is Catalan's constant. This holds for both parities of $n/2$. The details of the calculation are discussed in Section 8. This expansion is consistent with (2.6) with

$$
f_s - f_s' = -\frac{G}{\pi}, \qquad \Delta^{\mathrm{c}} = -\frac{3}{32}, \qquad \Delta^0 = 0.
\tag{3.87}
$$

**The general $(x,y)$ case.** In this case, we can write $\langle v^{\mathrm{c}}|w_0\rangle$ as

$$
\langle v^{\mathrm{c}}|w_0\rangle = \frac{(-1)^{(n-2)/2}\omega^{-1}}{\prod_{k=1}^{(n-2)/2}\kappa_k}\sum_{\substack{L\subset\{x,\dots,y\} \\ |L|=d/2}}\det P,
\tag{3.88}
$$

where $d = y - x + 1$ and the matrix $P$ is

$$
P = \begin{pmatrix}
h_{1,1} & h_{1,3} & \cdots & h_{1,x-2} & g_{1,\ell_1} & g_{1,\ell_2} & \cdots & g_{1,\ell_{d/2}} & h_{1,y+1} & h_{1,y+3} & \cdots & h_{1,n-1} \\
h_{2,1} & h_{2,3} & \cdots & h_{2,x-2} & g_{2,\ell_1} & g_{2,\ell_2} & \cdots & g_{2,\ell_{d/2}} & h_{2,y+1} & h_{2,y+3} & \cdots & h_{2,n-1} \\
\vdots & \vdots & & \vdots & \vdots & \vdots & & \vdots & \vdots & \vdots & & \vdots \\
h_{\frac{n-2}{2},1} & h_{\frac{n-2}{2},3} & \cdots & h_{\frac{n-2}{2},x-2} & g_{\frac{n-2}{2},\ell_1} & g_{\frac{n-2}{2},\ell_2} & \cdots & g_{\frac{n-2}{2},\ell_{d/2}} & h_{\frac{n-2}{2},y+1} & h_{\frac{n-2}{2},y+3} & \cdots & h_{\frac{n-2}{2},n-1} \\
0 & 0 & \cdots & 0 & \omega\,\mathrm{i}^{-(\ell_1-1)} & \omega\,\mathrm{i}^{-(\ell_2-1)} & \cdots & \omega\,\mathrm{i}^{-(\ell_{d/2}-1)} & 0 & 0 & \cdots & 0
\end{pmatrix}
\tag{3.89}
$$

and has entries that depend on $L = \{\ell_1, \ell_2, \dots, \ell_{d/2}\}$. To compute $C_n^{\mathrm{c}}(x,y)$, we multiply $P$ by $(\hat{M}^x)^{-1}$, see (3.53a), and find

$$
C_n^{\mathrm{c}}(x,y) = \frac{1}{2^{(d-2)/4}}\sum_{\substack{L\subset\{x,\dots,y\} \\ |L|=d/2}}\det R_L,
\tag{3.90}
$$

with

$$
R_{k,\ell} = \begin{cases} \frac{\omega}{\sqrt{2}}\mathrm{i}^{-(\ell-x)} & k = 1, \\ \omega^{-1}\tilde{R}_{k,\ell-1} + \omega\tilde{R}_{k,\ell} & k = 2,\dots,\frac{d}{2}, \end{cases} \qquad \tilde{R}_{k,\ell} = \tilde{f}_{\frac{x-3}{2}+k,\ell}.
\tag{3.91}
$$

The function $\tilde{f}_{k,\ell}$ is defined in (3.56b). For $d/2$ even, we use Lemma 3.1 and find

$$
C_n^{\mathrm{c}}(x,y) = \frac{1}{2^{(d-2)/4}}\mathrm{pf}_{k,\ell=1}^{d/2}(RXR^{\mathrm{t}})_{k,\ell},
\tag{3.92}
$$

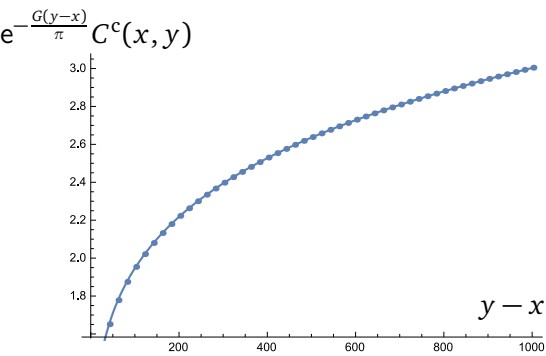

Figure 4: Values for $C^c(x,y)$ in the regime $1 \leqslant y - x \leqslant x, y$ obtained from the pfaffian formulas.

where $R$ is rectangular of size $\frac{d}{2} \times d$, with matrix entries $R_{k,\ell}$, $k = 1, \ldots, \frac{d}{2}$, $\ell = x, \ldots, y$. The matrix $(RXR^t)$ is antisymmetric and its elements read

$$
(RXR^t)_{k,\ell} = \begin{cases} \sqrt{2} \sum_{i=x}^{y-1} \tilde{R}_{\ell,i}, & k = 1, \ell > 1, \\ 2 \sum_{i=x+1}^{y-1} \sum_{j=x}^{i-2} \left( \tilde{R}_{k,j} \tilde{R}_{\ell,i} - \tilde{R}_{k,i} \tilde{R}_{\ell,j} \right) + \sum_{i=x+1}^{y-1} \left( \tilde{R}_{k,i-1} \tilde{R}_{\ell,i} - \tilde{R}_{k,i} \tilde{R}_{\ell,i-1} \right), & k, \ell > 1. \end{cases}
\tag{3.93}
$$

For $d/2$ odd, we use Corollary 3.2 and write the result in terms of the matrix $\hat{R}$ of size $(\frac{d}{2} + 1) \times (d + 1)$:

$$
C^c_n(x,y) = \frac{1}{2^{(d-2)/4}} \mathrm{pf}^{d/2+1}_{k,\ell=1}(\hat{R}X\hat{R}^t)_{k,\ell}.
\tag{3.94}
$$

In this case, it turns out that the only non-zero element of the first row is the last: $(\hat{R}X\hat{R}^t)_{1,\ell} = \delta_{\ell,d/2+1}$. The result is thus the pfaffian of the minor with $k, \ell = 2, \ldots, d/2$:

$$
C^c_n(x,y) = \frac{1}{2^{(d-2)/4}} \mathrm{pf}^{d/2}_{k,\ell=2}(RXR^t)_{k,\ell},
\tag{3.95}
$$

with the matrix elements given in the second line of (3.93).

To obtain the correlator $C^c(x,y)$ in the upper half-plane, we take the limit $n \to \infty$ of each matrix entry and consider the regime where $x, y \gg y - x \gg 1$ for each $\tilde{R}_{k,\ell}$. The function $\tilde{f}_{k,\ell}$ has a well-defined such limit which we compute in Section 6, with the final results given in (6.2) and (6.6). We find, as expected, that the corresponding pfaffians are invariant under translations, namely under $x \to x' + 2a$, $y \to y' + 2a$, $k \to k' + a$ and $\ell \to \ell' + a$, with $a \in \mathbb{Z}$.

We are unfortunately unable to evaluate the resulting pfaffian and obtain an expression in product form. The final pfaffian formula is however convenient for numerical computations. We have computed $C^c(x,y)$ with $y - x$ up to 1003. In Figure 4, we plot $\exp(-G(y-x)/\pi)C^c(x,y)$ as a function of $y - x$, using our prior knowledge from (3.86) that the difference in surface free energy between Neumann and Dirichlet boundary conditions is $G/\pi$. Expecting a power-law behaviour of the form

$$
e^{-G(y-x)/\pi}C^c(x,y) = \frac{K}{(y-x)^{\Delta^c}},
\tag{3.96}
$$

we extract the conformal weight using a fit and find $\Delta^c \simeq -0.09405$. This is consistent with the conformal dimension in (3.86): $\Delta^c = -\frac{3}{32} = -0.09375$.

## 3.6 Type d, e and f: cluster, loop and segment connectivities

The correlation functions for cluster, loop and segment connectivities are computed from (3.17) with $v^{\mathrm{d}}$ given in (2.38) and the parameters $s_i$ respectively fixed to

$$
s_i = \begin{cases} \omega^{-2} & i = 1,\ldots,x, \\ \omega & i = x+1,\ldots,y, \\ 1 & i = y+1,\ldots,n, \end{cases} \qquad
s_i = \begin{cases} \omega^{-2} & i = 1,\ldots,x-1, \\ \omega & i = x, \\ \omega^{-2} & i = x+1,\ldots,y-1, \\ \omega & i = y, \\ 1 & i = y+1,\ldots,n, \end{cases}
$$

$$
s_i = \begin{cases} \omega^{-2} & i = 1,\ldots,x, \\ \mathrm{e}^{\mathrm{i}\theta} & i = x+1,\ldots,y, \\ 1 & i = y+1,\ldots,n, \end{cases}
\tag{3.97}
$$

where we recall that $\omega = \mathrm{e}^{\mathrm{i}\pi/4}$. We have

$$
\begin{aligned}
\langle v^{\mathrm{d}}|w_0\rangle_s &= \langle 0|\prod_{\substack{i=1 \\ \text{step}=-1}}^{n}(s_i^{-1}+s_i c_i)|w_0\rangle = \Big(\prod_{i=1}^{n}s_i^{-1}\Big)\sum_{\substack{L\subset\{1,\ldots,n\} \\ |L|=n/2}}\Big(\prod_{j=1}^{n/2}s_{\ell_j}^2\Big)\langle 0|\prod_{\substack{\ell=1 \\ \text{step}=-1}}^{n/2}c_{\ell_j}|w_0\rangle \\
&= \frac{(-1)^{(n-2)/2}\omega^{-1}}{\prod_{k=1}^{(n-2)/2}\kappa_k}\Big(\prod_{i=1}^{n}s_i^{-1}\Big)\sum_{\substack{L\subset\{1,\ldots,n\} \\ |L|=d/2}}\Big(\prod_{j=1}^{n/2}s_{\ell_j}^2\Big)\det Q_L \\
&= \frac{(-1)^{(n-2)/2}\omega^{-1}}{\prod_{k=1}^{(n-2)/2}\kappa_k}\Big(\prod_{i=1}^{n}s_i^{-1}\Big)\sum_{\substack{L\subset\{1,\ldots,n\} \\ |L|=d/2}}\det(QS)_L \\
&= \frac{(-1)^{(n-2)/2}\omega^{-1}}{\prod_{k=1}^{(n-2)/2}\kappa_k}\Big(\prod_{i=1}^{n}s_i^{-1}\Big)\times\begin{cases} \mathrm{pf}(QSXS^{\mathrm{t}}Q^{\mathrm{t}}) & n/2 \text{ even}, \\ \mathrm{pf}(\hat{Q}\hat{S}X\hat{S}^{\mathrm{t}}\hat{Q}^{\mathrm{t}}) & n/2 \text{ odd}, \end{cases}
\end{aligned}
\tag{3.98}
$$

where $Q$ is defined in (3.75) and $S$ is an $n \times n$ matrix with entries $S_{k,\ell} = (s_k)^2\delta_{k,\ell}$. We used Lemma 3.1 and Corollary 3.2 at the last equality. Together with (3.27), this yields

$$
C_n^{\mathrm{d,e,f}}(x,y) = (-1)^{n(n-2)/8}\Big(\prod_{i=1}^{n}s_i^{-1}\Big)\frac{2^{(n-4)/4}}{n^{(n-2)/4}}\times\begin{cases} \mathrm{pf}(QSXS^{\mathrm{t}}Q^{\mathrm{t}}) & n/2 \text{ even}, \\ \mathrm{pf}(\hat{Q}\hat{S}X\hat{S}^{\mathrm{t}}\hat{Q}^{\mathrm{t}}) & n/2 \text{ odd}. \end{cases}
\tag{3.99}
$$

We have not found out how to push the calculation further and instead evaluated the exact formulas using a computer. For type d and e, the computation was performed for $n = 1500$, $x = 750$ and $750 < y \leqslant 1500$. The results are displayed in Figure 5. Using power-law fits of the form

$$
\mathrm{e}^{-\frac{Gn}{\pi}}C_n^{\mathrm{d,e}}(x,y) = \frac{K}{(y-x)^{\Delta^{\mathrm{d,e}}}},
\tag{3.100}
$$

we obtained $\Delta^{\mathrm{d}} \simeq 0.37247$ and $\Delta^{\mathrm{e}} \simeq 0.994845$. This is consistent with the values obtained in Section 4.4 and Section 4.5:

$$
\Delta^{\mathrm{d}} = \frac{3}{8} = 0.375, \qquad \Delta^{\mathrm{e}} = 1.
\tag{3.101}
$$

For correlators of type f, we performed the same numerical analysis as those presented in Figure 5, for multiple values of $\tau \in [0,2]$ and for $n = 1000$, and estimated the conformal weight in each case. The results are given in Figure 6. Our investigation separates the cases where $y-x$ is even and odd, for which one could expect different behaviours. Indeed, $Z^{\mathrm{f}}$ is a polynomial in $\tau$; for $y-x$ even, the constant term is non-zero and equals $Z^{\mathrm{d}}$. For $y-x$ odd,

$$e^{-\frac{Gn}{\pi}}C_n^d(x,y)$$ $$e^{-\frac{Gn}{\pi}}C_n^e(x,y)$$

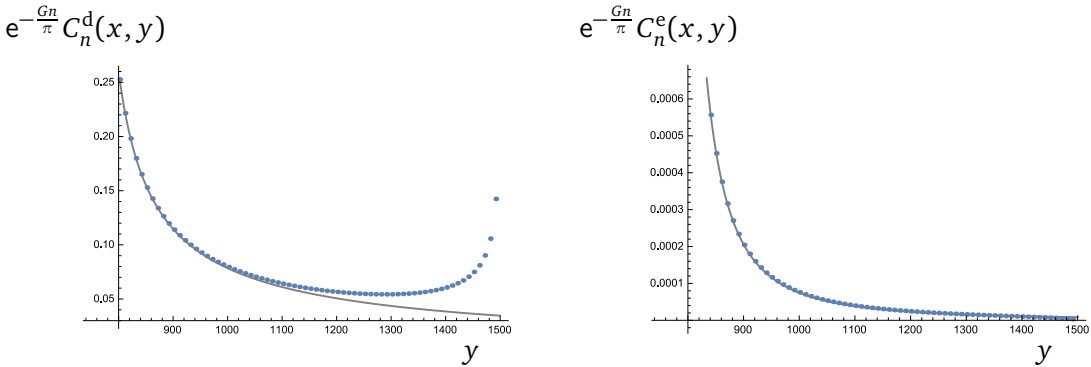

Figure 5: Values of $C_n^d(x,y)$ and $C_n^e(x,y)$ obtained from the pfaffian formulas, for $n = 1500$ and $x = 750$. The power-law behaviour is only approximate for finite system sizes and gets progressively worse as $y$ gets closer to the right corner.

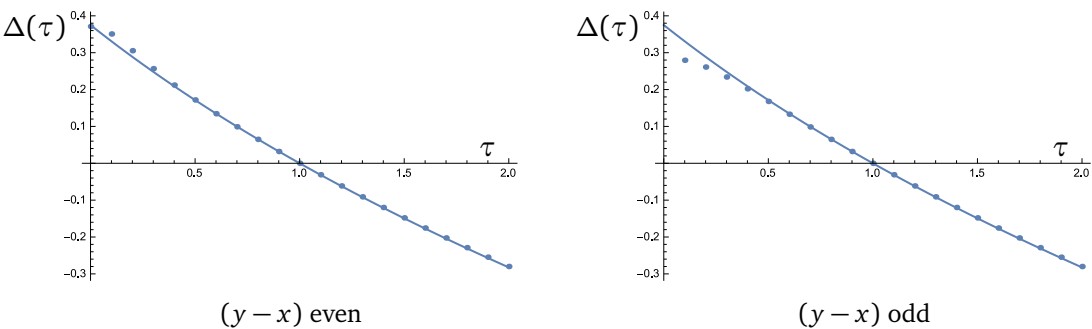

$(y-x)$ even $\qquad\qquad\qquad\qquad\qquad$ $(y-x)$ odd

Figure 6: The value of $\Delta$ for $C_n^f(x,y)$ as a function of $\tau$, for $y - x$ even and odd. Each data point is obtained from a power-law fit of the pfaffian expression (3.99) for $n = 1000$.

$Z^f|_{\tau=0} = 0$, so the constant term vanishes. In Figure 6 however, it seems that the conformal weights are identical in the odd and even cases. In both panels, we have plotted the curve

$$\Delta = \Delta_{4\theta+1,4\theta+1} = \frac{\theta}{\pi}(1 + \frac{2\theta}{\pi}). \tag{3.102}$$

We discuss in Section 4.5 how this curve is obtained and provide possible explanations for the deviation between the numerics and the theoretical curve near $\tau = 0$.

# 4 Predictions from conformal invariance

In this section, we use the hypothesis of conformal invariance to predict the leading behaviour of the correlation functions. In particular, it will become clear why some of the lattice correlators studied in Section 3 exhibit pure power-law behaviours whereas others have logarithmic corrections.

For two-point functions of primary fields of weight $\Delta$, the transformation law between two domains $\mathbb{D}_1$ and $\mathbb{D}_2$ is

$$\langle \phi(y_0)\phi(y_1)\rangle_{\mathbb{D}_1} = \left|\frac{dy}{dz}\right|_{y=y_0}^{-\Delta} \left|\frac{dy}{dz}\right|_{y=y_1}^{-\Delta} \langle \phi(z_0)\phi(z_1)\rangle_{\mathbb{D}_2}. \tag{4.1}$$

The derivations presented in this section combine (4.1) with the knowledge of the finite-size corrections for the eigenvalues of the lattice transfer matrices. These are given in Section 9 for the Temperley-Lieb algebra and its generalisations with blobs on one and two boundaries.

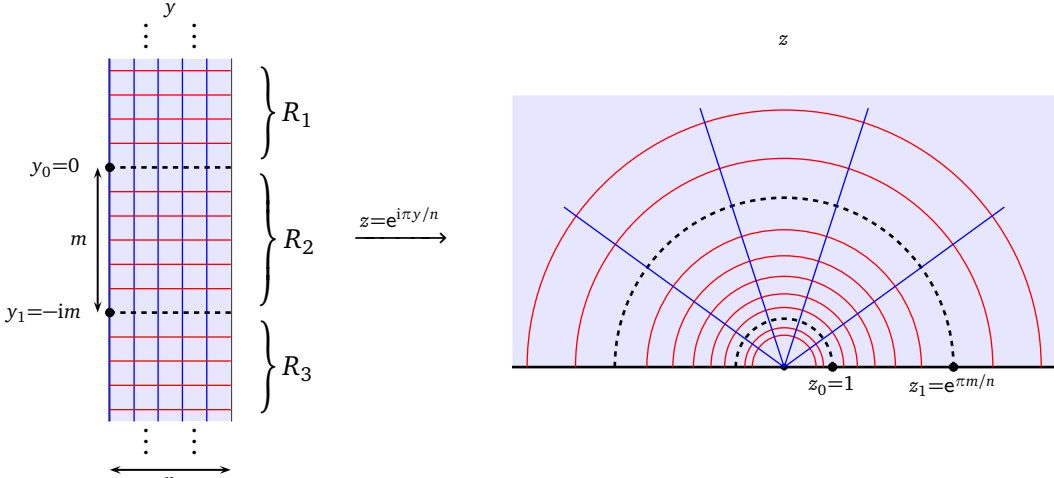

Figure 7: The conformal map between the domains $\mathbb{V}$ and $\mathbb{H}$. The two marked points $y_0$ and $y_1$ in $\mathbb{V}$ are mapped in $\mathbb{H}$ to $z_0$ and $z_1$.

Using these two ingredients, we are able to give predictions for the correlators that match the results found in Section 3 for critical dense polymers. The technique in fact gives predictions for $\beta \in (-2, 2)$.

For $\mathbb{D}_1$, we take the infinite vertical strip of width $n$ and denote it $\mathbb{V}$. For $\mathbb{D}_2$, we take the upper half-plane $\mathbb{H}$. The map between these two domains is

$$z = e^{i\pi y/n} \tag{4.2}$$

and is illustrated in Figure 7. In Sections 4.1 to 4.6, we decorate the boundary of $\mathbb{V}$ with simple arcs and defects corresponding to each type of correlation function. Each time, we split $\mathbb{V}$ in three regions $R_1$, $R_2$ and $R_3$. The horizontal dashed lines that bound $R_2$ in Figure 7 are leveled with the two marked points $y_0 = 0$ and $y_1 = -im$ on the left segment for which we want to compute the correlation function.

## 4.1 Type a: two isolated defects

For the correlators of type a, we decorate the domains $\mathbb{V}$ and $\mathbb{H}$ with simple arcs and two defects, as in Figure 8. The regions $R_1$ and $R_3$ are drawn as finite; in the calculation below, we consider the limit wherein their vertical length is infinite.

The partition function on $\mathbb{V}$ can be expressed in terms of two states $\psi_1$ and $\psi_3$. These are obtained as the linear combination of link states coming out of $R_1$ and $R_3$. Concretely, to construct $\psi_1$ and $\psi_3$, we first define

$$\phi_1 = \phi_3 = \lim_{m' \to \infty} \left( \frac{\hat{D}(\frac{\lambda}{2})}{\hat{\Lambda}_0} \right)^{m'} v_0, \tag{4.3}$$

where

$$\hat{D}(u) = \begin{array}{|c|c|c|c|c|} \hline u & u & \cdots & \cdots & u \\ \hline u & u & \cdots & \cdots & u \\ \hline \end{array} \tag{4.4}$$

and $v^0$ is defined in (2.28). The states $\phi_1$ and $\phi_3$ are elements of $\mathsf{V}_{n+1,0}$ and $\hat{\Lambda}_0$ is the ground-state eigenvalue of $\hat{D}(\frac{\lambda}{2})$ in this representation. We then obtain $\psi_1, \psi_3 \in \mathsf{V}_{n,1}$ from $\phi_1, \phi_3$ by converting the first node to a defect.

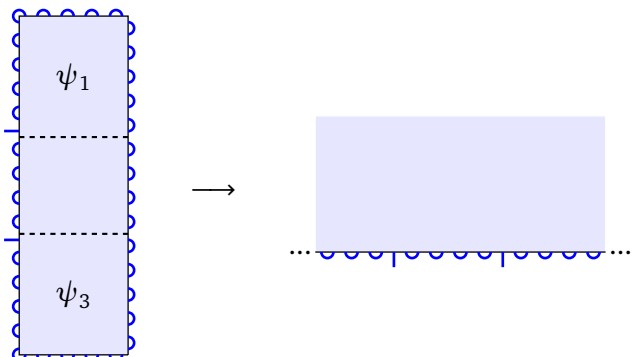

Figure 8: Boundary conditions on $\mathbb{V}$ and $\mathbb{H}$ corresponding to the correlator of type a.

With $y_0 = 0$ and $y_1 = -im$, we have

$$C_{\mathbb{V}}^{\mathrm{a}}(y_0, y_1) = \frac{1}{Z^0} \psi_3 \cdot (D(\tfrac{\lambda}{2})^{m/2} \psi_1)\big|_{\mathsf{V}_{n,1}}. \tag{4.5}$$

This computation is performed in the standard module with one defect, and with the bilinear form defined in Section 2.3 specialised to $\gamma = 1$. The generating function for the conformal spectrum of $D(u)$ on $\mathsf{V}_{n,d}$ is given in (9.4). For $d = 1$, there is a unique ground state $w_1$. Its eigenvalue $\Lambda_1$ has a $1/n$ expansion of the form (9.1) with $\Delta = \Delta_{1,2}$ [9]. This is true for $\beta$ generic.

For $m \gg n$, the leading contribution to the right-hand side of (4.5) is from the ground state. Decomposing $\psi_1$ in terms of the eigenstates of $D(\tfrac{\lambda}{2})$ as $\psi_1 = \sum_w \alpha_w w$, we find

$$C_{\mathbb{V}}^{\mathrm{a}}(y_0, y_1) \xrightarrow{m \gg n} \frac{\Lambda_1^{m/2}}{Z^0} \alpha_{w_1} (\psi_3 \cdot w_1)\big|_{\mathsf{V}_{n,1}}. \tag{4.6}$$

The factors $\alpha_{w_1}$ and $(\psi_3 \cdot w_1)$ do not depend on $m$, whereas $\Lambda_1^{m/2}$ behaves as

$$\Lambda_1^{m/2} = \exp\left(-\tfrac{mn}{2} f_b - \tfrac{m}{2} f_s - \tfrac{\pi m}{n}(\Delta_{1,2} - \tfrac{c}{24}) + \dots\right). \tag{4.7}$$

where $f_b$ is the bulk free energy. For generic values of $\beta$, the partition function in the denominator is computed with only simple arcs on the boundary of $\mathbb{V}$, and behaves as

$$Z^0 \simeq \Lambda_0^{m/2} = \exp\left(-\tfrac{mn}{2} f_b - \tfrac{m}{2} f_s - \tfrac{\pi m}{n}(\Delta_{1,1} - \tfrac{c}{24}) + \dots\right). \tag{4.8}$$

Recalling that $\Delta_{1,1} = 0$, we find

$$C_{\mathbb{V}}^{\mathrm{a}}(w_1, w_2) \xrightarrow{m \gg n} \tilde{K} e^{-\frac{\pi m}{n}\Delta_{1,2}}, \tag{4.9}$$

where $\tilde{K}$ is a constant that does not depend on $m$. This equality holds for generic and non-generic values of $\beta$, including $\beta = 0$. Indeed in this last case, $Z^0$ is instead the partition function where the two defects in Figure 8 are adjacent. The calculation is computed with the standard module with two defects and yields $Z^0 \simeq \Lambda_2^{m/2}$ where $\Lambda_2$ is the ground state eigenvalue in $\mathsf{V}_{n,2}$. The finite-size correction for $\Lambda_2$ involves $\Delta_{1,3}$ which is also zero.

On $\mathbb{V}$, the correlation function of type a thus varies exponentially in the distance $m = |y_0 - y_1|$ between the two points. From a CFT perspective, the insertion of a single defect on the boundary corresponds to the insertion of a boundary changing field $\varphi^{\mathrm{a}}$, and $C^{\mathrm{a}}(x, y)$ is the two-point function of this field. Supposing that this field is primary, the correlation function in the upper half-plane is

$$C_{\mathbb{H}}^{\mathrm{a}}(z_0, z_1) \simeq \langle \varphi^{\mathrm{a}}(z_0) \varphi^{\mathrm{a}}(z_1) \rangle_{\mathbb{H}} = \frac{K}{|z_0 - z_1|^{2\Delta^{\mathrm{a}}}}. \tag{4.10}$$

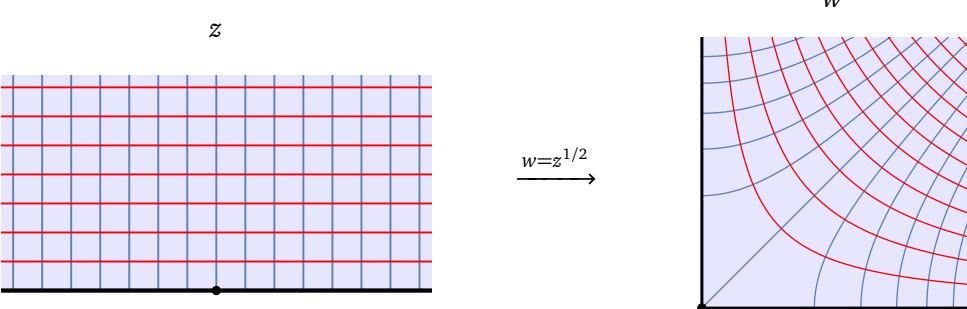

Figure 9: The map $w = z^{1/2}$ from $\mathbb{H}$ to $\mathbb{Q}$.

To determine $\Delta^{\mathrm{a}}$, we note that we can obtain $C_{\mathbb{V}}^{\mathrm{a}}(y_0, y_1)$ from $C_{\mathbb{H}}^{\mathrm{a}}(z_0, z_1)$ by using the transformation law (4.1) with the map (4.2):

$$C_{\mathbb{V}}^{\mathrm{a}}(y_0, y_1) = \frac{K' e^{\frac{\pi m}{n} \Delta^{\mathrm{a}}}}{(e^{\frac{\pi m}{n}} - 1)^{2\Delta^{\mathrm{a}}}} \xrightarrow{m \gg n} K' e^{-\frac{\pi m}{n} \Delta^{\mathrm{a}}}, \qquad K' = (\tfrac{\pi}{n})^{2\Delta^{\mathrm{a}}} K. \tag{4.11}$$

Comparing with (4.9), we find $K' = \tilde{K}$ and $\Delta^{\mathrm{a}} = \Delta_{1,2}$. For critical dense polymers, the conformal prediction for the correlation function of type a on $\mathbb{H}$ is therefore a power-law increase with

$$\Delta^{\mathrm{a}} = \Delta_{1,2} = -\frac{1}{8}. \tag{4.12}$$

This reproduces the exact result (3.45).

As a final remark, we note that the correlation function in the first quadrant $\mathbb{Q}$ can be obtained from (4.10) by applying the transformation law with $w = z^{1/2}$. This transformation is illustrated in Figure 9. The result is

$$C_{\mathbb{Q}}^{\mathrm{a}}(w_0, w_1) = \frac{K'' |w_0|^{\Delta^{\mathrm{a}}} |w_1|^{\Delta^{\mathrm{a}}}}{\left| w_0^2 - w_1^2 \right|^{2\Delta^{\mathrm{a}}}}, \qquad K'' = 2^{2\Delta^{\mathrm{a}}} K. \tag{4.13}$$

With $\Delta^{\mathrm{a}} = -\frac{1}{8}$, this precisely reproduces the lattice result (3.43).

## 4.2 Type b: two pairs of defects

For the correlators of type b, we decorate the domains $\mathbb{V}$ and $\mathbb{H}$ with simple arcs and two pairs of defects, as in Figure 10. Using the terminology of Section 3.4, we first consider the correlator $C^{\mathrm{b}}_{\frown}(x, y)$ where defects belonging to a same pair are not connected together. We follow the same ideas as in Section 4.1 and write the partition function on $\mathbb{V}$ using two states $\psi_1, \psi_3$, which in this case have two defects. These should not connect, so the computation is performed in $\mathsf{V}_{n,2}$. In this case, the conformal weight appearing in the finite-size term of the ground-state eigenvalue of $\mathsf{V}_{n,2}$ is $\Delta_{1,3}$ [9]. We find:

$$C^{\mathrm{b}}_{\mathbb{V},\frown}(y_0, y_1) = \frac{1}{Z^0} \psi_3 \cdot \left( D(\tfrac{\lambda}{2})^{m/2} \psi_1 \right)\Big|_{\mathsf{V}_{n,2}} \xrightarrow{m \gg n} \tilde{K} \left( \frac{\Lambda_2}{\Lambda_0} \right)^{m/2} \simeq \tilde{K} e^{-\frac{\pi m}{n} \Delta_{1,3}}. \tag{4.14}$$

To understand the conformal interpretation, we repeat the argument used in Section 4.1. Inserting a pair of adjacent defects should correspond to the insertion of a primary field $\varphi^{\mathrm{b}}$ of weight $\Delta^{\mathrm{b}}$ for which the two-point correlator on $\mathbb{H}$ has the same form as (4.10). Applying the conformal transformation, the same two-point correlator on $\mathbb{V}$ is found to have the form

$$C^{\mathrm{b}}_{\mathbb{V},\frown}(y_0, y_1) = \frac{K' e^{\frac{\pi m}{n} \Delta^{\mathrm{b}}}}{(e^{\frac{\pi m}{n}} - 1)^{2\Delta^{\mathrm{b}}}} \xrightarrow{m \gg n} K' e^{-\frac{\pi m}{n} \Delta^{\mathrm{b}}}. \tag{4.15}$$

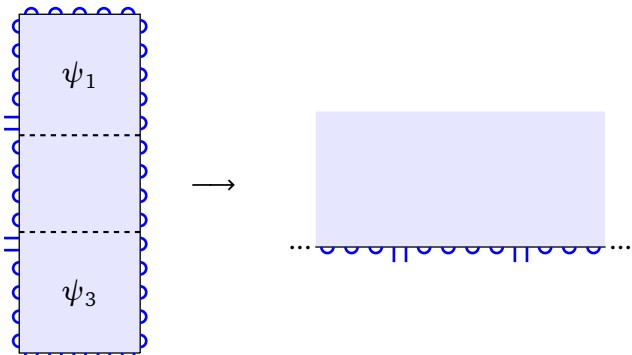

Figure 10: Boundary conditions on $\mathbb{V}$ and $\mathbb{H}$ corresponding to the correlators of type b.

Comparing with (4.14), we conclude that $\Delta^{\mathrm{b}} = \Delta_{1,3}$. For critical dense polymers, we have $\Delta_{1,3} = 0$, which is consistent with the exact result (3.48), where this correlator is independent of the distance between the two points.

To compute $C^{\mathrm{b}}_{\curvearrowright\curvearrowright}(x, y)$, we perform a calculation similar to (4.14) but in the link module $\mathsf{L}_n$, to allow defects in the same pair to connect:

$$C^{\mathrm{b}}_{\mathbb{V},\curvearrowright\curvearrowright}(y_0, y_1) = \frac{1}{Z^0}\,\psi_3 \cdot \bigl(\boldsymbol{D}(\tfrac{\lambda}{2})^{m/2}\psi_1\bigr)\big|_{\mathsf{L}_n}. \tag{4.16}$$

In contrast with (4.14), the states $\psi_1$ and $\psi_3$ in (4.16) are linear combinations of link states with zero or two defects. Acting with $\boldsymbol{D}(\tfrac{\lambda}{2})^{m/2}$ on $\psi_1$ mixes these two subsectors further. The product $v_1 \cdot v_2$ appearing in (4.16) is then defined to be zero if both $v_1$ and $v_2$ have two defects, with the defects of $v_1$ connected to those of $v_2$. This product is $\beta^{n_\beta}$ otherwise.

The spectrum of $\boldsymbol{D}(\tfrac{\lambda}{2})$ in $\mathsf{L}_n$ is the union of its spectra in the subsectors $\mathsf{V}_{n,d}$. The state $\psi_1$ has non-zero contributions along $w_0$ and $w_2$, the ground states corresponding to the two subsectors with $d = 0$ and $d = 2$. For $\beta$ generic, the two corresponding eigenvalues $\Lambda_0$ and $\Lambda_2$ are different and the leading and subleading contributions to $C^{\mathrm{b}}_{\mathbb{V},\curvearrowright\curvearrowright}(y_0, y_1)$, for $m \gg n$, are

$$C^{\mathrm{b}}_{\mathbb{V},\curvearrowright\curvearrowright}(y_0, y_1) \xrightarrow{m \gg n} \frac{1}{Z^0}\bigl(\Lambda_0^{m/2}\alpha_{w_0}\psi_3 \cdot w_0 + \Lambda_2^{m/2}\alpha_{w_2}\psi_3 \cdot w_2 + \dots\bigr)$$

$$= \tilde{K}_0\Bigl(\frac{\Lambda_0}{\Lambda_0}\Bigr)^{m/2} + \tilde{K}_2\Bigl(\frac{\Lambda_2}{\Lambda_0}\Bigr)^{m/2} + \dots = \tilde{K}_0 + \tilde{K}_2\mathrm{e}^{-\frac{\pi m}{n}\Delta_{1,3}} + \dots, \tag{4.17}$$

where $\tilde{K}_0$ and $\tilde{K}_2$ are independent of $m$.

In fact, subleading orders can also be computed in (4.14). In this case, one obtains a sum of the form $\sum_i \tilde{K}_i \mathrm{e}^{-\frac{\pi m}{n}\Delta(i)}$ where $\Delta(i)$ differs from $\Delta_{1,3}$ by an integer. Presumably, these subleading terms reproduce the Taylor expansion of the function $K'\mathrm{e}^{\pi m \Delta^{\mathrm{b}}/n}/(\mathrm{e}^{\pi m/n} - 1)^{2\Delta^{\mathrm{b}}}$ as in (4.15). The difference in (4.17) is that $\Delta_{1,1} = 0$ and $\Delta_{1,3}$ are not integer-spaced in general. This points to the fact that the conformal field that inserts two adjacent defects forced to connect together is not primary, and is instead a composite field that mixes two primary fields of dimensions $\Delta_{1,1}$ and $\Delta_{1,3}$. The field $\varphi^{\mathrm{b}}$ is then interpreted as the fusion of two fields of type a, which is consistent with the operator product expansion (OPE) for these fields:

$$\varphi^{\mathrm{b}} \simeq \varphi^{\mathrm{a}} \times \varphi^{\mathrm{a}} = \varphi_{1,2} \times \varphi_{1,2} = \varphi_{1,1} + \varphi_{1,3}. \tag{4.18}$$

The case $\beta = 0$ is special because $\Delta^{\mathrm{b}} = \Delta_{1,3} = 0$ is equal to $\Delta_{1,1}$. The eigenvalues $\Lambda_0$ and $\Lambda_2$ are equal and the two corresponding states form a Jordan cell in $\mathsf{L}_n$:

$$\boldsymbol{D}(\tfrac{\lambda}{2})w_0 = \Lambda_0 w_0, \qquad \boldsymbol{D}(\tfrac{\lambda}{2})w_2 = \Lambda_0 w_2 + f(\tfrac{\pi}{4})w_0. \tag{4.19}$$

From the representation theoretic point of view, this Jordan cell belongs to a projective module of $\mathsf{TL}_n(\beta = 0)$ that is reducible yet indecomposable. This will be discussed further in Section 5. We find

$$\psi_3 \cdot (D(\tfrac{\lambda}{2})^{m/2}\psi_1)\big|_{\mathsf{L}_n} \xrightarrow{m \gg n} \Lambda_0^{m/2}\Big(\alpha_{w_0}(\psi_3 \cdot w_0) + \alpha_{w_2}\big(\psi_3 \cdot w_2 + \tfrac{m}{2}\tfrac{f(\frac{\pi}{4})}{\Lambda_0}\psi_3 \cdot w_0\big)\Big) + \dots \quad (4.20)$$

and therefore

$$C^{\mathrm{b}}_{\mathbb{V},\frown\frown}(y_0, y_1) \xrightarrow{m \gg n} \tilde{K}_0 + \tilde{K}_2 m + \dots \,. \quad (4.21)$$

Because (4.21) is linear in $m$ instead of exponential, the assumption that the corresponding field is a primary field or a composition thereof fails here and must be modified.

For $c = -2$, let $\omega(z)$ be a conformal field of weight zero and a logarithmic partner of the identity field $\varphi_{1,1}(z)$. Under a conformal transformation $w = f(z)$, the transformation law for $\omega(z)$ is

$$\omega(z) \to \omega(w) + \lambda_0\, \varphi_{1,1}(w) \ln\left|\frac{\mathrm{d}w}{\mathrm{d}z}\right|^2, \quad (4.22)$$

where $\lambda_0$ is a constant. The one- and two-point functions on $\mathbb{H}$ are

$$\langle\varphi_{1,1}(z)\rangle_{\mathbb{H}} = 0, \qquad \langle\omega(z)\rangle_{\mathbb{H}} = \lambda_1, \qquad \langle\omega(z_0)\omega(z_1)\rangle_{\mathbb{H}} = -4\lambda_0\lambda_1 \ln|z_0 - z_1| + \lambda_1\lambda_2, \quad (4.23)$$

where $\lambda_1$ and $\lambda_2$ are constants. Applying the transformation law (4.1) with (4.2), we find

$$\langle\omega(y_0)\omega(y_1)\rangle_{\mathbb{V}} = -4\lambda_0\lambda_1 \ln(e^{\frac{\pi m}{n}} - 1) + 2\lambda_0\lambda_1\pi\tfrac{m}{n} + \lambda_1\lambda_2' \xrightarrow{m \gg n} -2\lambda_0\lambda_1\pi\tfrac{m}{n} + \lambda_1\lambda_2', \quad (4.24)$$

where $\lambda_2' = \lambda_2 + 4\lambda_0 \ln(\pi/n)$. This has the desired linear dependence in $m$, as in (4.21). The insertion of two defects constrained to connect together is thus interpreted as the insertion of a field $\omega(z)$. In this setting, the partition function $Z^0$ is the one-point function $\langle\omega(z)\rangle$. It is independent of $z$, consistent with the lattice result discussed at the end of Section 2.5. The correlation function of type b is

$$C^{\mathrm{b}}_{\frown\frown}(z_0, z_1) = \frac{\langle\omega(z_0)\omega(z_1)\rangle}{\langle\omega(z_0)\rangle}. \quad (4.25)$$

On $\mathbb{H}$, this yields

$$C^{\mathrm{b}}_{\mathbb{H},\frown\frown}(z_0, z_1) = -4\lambda_0 \ln|z_0 - z_1| + \lambda_2. \quad (4.26)$$

This is precisely the lattice result (3.67), with

$$\lambda_0 = -\frac{1}{2\pi} \qquad \lambda_2 = \frac{2}{\pi}(\gamma + \ln 2). \quad (4.27)$$

The constant $\lambda_0$ is in fact universal; the value found here is the same as the one obtained in [50, Equation (76)]. In contrast, $\lambda_1$ and $\lambda_2$ are not universal.

Finally, the result in the first quadrant is obtained from (4.26) by using the transformation law (4.1) with $w = z^{1/2}$:

$$C^{\mathrm{b}}_{\mathbb{Q}}(w_0, w_1) = -4\lambda_0 \ln|w_0^2 - w_1^2| + 2\lambda_0 \ln|w_0| + 2\lambda_0 \ln|w_1| + \lambda_2'', \qquad \lambda_2'' = \lambda_2 + 4\lambda_0 \ln 2. \quad (4.28)$$

This is identical to the lattice result (3.65).

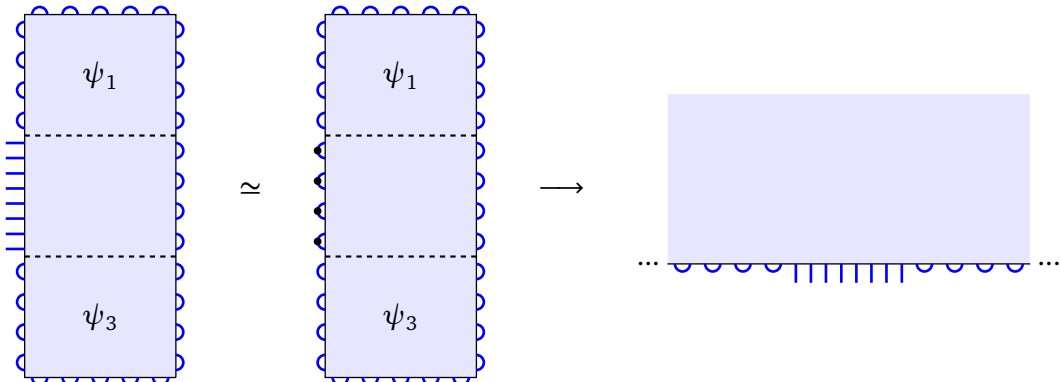

Figure 11: Boundary conditions on $\mathbb{V}$ and $\mathbb{H}$ corresponding to the correlators of type c.

### 4.3  Type c: macroscopic collections of defects

For the correlators of type c, we decorate the domains $\mathbb{V}$ and $\mathbb{H}$ with simple arcs and defects as in the first and last panels of Figure 11. On $\mathbb{V}$, the two states $\psi_1$ and $\psi_3$ for the regions $R_1$ and $R_3$ are

$$\psi_1 = \psi_3 = \lim_{m' \to \infty} \left( \frac{\boldsymbol{D}(\frac{\lambda}{2})}{\Lambda_0} \right)^{m'} v_0. \tag{4.29}$$

To obtain the partition function on $\mathbb{V}$, one should act $m/2$ times on $\psi_1$ with a double-row transfer tangle that has two defects attached to its left end, and then take the product with $\psi_3$. We do the calculation using the blob algebra, recalling that the fugacity of the boundary loops is set to one. The definition of this algebra is reviewed in Section 9. For the region $R_2$, we replace pairs of consecutive defects by arcs equipped with blobs, as in the central panel of Figure 11, and set the fugacity of loops containing a blob to $\beta_1 = 1$. In terms of the parameterisation (9.9), this holds for

$$r_1 = \frac{\pi - \lambda}{2\lambda}. \tag{4.30}$$

The corresponding transfer matrix is

$$\boldsymbol{D}^{(1)}(u) = \begin{array}{|c|c|c|c|c|} \hline u & u & \cdots & \cdots & u \\ \hline u & u & \cdots & \cdots & u \\ \hline \end{array} . \tag{4.31}$$

In this setting, the states $\psi_1$ and $\psi_3$ are elements of the standard module of the blob algebra with zero defects, $\mathsf{V}^{(1)}_{n,0}$. The two-point function on $\mathbb{V}$ is given by

$$C^{\mathrm{c}}_{\mathbb{V}}(y_0, y_1) = \frac{1}{Z^0} \psi_3 \cdot \left( (\boldsymbol{D}^{(1)}(\tfrac{\lambda}{2}))^{m/2} \psi_1 \right) \Big|_{\mathsf{V}^{(1)}_{n,0}}. \tag{4.32}$$

The corresponding bilinear form here is adapted so that the loops with a blob are given the fugacity $\beta_1 = 1$.

In the limit $m \gg n$, the leading behaviour is a contribution from the ground state. In this case, the maximal eigenvalues in the numerator and denominator belong to transfer matrices with different boundary conditions on the left. As pointed out in (3.87), the difference in surface energy is $-\frac{G}{\pi}$. Using the expression (9.12) for the conformal character corresponding to $\mathsf{V}^{(1)}_{n,0}$, we find

$$C^{\mathrm{c}}_{\mathbb{V}}(y_0, y_1) \xrightarrow{m \gg n} \mathrm{e}^{\frac{Gm}{\pi}} \tilde{K} \mathrm{e}^{-\frac{\pi m}{n} \Delta_{r_1, r_1}} = \mathrm{e}^{\frac{Gm}{\pi}} \tilde{K} \mathrm{e}^{-\frac{\pi m}{n} \Delta_{0, 1/2}}, \tag{4.33}$$

where we used the symmetries of the Kac formula (2.2) at the last step. The correlator of type c thus takes the form of a non-universal boundary term times the two point function $\langle \varphi^c(z_0)\varphi^c(z_1)\rangle$ of a primary field of weight $\Delta^c = \Delta_{0,1/2}$. On the upper half-plane, the universal part of the two-point function is then

$$C_{\mathbb{H}}^c(z_0, z_1) \simeq \langle \varphi^c(z_0)\varphi^c(z_1)\rangle_{\mathbb{H}} = \frac{K}{|z_0 - z_1|^{2\Delta_{0,1/2}}}. \tag{4.34}$$

This result holds for all $\beta$. The conformal dimension of this boundary changing field was previously obtained in [30, Equation (3.20)] in the case $p' = p + 1$, where it was written as $\Delta_{p/2, p/2}$. For dense polymers, the conformal weight is

$$\Delta^c = \Delta_{0,1/2} = -\frac{3}{32}, \tag{4.35}$$

consistent with the exact results (3.87) and the numerics of Figure 4.

The case of percolation, namely $\beta = 1$ corresponding to $t = 2/3$, provides a second verification of the result (4.34). In this case, the model does not distinguish between arcs and defects in the boundary condition, as both bulk and boundary loops have weight 1. For this model, one expects that $C_{\mathbb{H}}^c(z_0, z_1) = 1$ for all $z_0$ and $z_1$. In this case, $\Delta_{0,1/2} = 0$ and (4.34) is indeed independent of $|z_0 - z_1|$.

Another nice remark can be made. From the geometric definition of the fields $\varphi^a$ and $\varphi^c$, we expect their OPE to be of the form

$$\varphi^a \times \varphi^c = \varphi^c + \dots . \tag{4.36}$$

Indeed, on the lattice, this corresponds to imposing the boundary condition

$$\dots \underset{a \qquad c}{\smile\smile\smile\smile\mid\smile\smile}\mid\mid\mid\mid\mid\mid\mid \dots \tag{4.37}$$

to the lower edge of the rectangle. As the isolated defect approaches the transition midpoint between the Dirichlet and Neumann boundary conditions, it becomes indiscernable from the rest of the defects. Little is known about the fusion of fields that are not in the Kac table, yet one expects that fusing $\varphi_{1,2}$ with a field $\varphi_{r,s}$ changes the $s$ index by $+1$ or $-1$. In particular,

$$\varphi^a \times \varphi^c = \varphi_{1,2} \times \varphi_{0,1/2} = \varphi_{0,-1/2} + \varphi_{0,3/2}. \tag{4.38}$$

At the level of conformal weights, $\Delta_{0,-1/2} = \Delta_{0,1/2} = \Delta^c$, consistent with the geometric interpretation.

## 4.4 Type d: cluster connectivities

For the correlators of type d, we decorate the domains $\mathbb{V}$ and $\mathbb{H}$ with simple arcs and defects as in the first and last panels of Figure 12. The defects lying between the two marked points $y_0$ and $y_1$ are drawn in green. The cluster starting from $y_0$ reaches $y_1$ if and only if each green defect is connected to another green defect. Although this is not illustrated in Figure 12, the cluster is allowed to touch the boundary in multiple points, either between $y_0$ and $y_1$ or outside of this interval.

To carry out the calculation, we use the generalisation of the Temperley-Lieb algebra with blobs on both boundaries [27–29]. The definition of this algebra is reviewed in Section 9. In the regions $R_1$ and $R_3$, we replace pairs of adjacent defects by square blobs, as shown in the second panel of Figure 12. We impose that each closed loop containing a square blob has a weight 1. The states $\psi_1$ and $\psi_3$ are thus identical and are linear combinations of link states with arcs that may or may not be equipped with a square blob, and no defects.

For $R_2$, we replace the defects on the left and right boundaries by round and square blobs respectively and impose that a loop containing a single blob (round or square) has weight one, whereas a loop that contains both types of blobs has weight zero: $\beta_1 = \beta_2 = 1$, $\beta_{12}^e = 0$. In terms of the parameterisations given in Section 9, this holds for

$$r_1 = r_2 = \frac{\pi - \lambda}{2\lambda}, \qquad r_{12} = r_1 + r_2 + 1 = \frac{\pi}{\lambda}. \tag{4.39}$$

The computation of $C_{\mathbb{V}}^d(y_0, y_1)$ is then performed in the standard module with zero defects:

$$C_{\mathbb{V}}^d(y_0, y_1) = \frac{1}{Z^0} \, \psi_3 \cdot \left( (\boldsymbol{D}^{(2)}(\tfrac{\lambda}{2}))^{m/2} \psi_1 \right) \big|_{\mathsf{V}_{n,0}^{(2)}}, \tag{4.40}$$

where

$$\boldsymbol{D}^{(2)}(u) = \quad \text{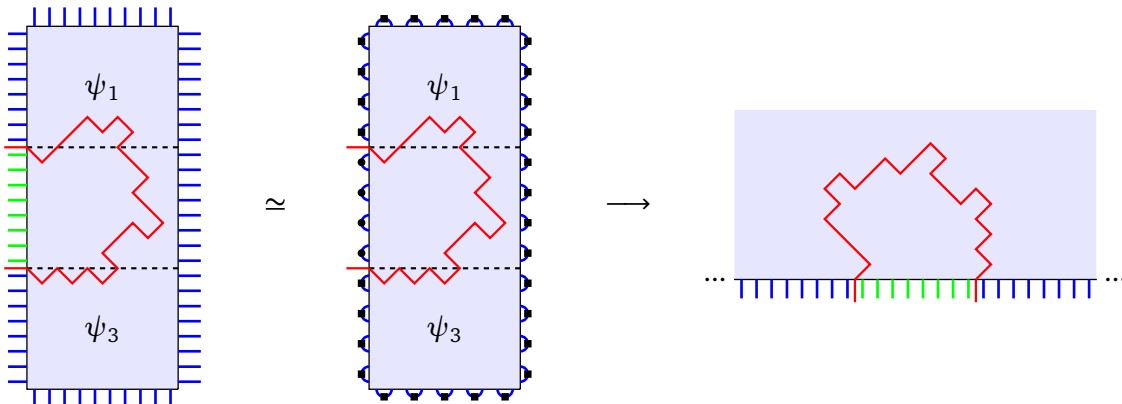} \tag{4.41}$$

and the corresponding bilinear form assigns to each type of loop the weights $\beta_1$, $\beta_2$ and $\beta_{12}^e$. The conformal character corresponding to the spectrum of $\boldsymbol{D}^{(2)}(u)$ in $\mathsf{V}_{n,0}^{(2)}$ is given in (9.24). For $r_{12} = \pi/\lambda$, the symmetries of the Kac formula allow us to write

$$\Delta_{r_{12}-2k, r_{12}} = \Delta_{2k-1,0}. \tag{4.42}$$

At the lowest order, we therefore have

$$C_{\mathbb{V}}^d(y_0, y_1) \xrightarrow{m \gg n} \tilde{K} e^{-\frac{\pi m}{n} \Delta_{r_{12}, r_{12}}} = \tilde{K} e^{-\frac{\pi m}{n} \Delta_{1,0}}. \tag{4.43}$$

The corresponding leading behaviour for the same correlator in $\mathbb{H}$ is

$$C_{\mathbb{H}}^d(z_0, z_1) \xrightarrow{m \gg n} \frac{K}{|z_0 - z_1|^{2\Delta_{1,0}}}. \tag{4.44}$$

The field that probes the connectivity of a cluster on a boundary with Neumann boundary conditions thus has conformal weight $\Delta_{1,0}$. For critical dense polymers, this conformal weight is

$$\Delta^d = \Delta_{1,0} = \frac{3}{8}, \tag{4.45}$$

consistent with the numerical data presented in Section 3.6.

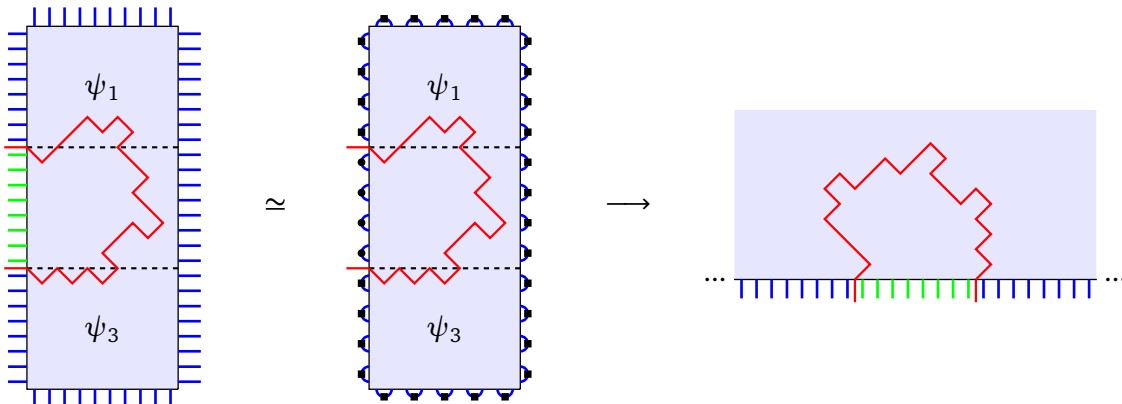

Figure 12: Boundary conditions on $\mathbb{V}$ and $\mathbb{H}$ corresponding to the correlators of type d. The red curve is one path connecting the two marked mid-points inside the corresponding cluster.

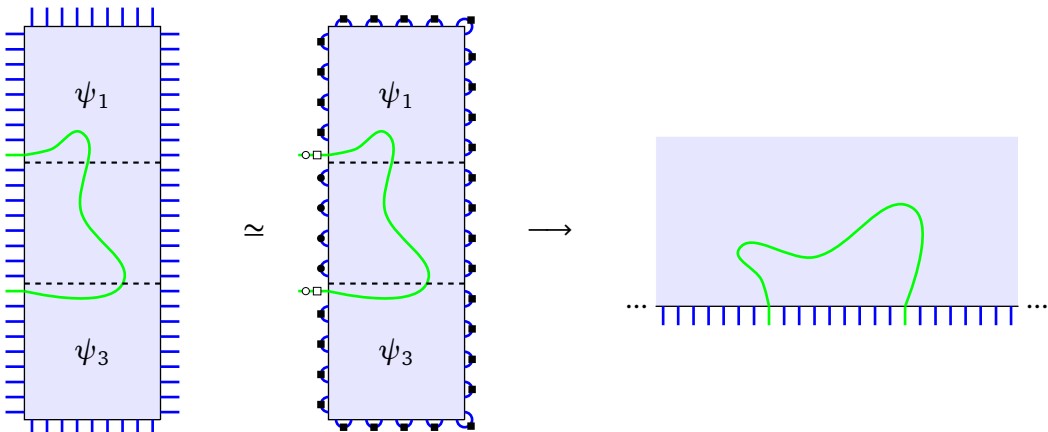

Figure 13: Boundary conditions on $\mathbb{V}$ and $\mathbb{H}$ corresponding to the correlators of type e.

The next leading orders in (4.43) are also exponentially decreasing in $m$, with the weights $\Delta_{2k-1,0}$, $k \in \mathbb{Z}$. For generic $\beta$, these are not integer-spaced. With the current technique, we cannot determine whether the constants multiplying these exponentials are zero or not. If they are not, one should conclude that the conformal field $\varphi^{\mathrm{d}}$ inserted to probe the connectivity of a cluster is not a primary field.

## 4.5 Type e: loop connectivities

For the correlators of type e, we decorate the domains $\mathbb{V}$ and $\mathbb{H}$ with blue and green defects, as in the first and last panel of Figure 13. There are only two green defects: those on the left segment that are just above and just below the region $R_2$, in $\mathbb{V}$. This colour code indicates that these two defects are constrainted to connect together. To perform the computation, we replace the blue defects by round and square blobs as in the second panel: round blobs on the left segment of $R_2$ and square blobs elsewhere. We impose $\beta_1 = \beta_2 = 1$, or equivalently

$$r_1 = r_2 = \frac{\pi - \lambda}{2\lambda}, \tag{4.46}$$

ensuring that blue defects can connect pairwise with weight 1. Only the two green defects are not transformed into arcs; $\psi_1$ and $\psi_3$ are therefore states with one defect. For these defects to connect together, they should avoid connecting to the blobs in the boundary. To impose this, we equip the green defects with two unblobs: a round one and a square one. In this setting, $\psi_1$ and $\psi_3$ are elements of $\mathsf{V}_{n,1}^{(2,\mathrm{u},\mathrm{u})}$, and we have:

$$C_{\mathbb{V}}^{\mathrm{e}}(y_0, y_1) = \frac{1}{Z^0} \, \psi_3 \cdot \left( (D^{(2)}(\tfrac{\lambda}{2}))^{m/2} \psi_1 \right)\Big|_{\mathsf{V}_{n,1}^{(2,\mathrm{u},\mathrm{u})}}. \tag{4.47}$$

The weight $\beta_{12}^{\mathrm{o}}$ does not appear in the sector $\mathsf{V}_{n,1}^{(2,\mathrm{u},\mathrm{u})}$, so we need not specify its value for the computation at hand. The conformal character corresponding to the spectrum of $D^{(2)}(u)$ in this sector is believed to be given by (9.23d). For $r_1$ and $r_2$ specified as in (4.46), the symmetries of the Kac formula yield

$$\Delta_{-r_1-r_2-1-2k,-r_1-r_2} = \Delta_{2k+1,-1}. \tag{4.48}$$

At the lowest order, the conformal part of the correlation functions is

$$C_{\mathbb{V}}^{\mathrm{e}}(y_0, y_1) \xrightarrow{m \gg n} \tilde{K} e^{-\frac{\pi m}{n}\Delta_{1,-1}}, \qquad C_{\mathbb{H}}^{\mathrm{e}}(z_0, z_1) \xrightarrow{m \gg n} \frac{K}{|z_0 - z_1|^{2\Delta_{1,-1}}}. \tag{4.49}$$

The field that probes the connectivitiy of a loop segment on a Neumann boundary has weight $\Delta_{1,-1}$. As mentioned in Section 2.6, the correlator of type e can be viewed as the correlator

between two adjacent clusters at $x$ and two others at $y$. The more general case of $\ell$ adjacent clusters will be discussed in Section 5. Remarkably,

$$\Delta^{\text{e}} = \Delta_{1,-1} = 1 \tag{4.50}$$

for all values of $\beta$. For critical dense polymers, this is consistent with the numerics presented in Section 3.6. Like for correlators of type d, the conformal dimensions of the next leading orders are not integer-spaced, implying that the field $\varphi^{\text{e}}$ that probes the connectivity of a loop in a Neumann boundary may not be primary.

In the geometric interpretation, we expect that the OPE of the fields $\varphi^{\text{a}}$ and $\varphi^{\text{d}}$ is of the form

$$\varphi^{\text{a}} \times \varphi^{\text{d}} = \varphi^{\text{e}} + \dots \, . \tag{4.51}$$

Indeed, fusing $\varphi^{\text{a}}$ and $\varphi^{\text{d}}$ corresponds to inserting an isolated defect near a starting cluster, which ends at a point far away to the right. This isolated defect can either connect to a point in its immediate neighbourhood, or to a point on the other side of the cluster:

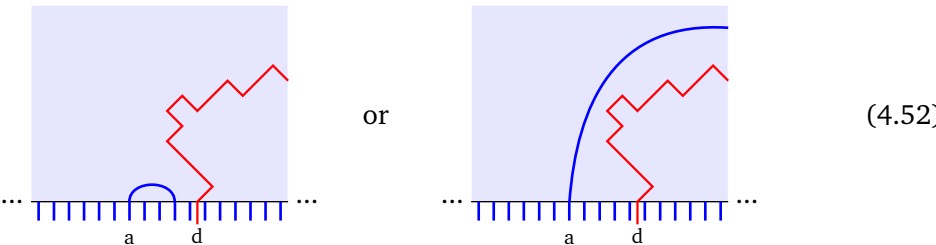

$$\tag{4.52}$$

We identify the latter with the case e, where a defect connects to a point far away to the right. This is consistent with the fusion rule

$$\varphi_{1,2} \times \varphi_{1,0} = \varphi_{1,1} + \varphi_{1,-1} \tag{4.53}$$

and the equality $\Delta^{\text{e}} = \Delta_{1,-1}$.

## 4.6 Type f: segment connectivities and valence bond entanglement entropy

For correlators of type f, we consider separately the cases where $y-x$ is even and odd. For $y-x$ even, we decorate the domains of $\mathbb{V}$ and $\mathbb{H}$ as in the first and last panels of Figure 12, with the red path removed and the green defects replaced by purple ones. To perform the computation, we replace pairs of adjacent purple defects by rounds blobs and set $r_1$ and $r_2$ to their values in (4.46), so that $\beta_1 = \beta_2 = 1$. A closed loop that contains both blobs is equivalent to two boundary loops that tie the region $R_2$ to the other two regions. We therefore have $\beta^{\text{e}}_{12} = \tau^2$ and, with the parametrisation (9.21a),

$$r_{12} = \frac{2\theta}{\lambda} + 1. \tag{4.54}$$

In this setting, we can write $C^{\text{f}}_{\mathbb{V}}(y_0, y_1)$ in terms of states $\psi_1$ and $\psi_3$ in $\mathsf{V}^{(2)}_{n,0}$:

$$C^{\text{f}}_{\mathbb{V}}(y_0, y_1) = \frac{1}{Z^0} \, \psi_3 \cdot \left( (\boldsymbol{D}^{(2)}(\tfrac{\lambda}{2}))^{m/2} \psi_1 \right)\big|_{\mathsf{V}^{(2)}_{n,0}}. \tag{4.55}$$

The conformal character in this case is (9.24). For $m \gg n$, at the lowest order, we have

$$C^{\text{f}}_{\mathbb{V}}(y_0, y_1) \xrightarrow{m \gg n} \tilde{K} e^{-\frac{\pi m}{n} \Delta_{r_{12}, r_{12}}}, \qquad C^{\text{f}}_{\mathbb{H}}(z_0, z_1) \xrightarrow{m \gg n} \frac{K}{|z_0 - z_1|^{2\Delta_{r_{12}, r_{12}}}}, \tag{4.56}$$

with $r_{12} = r_{12}(\tau)$ defined by (2.48) and (4.54). We thus find

$$\Delta^{\mathrm{f}}(\tau) = \Delta_{r_{12}(\tau),r_{12}(\tau)}. \tag{4.57}$$

This result is consistent with the special cases $\Delta^{\mathrm{f}}(0) = \Delta^{\mathrm{d}}$ and $\Delta^{\mathrm{f}}(1) = 0$. In the upper half-plane, the valence bond entanglement entropy is then given by

$$\langle n_{12} + n_{23} \rangle_{\mathbb{H}} = \frac{\mathrm{d}\theta}{\mathrm{d}\tau} \frac{\mathrm{d}r_{12}}{\mathrm{d}\theta} \frac{\mathrm{d}\Delta}{\mathrm{d}r_{12}} \frac{\mathrm{d}(\ln C_{\mathbb{H}}^{\mathrm{f}}(x,y))}{\mathrm{d}\Delta} \Big|_{\tau=1} = \frac{2(\lambda/\pi)}{\pi(1-\lambda/\pi)} \frac{\cos(\lambda/2)}{\sin(\lambda/2)} \ln(y-x), \quad (4.58)$$

which is identical to the result found in [43].

For $y - x$ odd, $\hat{Z}^{\mathrm{f}} = Z^{\mathrm{f}}/\tau$ is polynomial and non-zero at $\tau = 0$, and we define

$$\hat{C}_{\mathbb{V}}^{\mathrm{f}}(y_0, y_1) = \frac{\hat{Z}^{\mathrm{f}}}{Z^0}. \tag{4.59}$$

To give a prediction using CFT, we decorate the domains of $\mathbb{V}$ and $\mathbb{H}$ with purple and blue defects as in the first and last panels of Figure 14. The region $R_2$ contains $y - x - 1$ purple defects, and the last purple defect is in $R_1$. We proceed with the computation using the two-blob Temperley-Lieb algebra and replace the purple defects in $R_2$ with round blobs. Pairs of blue defects are replaced by square blobs, except for one which we choose for convenience to be the first defect on the left boundary of $R_3$. We attach a round blob and a black square blob to the remaining blue and purple defects. The states $\phi_1$ and $\phi_3$ are elements of $\mathsf{V}_{n,1}^{(2,\mathrm{b},\mathrm{b})}$. The fugacities of the loops touching the boundaries are set to $\beta_1 = \beta_2 = 1$ and $\beta_{12}^{\mathrm{o}} = \tau^2$ as before, assigning the correct weights to each configuration. This choice of the fugacities is satisfied for the values of $r_1$, $r_2$ and $r_{12}$ given in (4.46) and (4.54). Remarkably, the relation between $\theta$ and $r_{12}$ is the same as for the even case, even though (9.21a) and (9.21b) are different. We then have

$$\hat{C}_{\mathbb{V}}^{\mathrm{f}}(y_0, y_1) = \frac{1}{Z^0} \psi_3 \cdot \left( (\boldsymbol{D}^{(2)}(\tfrac{\lambda}{2}))^{m/2} \psi_1 \right) \Big|_{\mathsf{V}_{n,1}^{(2,\mathrm{b},\mathrm{b})}}, \tag{4.60}$$

where the bilinear form is adapted to produce the correct weights for each type of loop. For $m \gg n$, at the lowest order we have

$$C_{\mathbb{V}}^{\mathrm{f}}(y_0, y_1) \xrightarrow{m \gg n} \tilde{K} e^{-\frac{\pi m}{n}\Delta}, \qquad C_{\mathbb{H}}^{\mathrm{f}}(z_0, z_1) \xrightarrow{m \gg n} \frac{K}{|z_0 - z_1|^{2\Delta}}, \tag{4.61}$$

where $\Delta$ is the conformal weight of the ground state of the transfer matrix in $\mathsf{V}_{n,1}^{(2,\mathrm{b},\mathrm{b})}$. As mentioned at the end of Section 9, the character for this representation is unknown. We can however make an educated guess: We conjecture that the ground state in $\mathsf{V}_{n,1}^{(2,\mathrm{b},\mathrm{b})}$ has the conformal weight

$$\Delta = \Delta_{r_{12},r_{12}} = \frac{(t-1)^2\big((r_{12})^2 - 1\big)}{4t}, \tag{4.62}$$

as it does for $\mathsf{V}_{n,0}^{(2)}$. Here is the evidence that supports this claim. First, inspired by the other cases in Section 9, we expect that $\Delta$ is of the form $\Delta = \Delta_{r,s}$ where $r$ and $s$ are linear in $r_{12}$. This would imply that $\Delta$ is quadratic in $r_{12}$. We also know that $\Delta = 0$ for $r_{12} = \pm 1$, because in these cases $\tau = 1$ and there is no distinction between the three segments. Moreover, for $\beta_1 = \beta_2 = \beta$ and $\beta_{12}^{\mathrm{o}} = 1$, the ground-state conformal weight should coincide with the ground-state weight $\Delta_{1,2}$ for the standard module $\mathsf{V}_{n,1}$ of the ordinary Temperley-Lieb algebra, without blobs. For $r_1 = r_2 = 1$ and

$$r_{12} = \frac{\pi - 2\lambda}{\lambda} = -\frac{2t-1}{t-1}, \tag{4.63}$$

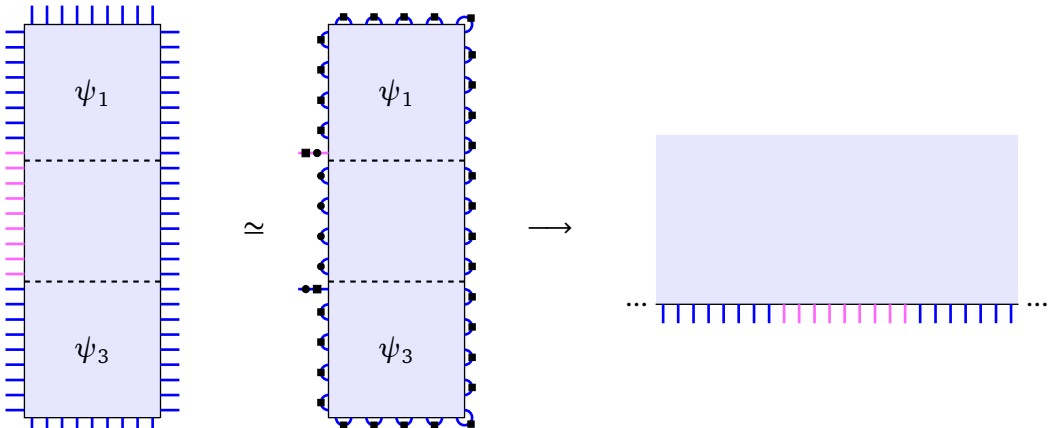

Figure 14: Boundary conditions on $\mathbb{V}$ and $\mathbb{H}$ corresponding to the correlators of type f, for $y - x$ odd.

the three fugacities are set to the desired values and we have

$$\Delta_{r_{12}, r_{12}} = \frac{3t}{4} - \frac{1}{2} = \Delta_{1,2} \tag{4.64}$$

as expected. Finally, with the conjecture (4.62), we find that in the scaling limit, the bond valence entanglement entropy is also given by (4.58) for $y - x$ odd. This is consistent with the results of [43], which is obtained in the scaling limit and does not distinguish between $y - x$ odd and even.

We note that the curve (4.62) appears to match the numerical data of Figure 6, except in the neighbourhood of $\tau = 0$ where there is a non-negligible deviation. We give two possible explanations. For the first, we note that the weights $\Delta_{r_{12}(\tau), r_{12}(\tau)}$ and $\Delta_{r_{12}(\tau)-2, r_{12}(\tau)}$, both of which have a contribution in (9.24), coincide at $\tau = 0$. In the neighbourhood of $\tau = 0$, the power-law behaviour of $C_{\mathbb{H}}^{\mathrm{f}}(z_0, z_1)$ may therefore be better approximated by a sum of two exponentials. This can serve to explain the deviations for both $y - x$ odd and even. For the second explanation, we remark that for $y - x$ odd, our computer program computes $C_n^{\mathrm{f}}(x, y)$ from (3.99), and not $\hat{C}_n^{\mathrm{f}}(x, y)$. The power law behaviour is thus multiplied by an overall factor of $\tau$ which may increase the error in the numerics for $\tau$ close to zero.

# 5 Conclusion

In this paper, we studied six types of boundary correlation functions of the dense loop model. For the model of critical dense polymers, we obtained exact expressions for these correlators and analysed their critical behaviour through exact calculations and numerical evaluations. Remarkably, in each case, the results are in agreement with the conformal predictions, wherein the two-point functions are interpreted as the expectation values of boundary changing conformal fields. For generic values of $\beta$, the dimensions of these six fields are $\Delta = \Delta_{1,2}$, $\Delta_{1,3}$, $\Delta_{0,1/2}$, $\Delta_{1,0}$, $\Delta_{1,-1}$ and $\Delta_{2\theta/\lambda+1, 2\theta/\lambda+1}$. For polymers, the field of dimension $\Delta_{1,3} = 0$ is a logarithmic field, partner of the primary field $\varphi_{1,1}$ in a rank two Jordan cell. Inserting this field in a corner produces a $\ln(\ln n)$ contribution to the corner free energy. For the correlation function of type f, the results also agree with the conformal prediction of Jacobsen and Saleur [30] for the valence bond entanglement entropy.

A key ingredient we used to produce the CFT predictions is the knowledge of the indecomposable structures of the representations, for the Temperley-Lieb algebra at finite size and for

the Virasoro algebra in the scaling limit. The presence of logarithmic corrections for the correlators is tied to the presence of Jordan cells in the representations [51]. The projective modules $\mathscr{P}_{n,d}$ over the Temperley-Lieb algebra are modules which exhibit this feature [33, 40, 52–54]. They consist of four composition factors organised in a diamond shaped diagram:

$$\mathscr{P}_{n,d}: \quad \begin{array}{ccc} & \mathscr{I}_{n,d} & \\ \swarrow & & \searrow \\ \mathscr{I}_{n,d'} & & \mathscr{I}_{n,d''} \\ \searrow & & \swarrow \\ & \mathscr{I}_{n,d} & \end{array}, \tag{5.1}$$

with the arrows indicating the action of the algebra. In these representations, the double-row transfer tangle and Hamiltonian have Jordan cells of rank two that tie the two $\mathscr{I}_{n,d}$ composition factors. For critical dense polymers, the $\ln|x - y|$ dependence for the correlator of type b is derived in Section 4.2 using conformal arguments where the Jordan cell plays a crucial role. This Jordan cell belongs to a projective module, as in (5.1), which has the particularity that its composition factor $\mathscr{I}_{n,d'}$ has dimension zero. Interestingly, the exact derivations in Section 3 for critical dense polymers do not highlight the role of these indecomposable structures. The results are instead derived from matrix elements that involve boundary states and only one eigenstate of the XX spin-chain. It is remarkable that the full complexity of the critical behaviour for the six types of correlators is encoded in this single eigenstate.

The methods we used have the potential to be extended to other two-point boundary correlation functions. First, one can consider generalisations of the correlators of type a and b wherein two collections of $d$ adjacent defects at positions $x$ and $y$ in a Dirichlet boundary are constrained to connect together. In the conformal setting, the calculation uses the standard module $V_{n,d}$ and yields a power-law behaviour with $\Delta = \Delta_{1,d+1}$. Second, one can also define correlators wherein two collections of $\ell$ adjacent clusters at $x$ and $y$ in a Neumann boundary are constrained to connect to one another. In the Fortuin-Kasteleyn random cluster model, every second cluster in these collections lives on the dual lattice. The correlators of type d and e then correspond to $\ell = 1, 2$. To extend the CFT arguments of Section 4 to the case $\ell \geqslant 2$, one uses the standard module $V_{n,\ell-1}^{(2,\mathrm{u},\mathrm{u})}$. The leading power-law has the conformal dimension

$$\Delta = \min(\Delta_{2k+1,1-\ell}|k \in \mathbb{Z}_{\geqslant 0}) = \Delta_{1,1-\ell}. \tag{5.2}$$

There thus seems to be a duality between the Dirichlet and Neumann boundary conditions: The field that corresponds to the insertion of $d$ defects in the Dirichlet boundary has weight $\Delta_{1,1+d}$, and the field that probes the connectivity of $\ell$ clusters in the Neumann boundary has weight $\Delta_{1,1-\ell}$.

In Section 4, we found three examples where the interpretation of the lattice results in terms of boundary changing fields was consistent at the level of the fusion of these fields. From the discussion of the last paragraph, in the geometric interpretation, inserting a field of type d near a collection of $\ell$ clusters should yield the following fusion rule:

$$\varphi_{1,0} \times \varphi_{1,1-\ell} = \varphi_{1,-\ell} + \varphi_{1,2-\ell}. \tag{5.3}$$

More generally, one could expect that fusing $\varphi_{1,0}$ and $\varphi_{r,s}$ produces two fields, $\varphi_{r,s-1}$ and $\varphi_{r,s+1}$. This would be consistent with the duality $\varphi_{1,0} \longleftrightarrow \varphi_{1,2}$.

However, naively setting $\ell = -1$ in (5.3) leads to an inconsistency with (4.53), since $\varphi_{1,-1}$ and $\varphi_{1,3}$ are certainly different in general. Therefore, our working hypotheses, that fusing $\varphi_{r,s}$ with $\varphi_{1,2}$ changes $s$ by $\pm 1$, and likewise for the fusion with $\varphi_{1,0}$, give at best incomplete results when the field $\varphi_{r,s}$ lies outside its natural domain of the Kac table. In other words, we expect that the various fusion rules discussed in this paper contain more channels, and maybe even

an infinite number of channels, in addition to those written explicitly. Establishing definitive fusion rules for non-Kac fields related to geometrical observables in loop models remains a challenge for future research. For some recent, partial progress in the context of the affine Temperley-Lieb algebra, see [55].

Finally, it will be interesting to see whether the techniques developed here can be extended to compute correlators for points in the bulk, and on lattices with periodic boundary conditions. On the cylinder, the transfer tangle has Jordan cells of rank $\rho \geqslant 2$ in certain representations [56, 57]. This is expected to result in correlation functions with $(\ln|x - y|)^{\rho-1}$ corrections. We expect that such an investigation will shed further light on the general fusion rules for the conformal fields $\varphi_{r,s}$.

### Acknowledgments

AMD is supported by the FNRS fellowship CR28075116. JLJ is supported by the ERC advanced grant *Non-unitary quantum field theory* (PI: Hubert Saleur). AMD and JLJ acknowledge the hospitality and support of the Matrix Institute and the Australian Mathematical Sciences Institute during the conference *Integrability in Low-Dimensional Quantum Systems* in Creswick where part of this work was done. The authors thank Philippe Ruelle and Romain Couvreur for useful discussions.

# 6 The function $\tilde{f}_{k,\ell}$

In this section, we study the function $\tilde{f}_{k,\ell}$ defined in (3.56b) and its $n \to \infty$ limit, in the range $\frac{x-1}{2} \leqslant k \leqslant \frac{y}{2}$ and $x \leqslant \ell \leqslant y - 1$. We first consider $\ell$ even. We use the identity

$$\frac{\sin(\frac{\pi \ell j}{n})}{\cos(\frac{\pi j}{n})} = 2 \sum_{t=0}^{\ell/2-1} (-1)^t \sin\left(\frac{\pi j}{n}(\ell - 1 - 2t)\right), \tag{6.1}$$

which is only valid for even $\ell$. Applying this to (3.56b), we interchange the order of the sums, perform the sum over $j$ and find

$$\begin{aligned}
\tilde{f}_{k,\ell \text{ even}} &= \sum_{t=0}^{\ell/2-1} (-1)^t \left(\delta_{\ell-1-2t,2k+1} + (-1)^{k-(x+1)/2} \delta_{\ell-1-2t,x}\right) \\
&= (-1)^{k-\ell/2}(\delta_{\ell-2 \geqslant x-1 \geqslant 0} - \delta_{\ell-2 \geqslant 2k \geqslant 0}).
\end{aligned} \tag{6.2}$$

For $\ell$ odd, we have

$$\tilde{f}_{k,\ell \text{ odd}} = \begin{cases} \dfrac{4}{n} \displaystyle\sum_{j=1}^{(n-2)/2} \dfrac{\sin\left(\frac{\pi j}{n}(k + \frac{1+x}{2})\right)}{\cos(\frac{\pi j}{n})} \cos\left(\frac{\pi j}{n}(k + \frac{1-x}{2})\right) \sin(\frac{\pi j \ell}{n}) & k - \frac{x+1}{2} \text{ even}, \\[4ex] \dfrac{4}{n} \displaystyle\sum_{j=1}^{(n-2)/2} \dfrac{\sin\left(\frac{\pi j}{n}(k + \frac{1-x}{2})\right)}{\cos(\frac{\pi j}{n})} \cos\left(\frac{\pi j}{n}(k + \frac{1+x}{2})\right) \sin(\frac{\pi j \ell}{n}) & k - \frac{x+1}{2} \text{ odd}. \end{cases} \tag{6.3}$$

In each case, we use (6.1) on the sin-cos ratios, interchange the order of the sums and perform the sum over $j$. After simplification, the result, which holds for both parities of $k - \frac{x+1}{2}$, is

$$\tilde{f}_{k,\ell \text{ odd}} = \frac{(-1)^{k+(\ell+1)/2}}{n} \sum_{t=(1+x)/2}^{k} \frac{\sin(\frac{2\pi t}{n})}{\sin\left(\frac{\pi}{n}(t + \frac{\ell}{2})\right) \sin\left(\frac{\pi}{n}(t - \frac{\ell}{2})\right)}. \tag{6.4}$$

In this form, it is easy to take the limit $n \to \infty$:

$$\lim_{n\to\infty} \tilde{f}_{k,\ell \text{ odd}} = \frac{(-1)^{k+(\ell+1)/2}}{\pi/2} \sum_{t=(1+x)/2}^{k} \frac{t}{(t+\frac{\ell}{2})(t-\frac{\ell}{2})}. \tag{6.5}$$

To obtain correlation function in the upper half-plane, we are interested in the regime $1 \ll y - x \ll x, y$. In this regime, $\ell - x$ and $k - \frac{1+x}{2}$ remain small compared to $x$ and $y$, and we find

$$\lim_{n\to\infty} \tilde{f}_{k,\ell \text{ odd}} \xrightarrow{x,y \gg y-x \gg 1} \frac{(-1)^{k+(\ell+1)/2}}{\pi} \sum_{t=0}^{k-(1+x)/2} \frac{1}{t-(\ell-x-1)/2}. \tag{6.6}$$

# 7 More results for correlators of type b

It is not hard to generalise the lattice result (3.70) to compute $C_n^b(x, n-x)$. We find

$$C_n^b(x,n-x) = 1 + (-1)^{n/2}\tilde{f}_x + \tilde{f}_{n-x}, \quad \tilde{f}_x = (-1)^{n/2}\tilde{f}_{n-x} = \frac{4}{n}(-1)^{n/2} \sum_{\substack{j=1 \\ j \equiv \frac{n-2}{2} \bmod 2}}^{(n-2)/2} \frac{\sin^2\left(\frac{\pi j x}{n}\right)}{\cos\left(\frac{\pi j}{n}\right)}. \tag{7.1}$$

Large $n$ asymptotics for $\tilde{f}_x$ and $C_n^b(x, n-x)$ can be performed in two ways. The first way is to take $n \gg 1$ while keeping $x$ finite. We apply the Euler-Maclaurin formula to (7.1) and find:

$$C_n^b(x,n-x) = \frac{4}{\pi}\ln n + 1 + \frac{4}{\pi}(\gamma + 2\ln 2 - \ln \pi) - \frac{2}{\pi}\sum_{k=0}^{x-1}\frac{1}{k+\frac{1}{2}} + O(n^{-1}). \tag{7.2}$$

This is consistent with (3.70) for $x = 1$. We obtain the critical behaviour in the regime $x \gg 1$ by using (3.64):

$$C_n^b(x,n-x) \xrightarrow{1 \ll x \ll n} \frac{4}{\pi}\ln n - \frac{2}{\pi}\ln x + 1 + \frac{2}{\pi}(\gamma + 2\ln 2 - 2\ln \pi). \tag{7.3}$$

This critical behaviour is described in terms of the distance $x$ between the points to the corners, and the prefactors in front of $\ln n$ and $\ln x$ are different.

The second way is to take $n, x \gg 1$ with the ratio $x/n$ fixed. Starting from (7.1) and (6.4), this is again achieved using the Euler-Maclaurin formula. We find:

$$C_n^b(x,n-x) \xrightarrow{n,x \gg 1} 1 + \frac{2}{\pi}(\ln n + \ln \cot(\tfrac{\pi x}{n}) - \ln \pi + 2\ln 2 + \gamma). \tag{7.4}$$

From this formula, one recovers (3.66) in the regime $x/n \simeq 1/2$, and (7.3) in the regime $x/n \simeq 0$. The resulting critical behaviour is thus independent of the order in which the limits are taken. In Figure 15, we have plotted the exact values of $C_n^b(x, n-x)$ for $n = 500$, along with three theoretical curves: i) (7.3) in orange, ii) (3.66) in red, and iii) (7.4) in blue. The crossover between the two regimes is smooth.

We also remark that the critical behaviour of $C_n^b(x, y)$ in the regime $x \ll y \ll n$ can be read off from (3.63) and (3.64):

$$C_n^b(x,y) \xrightarrow{x \ll y \ll n} \frac{3}{\pi}\ln y + 1 + \frac{1}{\pi}\left(3\gamma + 2\ln 2 - \sum_{k=0}^{x-1}\frac{1}{k+\frac{1}{2}}\right). \tag{7.5}$$

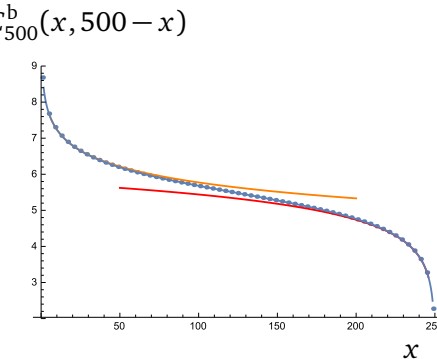

Figure 15: Values of $C_n^b(x, n-x)$ obtained from the exact formula (7.1), for $n = 500$.

Comparing this with (3.66) and (7.3), we see that the leading logarithmic term has the prefactor $(2+\iota)/\pi$, where $\iota \in \{0, 1, 2\}$ counts the number of points ($x$ and/or $y$) near a corner.

We end this section by noting that (7.4) can be obtained using a CFT argument. The map from the upper half $z$-plane $\mathbb{H}$ to the semi-infinite strip $\mathbb{U}$, with coordinates $y$ and width $n$, is $z = \sin^2(\frac{\pi y}{2n})$. From (4.22) and (4.23), we find

$$
\begin{aligned}
C_{\mathbb{U}}^b(y_0, y_1) &= \frac{\langle \omega(y_0)\omega(y_1)\rangle_{\mathbb{U}}}{\langle \omega(y_0)\rangle_{\mathbb{U}}} \\
&= -4\lambda_0 \ln \left| \sin^2(\tfrac{\pi y_0}{2n}) - \sin^2(\tfrac{\pi y_1}{2n}) \right| + 2\lambda_0 \ln \left| (\tfrac{\pi}{2n})^2 \sin(\tfrac{\pi y_0}{n}) \sin(\tfrac{\pi y_1}{n}) \right| + \lambda_2
\end{aligned}
\tag{7.6}
$$

and therefore

$$
C_{\mathbb{U}}^b(x, n-x) = -4\lambda_0 \ln n - 4\lambda_0 \ln \cot(\tfrac{\pi x}{n}) - 4\lambda_0(\ln 2 - \ln \pi) + \lambda_2.
\tag{7.7}
$$

With the values of $\lambda_0$ and $\lambda_2$ given in (4.27), this precisely reproduces (7.4). The coefficient in front of $\ln n$ in (3.70) is $-8\lambda_0$ and is therefore universal.

# 8 The asymptotic expansion of $C_n^c(1, n)$

We discuss the $\frac{1}{n}$ asymptotic expansions for $\langle v_0'|w_0\rangle$ and $\langle v^c|w_0\rangle$ leading to (3.86). For $\langle v_0'|w_0\rangle$, this expansion is obtained from (3.23) and (3.25), by applying the identity

$$
\prod_{k=1}^{(n-2)/2} \cos(\tfrac{\pi k}{n}) = \frac{\sqrt{n}}{2^{(n-1)/2}}.
\tag{8.1}
$$

This yields the exact expression

$$
\ln\left((-1)^{(n-2)(n-4)/8} \omega \langle v_0'|w_0\rangle \prod_{k=1}^{(n-2)/2} \kappa_k\right) = \tfrac{1}{4}n \ln n - \tfrac{1}{2}n \ln 2 - \tfrac{1}{2}\ln n + 2\ln 2.
\tag{8.2}
$$

For $\langle v^c|w_0\rangle$ and $n/2$ even, the asymptotic expansion is obtained separately for each product in (3.82):

$$
\ln\left(\prod_{k=1}^{n/4} \cos\left(\tfrac{\pi}{n}(k-\tfrac{1}{2})\right)\right) = n(\tfrac{G}{2\pi} - \tfrac{1}{4}\ln 2) + O(n^{-1}),
\tag{8.3a}
$$

$$
\ln\left(\prod_{k=1}^{n/4} \cos\left(\tfrac{2\pi}{n}(k-\tfrac{1}{2})\right)\right) = -\tfrac{n}{4}\ln 2 + \tfrac{1}{2}\ln 2,
\tag{8.3b}
$$

$$\ln\Big(\prod_{k=1}^{n/4}\sin\big(\tfrac{2\pi k}{n}\big)\Big) = -\tfrac{n}{4}\ln 2 + \tfrac{1}{2}\ln n, \tag{8.3c}$$

$$\ln\Big(\prod_{k=1}^{n/4}\sin\big(\tfrac{\pi}{n}(k-\tfrac{1}{2})\big)\Big) = n(-\tfrac{G}{2\pi} - \tfrac{1}{4}\ln 2) + \tfrac{1}{2}\ln 2, \tag{8.3d}$$

$$\ln\Big(\prod_{1\leqslant k<\ell\leqslant n/4}\sin\big(\tfrac{\pi}{n}(k+\ell-1)\big)\Big) = \tfrac{n^2}{64}(32\,\mathscr{I}_+ + 2\ln\pi - 3) + \tfrac{n}{8}\ln 2 + \tfrac{1}{24}\ln n$$
$$+ \tfrac{1}{24}(12\ln A - 7\ln 2 - \ln\pi - 1) + O(n^{-1}), \tag{8.3e}$$

$$\ln\Big(\prod_{1\leqslant k<\ell\leqslant n/4}\sin\big(\tfrac{\pi}{n}(k+\ell)\big)\Big) = \tfrac{n^2}{64}(32\,\mathscr{I}_+ + 2\ln\pi - 3) + n(\tfrac{G}{2\pi} + \tfrac{1}{8}\ln 2) - \tfrac{11}{24}\ln n$$
$$+ \tfrac{1}{24}(12\ln A - \ln 2 - \ln\pi - 1) + O(n^{-1}), \tag{8.3f}$$

$$\ln\Big(\prod_{1\leqslant k<\ell\leqslant n/4}\sin\big(\tfrac{\pi}{n}(k-\ell)\big)\Big) = \tfrac{n^2}{64}(32\,\mathscr{I}_- - 4\ln 2 + 2\ln\pi - 3) + \tfrac{n}{8}\ln n + \tfrac{n}{8}\ln 2 - \tfrac{1}{12}\ln n$$
$$+ \tfrac{1}{24}(-24\ln A + 2\ln\pi + \ln 2 + 2) + O(n^{-1}), \tag{8.3g}$$

$$\ln\Big(\prod_{k,\ell=1}^{n/4}\sin\big(\tfrac{\pi}{n}(k+\ell-\tfrac{1}{2})\big)\Big) = n^2(\mathscr{I}_+ + \tfrac{1}{16}\ln\pi - \tfrac{3}{32}) + n\tfrac{G}{2\pi} - \tfrac{1}{24}\ln n$$
$$+ \tfrac{1}{24}(-12\ln A + \ln\pi - 4\ln 2 + 1) + O(n^{-1}), \tag{8.3h}$$

$$\ln\Big((-1)^{\frac{n(n-4)}{32}}\prod_{k,\ell=1}^{n/4}\sin\big(\tfrac{\pi}{n}(k+\ell-\tfrac{1}{2})\big)\Big) = n^2(\mathscr{I}_- + \tfrac{1}{16}\ln\pi - \tfrac{3}{32} - \tfrac{1}{8}\ln 2) + \tfrac{n}{4}\ln 2 + \tfrac{1}{12}\ln n$$
$$+ \tfrac{1}{24}(24\ln A - 2\ln\pi - 3\ln 2 - 2) + O(n^{-1}), \tag{8.3i}$$

where

$$\mathscr{I}^{\pm} = \int_0^{\frac{1}{4}}\int_0^{\frac{1}{4}} \mathrm{d}y\,\mathrm{d}z\,\ln s[\pi(y\pm z)], \qquad s[y] = \frac{\sin y}{y}. \tag{8.4}$$

These $\frac{1}{n}$ expansions are obtained as follows. Let us define

$$X(a,b,x) = \sum_{k=0}^{nx}\ln s[\tfrac{\pi a}{n}(k+b)], \qquad Y_{\pm}(b,x) = \sum_{k,\ell=0}^{nx}\ln s[\tfrac{\pi}{n}(k\pm\ell+b)], \tag{8.5}$$

where $x \in [0,1/a)$ for $X(a,b,x)$ and $x \in [0,1)$ for $Y_{\pm}(b,x)$. Using the Euler-Maclaurin formula, we find the following $\frac{1}{n}$ expansions:

$$X(a,b,x) = n\int_0^x \mathrm{d}z\,\ln s[\pi a z] + (b+\tfrac{1}{2})\ln s[\pi a x] + O(n^{-1}), \tag{8.6a}$$

$$Y_+(b,x) = n^2\int_0^x\int_0^x \mathrm{d}y\,\mathrm{d}z\,\ln s[\pi(y+z)]$$
$$+ n\Big((1+b)\int_0^x \mathrm{d}z\,\ln s[\pi(x+z)] + (1-b)\int_0^x \mathrm{d}z\,\ln s[\pi z]\Big) \tag{8.6b}$$
$$+ \Big((\tfrac{1}{2}b^2 + b + \tfrac{5}{12})\ln s[2\pi x] - (b^2 - \tfrac{1}{6})\ln s[\pi x]\Big) + O(n^{-1}),$$

$$Y_-(b,x) = n^2 \int_0^x \int_0^x dy\, dz \ln s[\pi(y-z)]$$
$$+ 2n \int_0^x dz \ln s[\pi(z)] + (b^2 + \tfrac{5}{6}) \ln s[\pi x] + O(n^{-1}). \qquad (8.6c)$$

To show (8.3a), we have

$$\prod_{k=1}^{n/4} \cos(\tfrac{\pi}{n}(k-\tfrac{1}{2})) = e^{X(2,-\frac{1}{2},\frac{1}{4})-X(1,-\frac{1}{2},\frac{1}{4})} \frac{s[\frac{\pi}{n}]}{s[\frac{\pi}{2n}]} \qquad (8.7)$$

and the asymptotic expansion is read off from (8.6a). The relations (8.3b) and (8.3c) are exact identities:

$$\prod_{k=1}^{n/4} \cos\left(\tfrac{2\pi}{n}(k-\tfrac{1}{2})\right) = \frac{\sqrt{2}}{2^{n/4}}, \qquad \prod_{k=1}^{n/4} \sin(\tfrac{2\pi k}{n}) = \frac{\sqrt{n}}{2^{n/4}}. \qquad (8.8)$$

To show (8.3d), we have

$$\prod_{k=1}^{n/4} \sin\left(\tfrac{\pi}{n}(k-\tfrac{1}{2})\right) = \frac{(\tfrac{\pi}{n})^{n/4+1}}{\sin(\tfrac{\pi}{4}+\tfrac{\pi}{2n})} \prod_{k=0}^{n/4}(k+\tfrac{1}{2})s[\tfrac{\pi}{n}(k+\tfrac{1}{2})] = \frac{(\tfrac{\pi}{n})^{n/4+1}}{\sin(\tfrac{\pi}{4}+\tfrac{\pi}{2n})} \frac{\Gamma(\tfrac{n}{4}+\tfrac{3}{2})}{\Gamma(\tfrac{1}{2})} e^{X(1,\frac{1}{2},\frac{1}{4})} \quad (8.9)$$

and the asymptotic expansion is obtained from (8.6a) and the asymptotic expansion of $\ln\Gamma(z)$:

$$\ln\Gamma(z) = z\ln z - z - \frac{1}{2}\ln z + \frac{1}{2}\ln(2\pi) + O(z^{-1}). \qquad (8.10)$$

The other relations (8.3e)–(8.3i) involve double products. The calculation is more tedious, but the strategy is similar: We rewrite the products in terms of the functions $\Gamma(z)$, $G(z)$, $X(a,b,x)$ and $Y_\pm(b,x)$, and use the asymptotic expansions (3.41), (8.6) and (8.10). Let us give one example, for (8.3e):

$$\prod_{1\leqslant k<\ell\leqslant n/4} \sin\left(\tfrac{\pi}{n}(k+\ell-1)\right) = (\tfrac{\pi}{n})^{n(n-4)/32} \prod_{1\leqslant k<\ell\leqslant n/4} (k+\ell-1)s[\tfrac{\pi}{n}(k+\ell-1)]. \qquad (8.11)$$

We rewrite this using

$$\prod_{1\leqslant k<\ell\leqslant n/4} (k+\ell-1) = \prod_{k=1}^{n/4}\prod_{\ell=k+1}^{n/4} \frac{\Gamma(k+\ell)}{\Gamma(k+\ell-1)} = \prod_{k=1}^{n/4} \frac{\Gamma(k+\tfrac{n}{4})}{\Gamma(2k)}$$
$$= \frac{\pi^{n/8}}{2^{n^2/16}} \prod_{k=1}^{n/4} \frac{G(k+\tfrac{n}{4}+1)G(k)G(k+\tfrac{1}{2})}{G(k+\tfrac{n}{4})G(i+1)G(i+\tfrac{3}{2})} \qquad (8.12a)$$
$$= \frac{\pi^{n/8}}{2^{n^2/16}} \frac{G(\tfrac{n}{2}+1)G(1)G(\tfrac{3}{2})}{G(\tfrac{n}{4}+1)^2 G(\tfrac{n}{4}+\tfrac{3}{2})},$$

$$\prod_{1\leqslant k<\ell\leqslant n/4} s[\tfrac{\pi}{n}(k+\ell-1)] = \frac{s[\tfrac{\pi}{n}]\prod_{k,\ell=0}^{n/4} s[\tfrac{\pi}{n}(k+\ell-1)]^{1/2}}{\prod_{k=0}^{n/4} s[\tfrac{\pi}{n}(k-1)]s[\tfrac{2\pi}{n}(k-\tfrac{1}{2})]^{1/2}} \qquad (8.12b)$$
$$= s[\tfrac{\pi}{n}]e^{\frac{1}{2}Y_+(-1,\frac{1}{4})-X(1,-1,\frac{1}{4})-\frac{1}{2}X(2,-\frac{1}{2},\frac{1}{4})},$$

where for (8.12a), we used the duplication formula for $\Gamma(z)$:

$$\Gamma(2z) = \frac{2^{2z-1}}{\sqrt{\pi}}\Gamma(z)\Gamma(z+\tfrac{1}{2}). \qquad (8.13)$$

Putting the relations (8.3) together, we find, for $n/2$ even:

$$\ln\left((-1)^{(n-4)/4}\omega\,\langle v^{c}|w_{0}\rangle\prod_{k=1}^{(n-2)/2}\kappa_{k}\right)=\tfrac{1}{4}n\ln n+n(\tfrac{G}{\pi}-\tfrac{1}{4}\ln 2)-\tfrac{1}{8}\ln n$$
$$+\tfrac{1}{24}(-36\ln A+3\ln\pi+\ln 2+3). \tag{8.14}$$

Combining this with (8.2) yields (3.86). Repeating the calculation with $n/2$ odd, we find the same right-hand side as (8.14), but with $(-1)^{(n-4)/4}$ replaced by $(-1)^{(n-2)/4}$ on the left-hand side.

## 9 Conformal characters and blob algebras

In this section, we collect results and conjectures about the spectra of the double-row transfer matrices of the loop model, with the boundary conditions set to simple arcs or arcs with blobs. The logarithm of the leading eigenvalues $D(u)$ of the transfer matrices have $\frac{1}{n}$ expansions of the following form [20, 21]:

$$\ln D(u)=-2nf_{\rm b}(u)-f_{\rm s}(u)-\tfrac{2\pi}{n}\sin(\tfrac{\pi u}{\lambda})\bigl(\Delta-\tfrac{c}{24}\bigr)+o(n^{-1}), \tag{9.1}$$

where $\Delta$ is the weight of the underlying conformal field. The conformal character is then given by

$$e^{2mnf_{\rm b}(u)+mf_{\rm s}(u)}{\rm Tr}\bigl(D^{m}(u)\bigr)\xrightarrow{m,n\to\infty}Z(q)=\sum_{\rm eigenstates}q^{\Delta-c/24}, \tag{9.2}$$

where the ratio $m/n$ is taken to converge to a constant in the scaling limit and

$$q=\exp(-\tfrac{2\pi m}{n}\sin(\tfrac{\pi u}{\lambda})). \tag{9.3}$$

For $\mathsf{TL}_{n}(\beta)$, the transfer matrices are labeled by the number of defects $d$ of the standard modules $\mathsf{V}_{n,d}$ they are defined on. We denote the corresponding characters $Z_{d}^{(0)}(q)$. These were obtained in [9]:

$$Z_{d}^{(0)}(q)=\frac{q^{\Delta_{1,1+d}-c/24}}{(q)_{\infty}}(1-q^{1+d}),\qquad(q)_{\infty}=\prod_{k=1}^{\infty}(1-q^{k}). \tag{9.4}$$

For the one-boundary case, the corresponding loop model is described by the one-boundary Temperley-Lieb algebra [27–29], or equivalently by the blob algebra. The blob algebra is an extension of $\mathsf{TL}_{n}(\beta)$, with an extra generator $b_{1}$ in the form of a blob attached to the leftmost strand:

$$b_{1}=\vcenter{\hbox{\includegraphics{b1.png}}}. \tag{9.5}$$

The defining relations are (2.9) and

$$(b_{1})^{2}=b_{1},\qquad e_{1}b_{1}e_{1}=\beta_{1}e_{1},\qquad b_{1}e_{i}=e_{i}b_{1},\quad 2\leqslant i\leqslant n. \tag{9.6}$$

These last relations are equivalently expressed in diagrams as:

$$\vcenter{\hbox{\includegraphics{rel1.png}}}=\vcenter{\hbox{\includegraphics{rel2.png}}},\qquad\vcenter{\hbox{\includegraphics{rel3.png}}}=\beta_{1}\vcenter{\hbox{\includegraphics{rel4.png}}},\qquad\vcenter{\hbox{\includegraphics{rel5.png}}}=\vcenter{\hbox{\includegraphics{rel6.png}}}. \tag{9.7}$$

The parameter $\beta_1$ is the fugacity of the loops that contain a blob.

In the XXZ spin-chain, the generator $b_1$ is represented by [27]:

$$\mathsf{X}_n(b_1) = \frac{1}{2\mathrm{i}\sin(r_1\lambda)}\begin{pmatrix} -e^{-\mathrm{i}\lambda r_1} & \mathrm{i} \\ \mathrm{i} & e^{\mathrm{i}\lambda r_1} \end{pmatrix} \otimes \underbrace{\mathbb{I}_2 \otimes \cdots \otimes \mathbb{I}_2}_{n-1}. \tag{9.8}$$

The matrices (2.16) and (9.8) satisfy the relations (2.9) and (9.6), with $\beta_1$ parameterised in terms of $r_1$ as

$$\beta_1 = \frac{\sin\big((r_1+1)\lambda\big)}{\sin(r_1\lambda)}. \tag{9.9}$$

The standard representations $\mathsf{V}_{n,d}^{(1,\mathrm{b})}$ and $\mathsf{V}_{n,d}^{(1,\mathrm{u})}$ of the blob algebra are labeled by the number of defects $d$ and a letter u or b according to whether the leftmost defect is allowed or forbidden to touch the boundary. In the corresponding link states, the leftmost defect is decorated by a blob $b_1$ or by an *unblob* $u_1 = 1 - b_1$. As an element of the algebra, we draw $u_1$ as

$$u_1 = \begin{array}{c} \boxed{\phantom{x}} \\ {\scriptstyle 1\ 2\ 3} \quad {\scriptstyle n} \end{array}. \tag{9.10}$$

In a given link state, an arc to the left of the leftmost defect, if it is not overarched by a larger arc, is decorated by either a blob or an unblob. If there are no defects, there is no distinction between the blob and unblob sectors, and each arc that is not overarched is decorated with a blob or an unblob. For instance, the link states for $n = 4$ are

$$\begin{array}{ll} \mathsf{V}_{4,4}^{(1,\mathrm{b})}: & \text{[diagram]}, \\[4pt] \mathsf{V}_{4,4}^{(1,\mathrm{u})}: & \text{[diagram]}, \\[4pt] \mathsf{V}_{4,2}^{(1,\mathrm{b})}: & \text{[diagrams]}, \\[4pt] \mathsf{V}_{4,2}^{(1,\mathrm{u})}: & \text{[diagrams]}, \\[4pt] \mathsf{V}_{4,0}^{(1)}: & \text{[diagrams]}. \end{array} \tag{9.11}$$

We denote the corresponding conformal characters by $Z_d^{(1,\mathrm{b})}(q)$, $Z_d^{(1,\mathrm{u})}(q)$ and $Z_0^{(1)}(q)$. Their expressions were conjectured in [30] using numerical data and support from exact results for the root-of-unity cases. The results are expressed in terms of the parameter $r_1$:

$$Z_d^{(1,\mathrm{b})}(q) = \frac{q^{\Delta_{r_1,r_1+d}-c/24}}{(q)_\infty}, \qquad Z_d^{(1,\mathrm{u})}(q) = \frac{q^{\Delta_{r_1,r_1-d}-c/24}}{(q)_\infty}, \qquad Z_0^{(1)}(q) = \frac{q^{\Delta_{r_1,r_1}-c/24}}{(q)_\infty}. \tag{9.12}$$

For the two-boundary case, the relevant algebra is the two-boundary Temperley-Lieb algebra [32, 58]. In its blob formulation, there are two blob generators $b_1$ and $b_2$, one for each boundary. The defining relations are (2.9), (9.6) and

$$(b_2)^2 = b_2, \quad e_{n-1} b_2 e_{n-1} = \beta_2 e_{n-1}, \quad b_1 b_2 = b_2 b_1, \quad b_2 e_i = e_i b_2, \quad 1 \leqslant i \leqslant n-2. \tag{9.13}$$

We draw the blob and unblob of the right boundary as black and white squares:

$$b_2 = \begin{array}{c} \boxed{\phantom{x}} \\ {\scriptstyle 1\ 2} \quad {\scriptstyle n} \end{array}, \qquad u_2 = 1 - b_2 = \begin{array}{c} \boxed{\phantom{x}} \\ {\scriptstyle 1\ 2} \quad {\scriptstyle n} \end{array}. \tag{9.14}$$

For $n$ even, there is an extra algebraic relation which quotients out the closed loops containing both blobs:

$$\Big(\prod_{i=1}^{n/2} e_{2i-1}\Big) b_1 b_2 \Big(\prod_{i=1}^{(n-2)/2} e_{2i}\Big)\Big(\prod_{i=1}^{n/2} e_{2i-1}\Big) = \beta_{12}^{\mathrm{e}} \Big(\prod_{i=1}^{(n-2)/2} e_{2i}\Big). \tag{9.15}$$

It is depicted as

$$
\text{(diagram)} = \beta_{12}^{\text{e}} \ \text{(diagram)} \, .
\tag{9.16}
$$

For $n$ odd, there are two quotient relations:

$$
\Big( \prod_{i=1}^{(n-1)/2} e_{2i} \Big) b_1 \Big( \prod_{i=1}^{(n-1)/2} e_{2i-1} \Big) b_2 \Big( \prod_{i=1}^{(n-1)/2} e_{2i} \Big) b_1 = \beta_{12}^{\text{o}} \Big( \prod_{i=1}^{(n-1)/2} e_{2i} \Big) b_1,
\tag{9.17a}
$$

$$
\Big( \prod_{i=1}^{(n-1)/2} e_{2i-1} \Big) b_2 \Big( \prod_{i=1}^{(n-1)/2} e_{2i} \Big) b_1 \Big( \prod_{i=1}^{(n-1)/2} e_{2i-1} \Big) b_2 = \beta_{12}^{\text{o}} \Big( \prod_{i=1}^{(n-1)/2} e_{2i-1} \Big) b_2,
\tag{9.17b}
$$

which remove blobs by pairs as follows:

$$
\text{(diagram)} = \beta_{12}^{\text{o}} \ \text{(diagram)} \, , \qquad \text{(diagram)} = \beta_{12}^{\text{o}} \ \text{(diagram)} \, .
\tag{9.18}
$$

In the XXZ spin-chain, $b_1$ and $b_2$ are represented [31] by the matrices:

$$
\mathsf{X}_n(b_1) = \frac{1}{2\mathrm{i}\sin(r_1\lambda)} \begin{pmatrix} -e^{-\mathrm{i}\lambda r_1} & \mathrm{i}e^{-\mathrm{i}\lambda r_{12}} \\ \mathrm{i}e^{\mathrm{i}\lambda r_{12}} & e^{\mathrm{i}\lambda r_1} \end{pmatrix} \otimes \underbrace{\mathbb{I}_2 \otimes \cdots \otimes \mathbb{I}_2}_{n-1},
\tag{9.19a}
$$

$$
\mathsf{X}_n(b_2) = \frac{1}{2\mathrm{i}\sin(r_2\lambda)} \underbrace{\mathbb{I}_2 \otimes \cdots \otimes \mathbb{I}_2}_{n-1} \otimes \begin{pmatrix} e^{\mathrm{i}\lambda r_2} & \mathrm{i} \\ \mathrm{i} & -e^{-\mathrm{i}\lambda r_2} \end{pmatrix}.
\tag{9.19b}
$$

With (2.16), these matrices satisfy the defining relations (2.9), (9.6), (9.13), (9.15) and (9.17), with the fugacities of the loops containing blobs parameterised by

$$
\beta_1 = \frac{\sin\big((r_1+1)\lambda\big)}{\sin(r_1\lambda)}, \qquad \beta_2 = \frac{\sin\big((r_2+1)\lambda\big)}{\sin(r_2\lambda)},
\tag{9.20}
$$

and

$$
\beta_{12}^{\text{e}} = \frac{\sin\big((r_1+r_2-r_{12}+1)\frac{\lambda}{2}\big) \sin\big((r_1+r_2+r_{12}+1)\frac{\lambda}{2}\big)}{\sin(r_1\lambda)\sin(r_2\lambda)},
\tag{9.21a}
$$

$$
\beta_{12}^{\text{o}} = \frac{\cos\big((r_1-r_2-r_{12})\frac{\lambda}{2}\big) \cos\big((r_1-r_2+r_{12})\frac{\lambda}{2}\big)}{\sin(r_1\lambda)\sin(r_2\lambda)}.
\tag{9.21b}
$$

The standard modules over this algebra are characterised by the number $d$ of defects and two labels b or u, one for each boundary. For $d \geqslant 1$, there are four sectors, $\mathsf{V}_{n,d}^{(2,\text{b},\text{b})}$, $\mathsf{V}_{n,d}^{(2,\text{b},\text{u})}$, $\mathsf{V}_{n,d}^{(2,\text{u},\text{b})}$ and $\mathsf{V}_{n,d}^{(2,\text{u},\text{u})}$. Arcs to the left of the leftmost defects are decorated by a round blob or unblob if they are not overarched. In the same way, arcs to the right of the rightmost defect are decorated with square blobs or unblobs. For instance, here are the link states for $\mathsf{V}_{4,2}^{(2,\text{u},\text{b})}$:

$$
\text{(link states diagram)} \, .
\tag{9.22}
$$

For $d = 0$, there is a single sector $V_{n,0}^{(2)}$. All the arcs that are not overarched by larger ones are equipped with two decorations: a circle and a square, each one either black or white. This is the only sector wherein $\beta_{12}^{\mathrm{e}}$ comes up. For $d = 1$, the unique defect has two decorations, one from each boundary, and $V_{n,1}^{(2,\mathrm{b},\mathrm{b})}$ is the only sector wherein $\beta_{12}^{\mathrm{o}}$ comes up.

The conformal characters corresponding to the scaling limit of the standard modules are denoted $Z_d^{(2,\mathrm{b},\mathrm{b})}(q)$, $Z_d^{(2,\mathrm{b},\mathrm{u})}(q)$, $Z_d^{(2,\mathrm{u},\mathrm{b})}(q)$ and $Z_d^{(2,\mathrm{u},\mathrm{u})}(q)$ for $d \geqslant 1$, and $Z_0^{(2)}(q)$ for $d = 0$. In [31], the authors investigate the case where $n$ is even. With an argument that uses modular invariance, they conjecture the conformal characters in this case:

$$Z_d^{(2,\mathrm{b},\mathrm{b})}(q) = \frac{q^{-c/24}}{(q)_\infty} \sum_{k=0}^{\infty} q^{\Delta_{r_1+r_2-1-2k,r_1+r_2-1+d}}, \tag{9.23a}$$

$$Z_d^{(2,\mathrm{b},\mathrm{u})}(q) = \frac{q^{-c/24}}{(q)_\infty} \sum_{k=0}^{\infty} q^{\Delta_{r_1-r_2-1-2k,r_1-r_2-1+d}}, \tag{9.23b}$$

$$Z_d^{(2,\mathrm{u},\mathrm{b})}(q) = \frac{q^{-c/24}}{(q)_\infty} \sum_{k=0}^{\infty} q^{\Delta_{-r_1+r_2-1-2k,-r_1+r_2-1+d}}, \tag{9.23c}$$

$$Z_d^{(2,\mathrm{u},\mathrm{u})}(q) = \frac{q^{-c/24}}{(q)_\infty} \sum_{k=0}^{\infty} q^{\Delta_{-r_1-r_2-1-2k,-r_1-r_2-1+d}}. \tag{9.23d}$$

and

$$Z_0^{(2)}(q) = \frac{q^{-c/24}}{(q)_\infty} \sum_{k\in\mathbb{Z}} q^{\Delta_{r_{12}-2k,r_{12}}}. \tag{9.24}$$

The case of $n$ odd is not discussed in that paper. We formulate the following conjecture: For $n$ odd, (9.23) holds for all sectors except for $Z_1^{(2,\mathrm{b},\mathrm{b})}(q)$. This last conformal character remains unknown.

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
