# Peer review of "Two-point boundary correlation functions of dense loop models"

_SciPost Physics, doi:SciPost Phys. 4, 034 (2018)_

## Round 1 · Referee Report · Anonymous (Referee 2) · 2018-2-21

Strengths

  1. The authors make the aim of their paper clear from the very beginning.
  2. They set up the problem in a clear fashion and give some simple examples to illustrate mathematical objects they will be using.
  3. Despite the technical calculations, Sections 3 and 4 were still quite readable. The authors allowed the reader to get a sense of the methods used without having to understand all the finer points.

Weaknesses

  1. I felt that the conclusion of the paper itself introduced new ideas that could maybe be referred to earlier in the paper (eg. the relation between the projective modules (5.1)).

Report

In this paper the authors calculate correlation functions for critical dense polymers for six different types of boundary conditions and use these results to determine the conformal dimensions in the corresponding conformal field theory. They also find logarithmic behaviour in the correlation functions, as expected for these types of loop models. The authors then proceed to use knowledge of the characters in the conformal field theory to find results for the more general class of dense loop models. Overall the paper is well-written and clear in its set up and results. I recommend this paper for publication after the authors address the points listed under "requested changes."

Requested changes

Below I list some points the authors may wish to check.

  1. Authors may wish to fix one convention of either US or British English (eg. page 19 contains both "behaviour" and "behavior").

  2. Page 5, 1st sentence: rectangular lattice -> square lattice (as in the abstract)

  3. Page 5, 2nd sentence: 2mn -> mn tiles (since there are mn tiles on the mxn lattice)

  4. Page 5, equation (2.9): I think the first relation should be $(e_j)^2 = \beta e_j$

  5. Page 6, 2nd last sentence "We note that for $\gamma=0$, $v\dot v'$ is zero unless $v$ and $v'$ have the same number of defects": It seems that (2.14) has a factor of $\gamma^d$ in the case $d'=d$, which seems to indicate that the product is zero also in that case.

  6. Pages 6-8: I'm not sure why in equation (2.22) the authors flip the ket v' and attach it to v, but in (2.14) they do it the other way around. It seems an arbitrary choice, so I don't understand the motivation for having them different in the two cases. It also seems that in (2.29) the convention is that the ket is on the top (unflipped) and the bra is flipped, which corresponds to the description in (2.14) not (2.22).

  7. Page 11, after equation (2.44): have either $q^{1/2}+q^{-1/2}$ with parenthesis both or neither times.

  8. Section 3. Since the authors take $\beta=0$, it seems that $\ket{v_0}$ should be the zero vector, since $\bra{v_0}\ket{v_0}=0$. This can't be true, however, since then the righthand side of (3.17) would be zero.

  9. Page 14, equation (3.17). Should it be $O((\Lambda_1/\Lambda_0)^{m/2})$?

  10. Page 16, below (3.42): $x=rx'$ is repeated; I guess one should be $y=ry'$.

  11. Section 3.3-3.4: how do the authors know to compare equation (3.45) with (2.4), but (3.68) with (2.5)? I suppose it is because (2.4) and (2.5) are the only options for the form of the correlation function and so the approach is to match it with whichever is of the correct form. If that is the case (or otherwise), some extra explanation could be beneficial here.

  12. Page 28, 3rd sentence: Section 4.2 -> Section 4.1.

  13. Page 31, 2nd sentence after (4.33): the first "as" can be removed.

  14. Pages 43-44, equations (D.6) and (D.13): I think the relation $b_1 e_1 b_1 = \beta_1 b_1$ should be $e_1 b_1 e_1 = \beta_1 e_1$, similarly for the second relation in (D.13).

---

## Round 1 · Referee Report · Anonymous (Referee 1) · 2018-2-21

Strengths

  1. Largely analytic.
  2. The CFT arguments are quite general.

Weaknesses

None.

Report

The authors present a wealth of results for two-point boundary correlation functions of dense loop models. In particular, using a mapping onto the XX model and fermion techniques, they obtain expressions for 6 different types (a through f) of two-point correlators reproducing (either analytically or analytically with a final numerical evaluation) the known conformal weights -1/8, 0, -3/32, 3/8, 1 of critical dense polymers. They demonstrate that these results are entirely consistent with the predictions of logarithmic CFT including (i) the asymptotic growth (involving logarithms) of two point correlations due to nonunitarity (type a) and (ii) the compatability with the existence of Jordan cells and a Jordan partner field (type b). I believe these are the first such results obtained exactly starting with the lattice. This is a wonderful paper! I have no hesitation in recommending that it is accepted by SciPost in its current form.

Requested changes

None.

---

## Round 2 · Referee Report · Anonymous · 2018-3-7

Report

I am satisfied with the changes made by the authors and recommend this article for publication.

---

## Round 2 · Author Response

Dear editor,

We thank the referees for their reports on our article. Please find below a list of comments and changes made in response to the second referee’s report.

All the best,
Alexi Morin-Duchesne and Jesper Jacobsen

---

## Round 2 · List of Changes

——————

0. We have adapted the spelling to the best of our knowledge so that British English is used throughout.

1. 2. 3. We have made the changes requested by the referee.

4. Our claim was indeed incorrect. We have changed the text to remove this false statement.

5. The sentence just below (2.22) has been fixed.

6. The parentheses have been removed.

7. As explained above (2.21), for generic values of q and v a linear combination of link states, <v| is obtained by taking the dagger of |v>, with q mapped to q, and not to 1/q even if in the end we consider q on the unit circle. In section 3, this applies to the state v_0: <v_0| is then the transpose of |v_0>, with no complex conjugation. We indeed have <v_0|v_0> = 0, but because the entries of |v_0> are complex-valued, this does not imply that |v_0> = 0. No changes were made to the text.

8. 9. We have corrected the two typos.

10. Experience from other lattice models indicates that changes in boundary conditions often correspond to primary fields in CFT. Comparing our lattice result with the correlation functions of primary fields is indeed the first natural things to do. But in some cases, such changes in the boundary instead correspond to logarithmic fields or to compositions of primary fields, for instance.

To justify the type of correlator that arises from the lattice computation, one must dig deeper in the CFT arguments. This is precisely the goal pursued in Sections 4.1 and 4.2: to give the detailed CFT argument that explains why (3.45) and (3.68) must respectively be compared with (2.4) and (2.5). We have added a sentence at the beginning of Section 4 to make this clearer.

11. 12. 13. These typos have been fixed.

We have also performed these two extra changes:

(i) We have added a sentence in Section 4.2 about projective modules, announcing that these will be discussed further in the Conclusion.

(ii) We have added two new references in Section 3.1.

You are currently on this page

Resubmission 1712.08657v2 on 5 March 2018

---

## Editorial Decision

published